# Sea ice volume variability and water temperature in the Greenland Sea

Valeria Selyuzhenok[1,2], Igor Bashmachnikov[1,2], Robert Ricker[3], Anna Vesman[1,2,4], and Leonid Bobylev[1]

[1]Nansen International Environmental and Remote Sensing Centre, 14 Line V.O., 7, St.Petersburg, 199034, Russia
[2]the St. Petersburg State University, Department of Oceanography, 10 Line V.O, 33, St.Petersburg, 199034, Russia
[3]Alfred-Wegener-Institut, Helmholtz-Zentrum für Polar- und Meeresforschung, Klumannstr., 3d, Bremerhaven, 27570, Germany
[4]Arctic and Antarctic Research Institute, Bering str., 38, St.Petersburg, 199397, Russia

**Correspondence:** Valeria Selyuzhenok (valeria.selyuzhenok@niersc.spb.ru)

**Abstract.** This study explores a link between the long-term variations in the integral sea ice volume (SIV) in the Greenland Sea and oceanic processes. Using Pan-Arctic Ice Ocean Modelling and Assimilation System (PIOMAS, 1979-2016), we show that the increasing sea ice volume flux through Fram Strait goes in parallel with a decrease of SIV in the Greenland Sea. The overall SIV loss in the Greenland Sea comprises 113 km$^3$ per decade, while the total SIV import through Fram Strait increases by 115 km$^3$ per decade. An analysis of the ocean temperature and the mixed layer depth (MLD) over the climatic mean area of the winter marginal sea ice zone (MIZ) revealed doubling of the amount of the upper ocean heat content available for the sea ice melt from 1993 to 2016. This increase can solely explain the SIV loss in the Greenland Sea over the 24-year study period, even when accounting for the increasing SIV flux from the Arctic. The increase in the oceanic heat content is found to be linked to an increase in temperature of the Atlantic Water along the main currents of the Nordic Seas, following an increase of the oceanic heat flux form the subtropical North Atlantic. We argue that the predominantly positive winter North Atlantic Oscillation (NAO) index during the four recent decades, together with an intensification of the deep convection in the Greenland Sea, are responsible for the intensification of the cyclonic circulation pattern in the Nordic Seas, which results in the observed long-term variations of the SIV.

## 1 Introduction

The Greenland Sea is one of the key regions of deep ocean convection (Marshall and Schott, 1999; Brakstad et al., 2019), an inherent part of the Atlantic Meridional Overturning Circulation (AMOC) (Rhein et al., 2015; Buckley and Marshall, 2016). The intensity of convection is governed by buoyancy (heat and freshwater) fluxes at the ocean-atmosphere boundary, as well as oceanic buoyancy advection into the region. The freshwater is thought to play the principal role in long-term buoyancy balance of the upper Greenland Sea (Meincke et al., 1992; Alekseev et al., 2001a). The positive local precipitation-evaporation

exchange accounts for only 15% of the freshwater balance in the Nordic Seas. Approximately half of the fresh water anomaly in the Nordic Seas originates from the freshwater flux through Fram Strait, which forms by freshening of the upper ocean due to sea ice melt in the Arctic Ocean and by solid sea ice transport melting outside the Arctic Ocean (Serreze et al., 2006; Peterson et al., 2006; Glessmer et al., 2014).

The general surface circulation in the region is shown in Fig.1a. The upper 500 m in the western Greenland Sea is formed by mixing the Polar Water (PW), with temperature close to freezing and salinity from 33 to 34, and the Atlantic Water (AW), with temperature over 3 °C and salinity around 34.9 recirculating in the southern part of the Fram Strait (Moretskij and Popov, 1989; Langehaug and Falck, 2012; Jeansson et al., 2017). The maximum PW content is found in the upper 200 m of the Greenland shelf, it quickly decreases in the off shelf direction (Håvik et al., 2017). The AW is found below the PW. Its core is observed in
the seawards branch of the EGC, trapped by the continental slope. The central parts of the Greenland Sea represents a mixture of the AW and the PW with the Greenland Sea Intermediate Water (with temperature -0.4 – -0.8 °C and salinity ∼34.9). The core of the Greenland Sea Intermediate Water is found at 500-1000 m. The Greenland Sea Deep Water (with temperature -0.8 – -1.2 °C and salinity ∼34.9) is found below 1000 m. The latter two water masses are formed by advection of the intermediate and deep water, coming from the Arctic Eurasian basin through Fram Strait, mixed with the recirculating Atlantic Water by
winter convection (Moretskij and Popov, 1989; Alekseev et al., 1989; Langehaug and Falck, 2012). The convection depth in the Greenland Sea often exceeds 1500 m (Wadhams et al., 2004; Latarius and Quadfasel, 2016; Bashmachnikov et al., 2019).

The sea ice conditions in the Greenland Sea are defined by sea ice import through Fram Strait and by local ice formation and melt. The Fram Strait sea ice area (Vinje and Finnekåsa, 1986; Kwok et al., 2004) and volume flux (Kwok et al., 2004; Ricker et al., 2018) are primarily controlled by variations in the sea ice drift, which, in turn, are driven by the large-atmospheric
circulation patterns. Most of the variability of the atmospheric circulation and drift patterns is captured by the phase of the Arctic Oscillation (AO) or of its regional counterpart – the North Atlantic Oscillation (NAO) (Marshall et al., 2001). The positive AO (or NAO) phase intensifies northerly winds that drive more intensive ice transport through Fram Strait (Kwok et al., 2004). There is a moderate correlation (0.62) between NAO index (excluding extreme negative NAO events) and winter sea ice area flux through Fram Strait over 24 years of satellite observations (1978-2002) (Kwok et al., 2004). A higher correlation
(0.70) between NAO index and winter sea ice volume flux (2010-2017) is reported by Ricker et al. (2018). It is also argued that the interannual variations of the sea ice area flux through Fram Strait is even stronger linked to the Arctic Dipole pattern, since it explains a higher fraction of the observed interannual variations in the sea ice area flux than either the AO or the NAO (Wu et al., 2006). The Arctic Dipole pattern is derived as the second sea-level pressure EOF over the Arctic, which has two centers of action: over the Laptev-Kara seas and over the Canadian Archipelago. The pattern represents an important mechanism
regulating the ice export through Fram Strait (Wu et al., 2006).

The sea ice production in the Greenland Sea takes place east of the shelf between 71-75 °N and north of 75 °N within the highly dynamic pack ice transported southwards along the Greenland coast. The latter fills in cracks and leads and can reach considerable thickness. While the sea ice forming east of the shelf is mainly thin newly-formed ice. The highest interannual variations of sea ice area is observed between 71-75 °N (Germe et al., 2011). In the region the Odden sea ice tongue was
occasionally formed, a sea ice pattern extending eastwards from the east Greenland shelf northwest of Jan Mayen (Wadhams

et al., 1996; Comiso et al., 2001). The regression of the first empirical orthogonal function (EOF) of the sea ice extent to sea-level pressure shows a weak inverse relation with the NAO-like pattern with correlation coefficient -0.4. During the negative NAO phase, a reduction of the northerly wind, permits a more intensive westward Ekman drift of sea ice into the Greenland Sea interior which favours formation of the large Odden tongue (Shuchman et al., 1998; Germe et al., 2011). The Odden tongue area shows a strong negative correlation with the air temperature (-0.7) over Jan Mayen and with the local sea surface temperature (-0.9) (Comiso et al., 2001). Having stronger correlations with water temperature, the negative correlation of the sea ice area with the air temperature might be an artifact, as both are oppositely affected by the oceanic heat release to the atmosphere (Germe et al., 2011).

The ocean clearly plays an important role in the sea ice formation and melt in the region. In particular, it is speculated that the oceanic convection in the region favours a more intensive warm water flux from the south, affecting the air temperature and the sea ice extent (Visbeck et al., 1995). However, presently there is a lack of investigation linking oceanic processes with the sea ice variability in the Greenland Sea (Comiso et al., 2001; Kern et al., 2010).

Both sea ice area flux through Fram Strait and local sea ice processes in the Greenland Sea show changes over the recent decades. An overall reduction in sea ice extent is observed in the region since 1979 (Moore et al., 2015; Onarheim et al., 2018). In particular, a reduction in winter sea ice area is observed in the region of Odden ice tongue formation since 2000s (Rogers and Hung, 2008; Kern et al., 2010; Germe et al., 2011). Concurrently, an increase of the sea ice area flux through Fram Strait since 1979 was reported by Kwok et al. (2004); Smedsrud et al. (2017). A combined time series of of sea ice volume flux through Fram Strait (1990-1996 (Vinje et al., 1998), 1991-1999 (Kwok et al., 2004) and 2003-2008 (Spreen et al., 2009)) shows a shift towards lower fluxes in early 2000s compared to the 1990s (Spreen et al., 2009). However, the later study of Ricker et al. (2018) revealed that the sea ice volume flux in 2010-2017 is similar to that in 1990s. Due to different uncertainties in the data and different methodologies used in those studies, it is not possible to merge the results to get an uninterrupted data-set for the entire period from 1990 to 2017. Although individual studies do not reveal significant trends in the sea ice volume flux through Fram Strait, the overall tendency remains unknown.

In this paper we further explore a link between sea ice volume variability in the Greenland Sea and oceanic processes. The first objective is to estimate the sea ice mass balance in the Greenland Sea from local sea ice formation/melt and from sea ice advection in/out of the sea. We extend this analysis back to 1979 using the PIOMAS sea ice volume data. Further, we link the detected variations of sea ice mass balance to heat flux of the AW with the West Spitsbergen current (WSC) into the region.

## 2  Data

### 2.1  PIOMAS sea ice volume

PIOMAS (Pan-Arctic Ice Ocean Modeling and Assimilation System) is a coupled sea ice-ocean model developed to simulate Arctic sea ice volume. It assimilates NSIDC (National Snow and Ice Data Center) near-real time daily sea ice concentration, daily surface atmospheric forcing and sea-surface temperature in the ice-free areas from NCEP (National Centers for Environmental Prediction)/NCAR (National Center for Atmospheric Research) reanalysis (Zhang and Rothrock, 2003; Schweiger

et al., 2011). The PIOMAS provides monthly effective sea ice thickness (mean sea ice thickness over a grid cell) on a curvilinear model grid from 1978. A comparison of PIOMAS effective sea ice thickness with in situ, submarine and ICESat (Ice, Cloud, and land Elevation Satellite) data, mainly covering the western Arctic, showed that the PIOMAS uncertainty for monthly mean effective sea ice thickness does not exceed 0.78 m (Schweiger et al., 2011). The spatial pattern of PIOMAS ice thickness agrees
well with those derived from in situ and satellite data. The model overestimates the thickness of thin ice and underestimates the thickness of thick ice. Such systematic differences might affect long-term trends in sea ice thickness and volume. There is an indication that the PIOMAS shows a conservative sea ice volume trend (1979-2010) (Schweiger et al., 2011).

Since PIOMAS performance has not been assessed south of the Fram Strait, the first part of this study is devoted to inter-comparison of the PIOMAS sea ice thickness in the Greenland Sea with satellite data, as well as of the PIOMAS sea ice
volume flux through Fram Strait with observation-based flux values know from literature (Sect. 4.1 and 4.2). The original monthly PIOMAS sea ice thickness data were gridded to 25 km EASE-2 grid. The PIOMAS data were further used to derive time series of montly mean annual (September-August), mean winter (October-April) and mean summer (May-September) sea ice volume in the Greenland Sea for 1979 – 2016. The grid cell sea ice volume was computed as a product of PIOMAS effective sea ice thickness and the grid cell area.

## 2.2   AWI Cryosat-2 sea ice thickness

The PIOMAS effective sea ice thickness was inter-compared against sea ice thickness from Cryosat-2 satellite data-set (CS2, version 1.2, Ricker et al. (2014); Hendricks et al. (2016)) for the Greenland Sea region (see green box in Fig. 1). The CS2 data-set provides monthly average sea ice thickness on EASE-2 grid with 25x25 km spatial resolution from 2010 to 2017. Due to limitations of ice thickness retrieval from satellite altimetry, CS2 data-set used was limited only to the cold season
(October-April). The sea ice concentration data, provided along with CS2 thicknesses, was used to derive the effective sea ice thickness ($H_{eff}$) for the comparison with the PIOMAS data. The conversion was performed for each grid cell:

$$H_{eff} = HC \tag{1}$$

where H – CS2 sea ice thickness, C - sea ice concentration.

Uncertainties of CS2 ice thickness increase below 78°N due to sparse orbit coverage (Ricker et al., 2014). The CS2 retrieval
is based on sea ice freeboard measurements that are converted into sea ice thickness assuming hydrostatic equilibrium. Estimates of snow depth, required for the conversion, are based on the modified Warren climatology (Warren et al., 1999; Ricker et al., 2014). This climatology is not defined in the Fram Strait or Greenland Sea, therefore, snow depth estimates are extrapolated. Moreover, interannual variability in snow depth is not captured by the climatology, which can potentially cause biases in the final sea ice thickness retrieval. High drift speeds can also cause biases in the ice thickness retrieval due to the timing
of satellite passes within one month. The typical uncertainty is in the range of 0.3 - 0.5 m, but may potentially reach higher values.

## 2.3 ARMOR data-set

The long-term time series of water temperature at different depth levels and the mixed layer depth (MLD) were derived from the ARMOR data-set (http://marine.copernicus.eu/, 1993-2015). The data-set combines in situ temperature and salinity profiles with satellite observations and is constructed as the following. First, based on a joint analysis of the variations of satellite-derived anomalies (sea-surface temperature and sea-level from satellite altimetry) and of in situ thermohaline characteristics at different depth, linear multiple regressions are obtained. The regressions allow extrapolating satellite data from the sea-surface to standard oceanographic depth levels in a regular mesh of 1/4° x 1/4°, constructing the so-called "synthetic" vertical temperature and salinity profiles. The final monthly mean 3D temperature/salinity distributions are obtained through optimal interpolation of all in situ observations for this month together with the derived "synthetic" profiles, taken with different weights based on the inverse distance and type of measurement (in situ observations were given higher weights). (Guinehut et al., 2012). The number of in situ vertical temperature profiles in the MIZ area of the Greenland Sea (Fig. 1) is very limited. Between 1993 and 2016 the number of casts varies from 13 to 350 per year, with the median of 90 casts per year. Even less profiles are obtained in the Greenland shelf, which is out of the scope of this study. In the ARMOR dataset, use of satellite information provides a more precise and detailed picture of spatial and temporal variability of the thermohaline characteristics, than from interpolation of in situ profiles alone (as, for example, it is done in the World Ocean Atlas data-set, $https://www.nodc.noaa.gov/OC5/indprod.html$), and adds robustness to the results. The oceanic heat fluxes are estimated using currents from the ARMOR data-set with the same spatial and temporal resolution. The current velocities at various depth levels are obtained by extrapolating the sea-surface current from satellite altimetry, downwards using the thermal wind relations. The vertical density profiles, used for the computations, are assessed from the previously obtained temperature and salinity profiles (Mulet et al., 2012).

## 2.4 Long timeseries of water temperature of the West Spitsbergen Current

Long-term monthly gridded water temperatures were obtained from "The Climatological Atlas of the Nordic Seas and Northern North Atlantic" (Korablev et al., 2007). The data-base merges together data from ICES (International Counsel for Exploration of the Sea), from IMR (Institute of the Marine Research), from a number of international projects (ESOP, VEINS, TRACTOR, CONVECTION, etc.), as well as from Soviet Union cruises in the study region. Since there are too few observations in the EGC before the 2000s, we use long-term temperature time series in the much better sampled upper WSC (West Spitsbergen Current) at 78°N, west of East Fjord (Fig. 1b). The depth averaged water temperature at 100-200 m is used, as this layer is dominated by the AW and it is not directly affected by heat exchange with the atmosphere all year round. This results in the highest temperature at these depths during the cold season. Even this region was sampled in a quite irregular manner, with a lower sampling frequency in winter. Since 1979, the average number of samples was 161 per year, varying from, on average, 2-5 per year from November to May to 20-35 per year from June to October. The data-gaps in the time series were filled in by kriging with a 30-km window. The interannual variations presented in this study were averaged over the months with the densest data coverage (June to September).

## 3 Methods

### 3.1 Fram Strait and Denmark Strait sea ice volume flux from PIOMAS

The sea ice volume flux through Fram Strait was calculated as a product of monthly average PIOMAS effective sea ice thickness, area of the grid cell and the sea ice drift velocity (Ricker et al., 2018). The sea ice drift data was taken from the Polar Pathfinder Sea Ice Motion Vectors data set (version 3), distributed by the National Snow and Ice Data Center (NSIDC) (Tschudi and Maslanik., 2016). The data is provided on EASE-2 grid with 25x25 km spatial resolution. The gate was selected as a combination of a meridional section (82°N and 12°W - 20°E) and a zonal section (20°E and 80.5°N - 82°N), as suggested by Krumpen et al. 2016. (Fig. 1a). The location of the meridional gate at 82 °N was chosen to reduce biases and errors in sea ice drift that become larger with increasing velocities south of the gate (Sumata et al., 2014, 2015). The meridional and zonal sea ice volume flux, $Q_v$ and $Q_u$ correspondingly, were computed as:

$$Q_v = l/cos(\lambda)H(D_x sin(\lambda) - D_y cos(\lambda)) \tag{2}$$

$$Q_u = l/cos(\lambda)H(D_x cos(\lambda) - D_y sin(\lambda)) \tag{3}$$

where l = 25 km is the distance between 2 data-points, H is the PIOMAS effective sea ice thickness and $D_x$, $D_y$ represents sea ice drift velocity in *x* and *y* directions of the grid, respectively, and $\lambda$ is the longitude of the respective grid cell.

The total sea ice volume flux through Fram Strait ($QF$, positive – into the Greenland Sea) was obtained as a sum of the meridional and zonal fluxes along the gate:

$$QF = Q_u + Q_v \tag{4}$$

The total sea ice volume flux through Fram Strait was derived for the period from 1979 to 2016 for each month. A similar methodology was used to assess the sea ice volume flux through Denmark Strait ($QD$) along the meridional section (66°N and 35°W – 20°W). The positive sign of QD corresponds to a sea ice volume outflow from the Greenland Sea.

In order to assess the data quality, the resultant sea ice volume fluxes through Fram Strait gate at 82°N were inter-compared against available observation-based estimates in the Fram Strait (Kwok et al., 2004; Spreen et al., 2009; Ricker et al., 2018). The gate and the methodology used here were adopted from Ricker et al. (2018), while in the other two studies somewhat different methodologies and gate locations (Fig. 1a) were used. Each of the studies is also based on different data-sets of sea ice concentration (SIC), thickness (SIT) and drift (SID) (Table 1).

### 3.2 Greenland Sea sea ice mass balance

In order to analyse the sea ice volume lost or gained due to local melt or freezing, we calculated the sea ice mass balance (MB) in the Greenland Sea. It was derived for each month from 1979 to 2016 as:

$$MB = (V_m - V_{(m-1)})t - (QF_m - QD_m)t \tag{5}$$

where $V_m$ and $V_{(m-1)}$ are regional sea ice volume of the current *m*-th and previous *(m-1)*-th months, $QF_m$ and $QD_m$ are Fram Strait and Denmark Strait sea ice volume flux of the current *m*-th month, *t* - time period equal to 1 month. The regional sea ice volume was calculated for the area limited by 82°N and 66°N latitudes and by the boarder in the east shown in Figure 1a (green box). We slightly extended the eastern boundary of the Greenland Sea to the south-east, compared to its classical definition in order to include the entire area of the Odden ice tongue formation. The mass balance shows month-to-month increase or loss in sea ice volume within the Greenland Sea due to sea ice formation or melt. Positive MB values correspond to sea ice formation and negative values correspond to sea ice melt within the region. The monthly MB values were averaged over annual, winter and summer periods. Note that due to averaging negative annual values (sea ice volume loss, Fig.4) can occur due to both an increase in sea ice melt and a decrease in sea ice formation.

### 3.3 Mixed layer depth (MLD) and marginal ice zone (MIZ) ocean temperature

The MLD was derived using vertical profiles from the ARMOR data-set by the method of Dukhovskoy (Bashmachnikov et al., 2018, 2019). The method is similar to that used by Pickart et al. (2002), but is applied to the vertical profiles of the potential density gradients. Before processing, the small-scale noise in the potential density profiles were filtered out with 10-m sliding means. The gravitationally unstable segments were artificially mixed to neutral stratification. The MLD is defined as the depth where the vertical density gradient exceeds its two local standard deviations within a 50-m window, centered at the tested depth (see Bashmachnikov et al. (2018)). The visual control shows that the results are mostly similar to the widely used methods by de Boyer Montégut et al. (2004) and Kara et al. (2003), except for weakly stratified areas where the Dukhovskoy's method defines the MLD with higher accuracy. The obtained mean distribution of the MLD, seasonal and interannual variations of the MLD in the central Greenland Sea are consistent with observations (Våge et al., 2015; Latarius and Quadfasel, 2016; Brakstad et al., 2019). All the results show an increase of the convection depth from the mid-1990-s to the 2000-s. There are some minor differences in the absolute values of MLD which arise from the use of different data sets (e.g. Latarius and Quadfasel (2016) used only Argo floats) and methodologies for MLD detection. These minor differences do not break the tendency for the maximum winter MLD to increase since mid-1990s.

The position of the real MIZ strongly varies in time and along the EGC, being a function of local direction and intensity of sea ice transport by wind and current, variation in the characteristics of ice transport from the Arctic and interaction of ice floes, local ice thermodynamics, etc. Presence of melting sea ice, in turn, affects the upper ocean and air temperatures. A warmer winter ocean warms up the air, which can further be advected over the sea ice causing its melt away from the sea ice edge. Furthermore, an anomalously warmer ocean may prevent (or delay) formation of new ice. All these factors certainly affect the MIZ position. However, if we estimate ocean temperature variations only along the actual MIZ, we do not account for these effects. The considerations above show that defining the oceanic region directly and indirectly affecting the sea ice volume is not straightforward. In this study we examine interannual variations of ocean temperature in a fixed region, which is defined as an area enclosed between the 500-m isobath, marking the Greenland shelf break, and the mean winter location of the sea ice edge (Fig. 1). Using the fixed region also assures compatibility of interannual temperature variations. For the computations, the sea ice edge was defined as the 15% mean winter NSIDC sea ice concentration for 1979-2016. For brevity we further,

somewhat deliberately, call this region the MIZ area. We further will see that temperature trends remain positive and of the same order of magnitude all over the western Greenland Sea, except for a few limited areas along the shelf break. This assure robustness of the results to the choice of the study region.

### 3.4 Oceanic horizontal heat flux

The ARMOR data was used to derive a time series of oceanic heat flux into the Nordic Seas. Total oceanic heat flux through the Svinøy transect ($Q_{Svinoy}$) is calculated by integrating the heat flux values in the grid points:

$$Q_{Svinoy} = \int \int [\rho c_p (T - T_{ref}) v] d_x d_z \qquad (6)$$

where $\rho$ =1030 kg m$^{-3}$ is the mean sea water density; $c_p$ = 3900 J kg$^{-1}$ ° C$^{-1}$ is specific heat of sea water; $T$ is sea water temperature, $T_{ref}$ =-1.8°C is the "reference temperature" and $v$ is current velocity perpendicular to the transect. The reference temperature was set to sea ice melt temperature in order to investigate the contribution of ocean heat fluxes to sea ice melt.

### 4 Results

### 4.1 Assessment of PIOMAS-derived ice volume flux through Fram Strait and sea ice volume in the Greenland Sea

In order to assess the quality of the PIOMAS data, monthly effective sea ice thickness in the Greenland Sea was compared to that derived using the CS2 data-set (Fig. 2). In general, PIOMAS underestimates effective sea ice thickness compared to CS2 (Fig. 1b). The mean difference between PIOMAS and CS2 grid cell values is - 0.70 m. There are only two locations where PIOMAS shows thicker ice compared to CS2 – north of Spitsbergen and along the sea ice edge. On the other hand, CS2 also tends to overestimate sea ice thickness in the marginal ice zone (Ricker et al., 2017). The highest absolute differences between the data sets are attributed to the areas along the Greenland coast (dark blue) and north of Spitsbergen (dark red) (Fig. 1b). The monthly scatter plots (Fig. 2a-g) show that PIOMAS tends to overestimate thin sea ice and underestimate thick sea ice thickness, which is in agreement with the tendency reported for the central Arctic (Schweiger et al., 2011). This results in moderate correlations between the two data sets (0.63 < r < 0.77) for all winter months. The major discrepancies correspond to sea ice of 3 m and higher thickness, which form "tails" to the lower right corner of the scatter plots (Fig. 2 a-g).

PIOMAS sea ice volume flux through Fram Strait (October to April) was cross-compared with the fluxes derived using observation-based sea ice thickness data (see Tab.1). The analysis shows that PIOMAS-based sea ice volume flux is in good agreement with the estimates from other data sets (Fig. 3, Tab. 2). The correlation coefficients between the three data sets and PIOMAS are over 0.6. The highest correlation of over 0.8 with the Ricker et al. (2018) data can be explained by using identical gates and methodology for estimating ice volume fluxes (Fig. 1a). However, other statistical criteria (bias, relative percentage difference (RPD), root mean square error (RMSE), Table 2) indicate somewhat stronger mismatch between the PIOMAS and Ricker et al. (2018) estimates compared to those between PIOMAS and Kwok et al. (2004) or Spreen et al. (2009). The possible sources of this discrepancy are discussed in Sec. 5. Overall, PIOMAS shows lower sea ice volume fluxes compared to the observation-based estimates (Fig. 3c). The interannual variations in the PIOMAS monthly and total winter sea ice volume

flux agree well with other data-sets (Fig. 3a; Tab. 2). At intra-annual time scales all three data-sets show similar patterns with the minimum flux in October and maximum flux in March (Fig. 3b). Overall, moderate to high correlation between the data-sets, low relative variance and low bias (Tab. 2) suggest that PIOMAS provides a realistic estimate of seasonal and interannual variations of the winter sea ice volume flux through Fram Strait. Figures 12h and 13c suggest that PIOMAS correctly captures year-to-year variations of the mean effective sea ice thickness in the Greenland Sea and Fram Strait sea ice volume flux. This justifies using PIOMAS for analysing interannual variations of the integral sea ice volume over the Greenland Sea.

## 4.2 Interannual variations of sea ice flux through Fram Strait and sea ice volume in the Greenland Sea

The sea ice volume in the Greenland Sea derived from PIOMAS revealed statistically significant (at 99% confidence level) negative trends in monthly winter, summer and annual values (Fig. 4a, Tab. 3). The strongest negative trend of 84.8 km$^3$ per

58.2 km$^3$ per decade or 9.3% of long-term annual mean volume. The sea ice volume in the Greenland Sea shows an overall reduction by 72.4 km$^3$ or 11.5% of its long-term mean per decade.

The reduction of the sea ice volume in the Greenland Sea coincides with an increased sea ice volume import through Fram Strait by 9.6 km$^3$ per decade or 8.8% of its long-term mean (significant at 90% confidence level). Thus, the total increase in the sea ice volume imported to the Greenland Sea through Fram Strait is 115.2 km$^3$ per decade, which accounts for 18.2% of the Greenland Sea annual mean sea ice volume. The sea ice volume flux through Denmark Strait comprises about 2% (Fig. 3) of that through Fram Strait and shows no significant tendency. This flux has no considerable effect on the sea ice mass balance of the Greenland Sea.

A balance between sea ice volume import/export to the Greenland Sea through the straits and regional changes in the sea ice volume shows the volume of sea ice formed or lost due to thermodynamic processes within the region (Sec. 3.2). The sea ice mass balance in the Greenland Sea expressed in sea ice volume loss is shown in Fig. 4b. For about half of the years during the study period, sea ice volume loss in summer is higher than that in winter. However, there are a few years (1992, 1994, 2004-2007) when winter sea ice volume loss significantly exceeds the summer one. During these years an increased sea ice volume flux thought the Fram Strait is detected (Fig. 4c). There is a positive statistically significant trend in annual and summer monthly mean sea ice volume loss, while winter trend shows low statistical significance (Tab. 3). Overall, the monthly Greenland Sea sea ice volume loss increases by 9.4 km$^3$ per decade (Fig. 4, Tab. 3).

## 4.3 Interannual variations of water temperature and MLD in the MIZ of the Greenland Sea

In order to find the reason for the opposite trends the sea ice volume in of the Greenland Sea and the sea ice volume flux through Fram Strait, we investigate water temperature in the study region (Sec. 2.3, 3.3, 3.4). A relatively warm AW is observed in the East Greenland Current (EGC), off the Greenland shelf break, below a thin upper mixed layer dominated by the cold PW. Our estimates of winter MLD show that the AW should be regularly brought to the ocean surface by vertical winter mixing, which is consistent with observations (Håvik et al., 2017; Våge et al., 2018). The presence of the AW is observed in the climatology as water temperature (and salinity) in the EGC increasing with depth from about 0 °C near the sea-surface to 2-4°C at 500

m. In the 24-year means, the northern temperature maximum (Fig. 5a) results from recirculation of AW of the WSC in the southern Fram Strait, while the southern maximum is due to the northwards heat flux with the North Icelandic Irminger Current (NIIC) through Denmark Strait (Hansen et al., 2008; Ypma et al., 2019). The latter is a northern branch of the Irminger Current. The sea ice is affected by the heat in the upper mixed layer, the depth of which varies on synoptic, seasonal and interannual time scales. Our analysis shows that the obtained tendencies of increase of water temperature with time, derived in the next paragraphs, are largely independent from the choice of the water layer, at least within the upper 200 m of the water column. In further analysis we present results for the upper 50 m layer (the typical summer mixed layer in the MIZ) and the upper 200 m layer (the typical winter mixed layer in the MIZ, (Fig. 6c). In the annual means, the water temperature, averaged over upper 50-m layer of the MIZ, has a maximum of 2°C in September and decreases to 0.1-0.2°C in March-April. Averaged over the upper 200-m the patterns of the mean distribution and of (somewhat weaker) tendencies in temperature and salinity closely repeat those in Figure 15. When averaged over the fixed region, corresponding to the mean winter MIZ area (Fig.1), the mixed layer seawater temperature is always above the freezing point, i.e., overall, the ocean melts sea ice in this area all the year-round.

Figure 5a shows interannual variations of November 2 °C sea water isotherm (averaged over the upper 200-m layer). Water temperature in November reflects the heat fluxes accumulated during the warm period. It shows the background conditions at the beginning of the winter cooling, when sea ice start forming locally. From the 1990s to the 2000s the 2°C isotherm approached the shelf break. The largest westward propagation is observed in the WSC recirculation area (76-78°N) and northwest of Jan Mayen (70-73°N), in the southern Odden tongue region. The tendency of the isotherm to approach the shelf break is consistent for different isotherms (from 1 to 3°C), for different layer thickness (50 to 200 m), as well as for different months. Only for winter months, when the whole upper 200-m mixed layer effectively releases heat to the atmosphere, the interannual trends become insignificant. The linear temperature trend (Fig. 5b) shows warming in the whole area of the eastern MIZ. The strongest warming follows the pathway of the recirculating AW in the northern Greenland Sea (Glessmer et al., 2014; Håvik et al., 2017) which is known to strongly affect the central regions of the sea (Rudels et al., 2002; Jeansson et al., 2008). The warming in the northern Greenland Sea is linked to a strong warming of the WSC and of the Norwegian Atlantic Front Current (NwAFC), while that in the southernmost part of the sea – with the NIIC. Two exceptions can be noted: the northwestern part of the coastally trapped EGC (where negative trends are obtained in the area dominated by a colder PW outflow from the Arctic) and the area of the EGC recirculation into the Greenland Sea at 72-74°N extended from the continental shelf break to 8-9°W (here the tendencies in the upper ocean temperature are close to zero). The latter is the area where the Odden ice tongue starts spreading into the Greenland Sea interior (Germe et al., 2011). The decreasing temperature in both of these areas is consistent with a stronger sea ice/PW transport from the Arctic (Sec. 4.2).

With a stronger melting of sea ice at the seawards part of the MIZ, together with the ice volume loss, we should observe a sea ice area loss. This is consistent with Germe et al. (2011). In particular, positive water temperature trend over the eastern part of the Odden region suggest an overall decrease of the Odden formation by the end of the study period. The mean temperature trend over the Odden region (the area within the dotted line in Fig.5b) is 0.08 °C per year, i.e. there is an area-mean increase by 1.8°C from 1993 to 2016. This exceeds the mean ocean temperature increase, averaged in the MIZ area (Eq.7), which

includes the northern shelf break regions with negative temperature trends. Therefore, the estimates of the heat available for the ice melt, based on the values presented in Eq.(7), should be considered as the lower limit of the heat release within the Odden region.

Interannual variations of water characteristics, averaged over the upper 200-m and in the MIZ area, are shown in Figure 6. From 1993 an overall increase of annual mean temperature in the MIZ is observed, suggesting an increasing intensity of the sea ice melt. The temperature increases during all seasons, but the strongest increase is detected in autumn (by 0.5 and 0.6°C over the 24 years). The winter convection efficiently uplifts heat to the sea surface. The heat accumulated in summer is mostly released during winter. Figure 4d suggests that the results can be extrapolated back to, at least, 1980, as the slope of the trend lines in temperature of the advected AW for 1980-1992 is practically the same as for the period discussed above. We observe a growing difference between September and March temperatures (Fig. 6a) together with a decrease of interannual temperature trend to insignificant in winter. The growing difference in temperature is observed in spite of the equal winter and summer trends in the heat inflow with the NwAC (see $T_w$ and $Q_{Svinoy}$ in Tab.3). Therefore, in the MIZ region, all additional heat accumulated in the upper 200-m layer during summer is uplifted to the sea surface by winter convection, preventing ice formation in the ice-free areas or melting the ice in the ice-covered ones.

Not only the autumn temperature increases in the MIZ, but also the zonal thermal gradient across the MIZ increases 1.7 times from 1993 in the annual means (Fig. 6 b), and nearly 4 times in winter. This goes along with a decrease of the annual mean distance between the 2°C or 3°C isotherm and the shelf break (Fig. 6d): from 120 km in 1993 to 50 km in 2016 (see also Fig.5 a). The direct result of this is a faster melt of the sea ice episodically advected from the MIZ eastwards by EGC filaments and mesoscale eddies (Kwok, 2000; von Appen et al., 2018). These processes can transport sea ice dozens of kilometers eastward (von Appen et al., 2018). The most favourable conditions for eddy formation are observed during northerly winds. The eddies sweep sea ice and PW seawards and advect warm AW closer to the ice edge, resulting in increased bottom and lateral sea ice melt (Bondevik, 2011). However, a few episodic observations of the ice dynamics in the MIZ do not presently allow quantifying the importance of this effect.

The 24-year mean winter mixed layer depth (MLD) in the MIZ off the Greenland shelf vary from 120 m to 250 m with the mean value around 150 m, as derived from ARMOR data-set. Averaged over the MIZ, MLD increases from the mean value of 130 m in 1993 to around 180 m in 2016 (Fig. 6c). Since winter mixing does not reach the lower limit of the warm Atlantic water at 500-700 m, the deeper the mixing, the more heat is uplifted towards the sea-surface, melting the ice in the MIZ. The increase in MLD results from a higher upper ocean density due to increasing salinity of the AW, tempered by the increasing temperature (Fig. 5b,d), which is consistent with the findings of Lauvset et al. (2018). Given the increase in ocean temperature in the upper 200-m layer in the MIZ from 1.3°C in September 1993 to 1.8°C in September 2016 together with an increase in the mean winter MLD from 130 m in 1993 to 180 m in 2016, we can make a rough estimate of the increase (over the 24 years) in the heat released by winter MLD in the MIZ:

$$dQ = dQ_{2016} - dQ_{1993} = c_p * \rho_{water} * (1.8 * 180 - 1.3 * 130) * MIZarea \tag{7}$$

where $c_p$ = 3900 J $°C^{-1}$ $kg^{-1}$, $\rho_{water}$ = 1030 kg m$^{-3}$ and the MIZ area is estimated as 2.3 10$^{11}$ m$^2$. The computations give an additional heat release of 1.5 10$^{20}$ J, following the observed water temperature seasonal cycle, we assume that all the heat from the growing winter MLD is released at the sea-surface. If all this heat would go to melt ice in the MIZ, we get an increase in the sea ice volume loss during winter of:

$$dV = dQ/(L * \rho_{ice}) \approx 500km^3 \tag{8}$$

where the specific heat of ice fusion L=3.3 10$^5$ J kg$^{-1}$ and the ice density of $\rho_{ice}$ = 920 kg m$^{-3}$ (Petrich and Eicken, 2010). This far exceeds the observed sea ice volume loss in the region (SIV loss monthly winter trend * 12 month * 24 years $\approx$ 200 km$^{-3}$). Certainly, not all heat released by the upper ocean in the MIZ area goes to the ice melt. An unknown fraction of heat is directly transferred to the atmosphere through open water, ice leads or is advected away from the MIZ area by ocean currents and eddies. The sea ice melt may additionally increase haline stratification at the lower boundary of the ice, preventing ocean heat to reach the ice cover. However, the estimates above suggest that the autumn warming of the upper MIZ region, limited from below by the winter mixed layer, is able to release more than enough heat to account for the observed reduction of sea ice volume in the region.

## 5 Discussion

### 5.1 PIOMAS-derived trends

The revealed regional trends in sea ice volume rely on the PIOMAS model data. A comparison of interannual variations of PIOMAS regional sea ice thickness and the sea ice volume flux through Fram Strait showed that PIOMAS estimates are in agreement with the observation-based estimates during the recent decades. However, the PIOMAS systematic overestimation of thin ice and underestimation of thick ice thickness, reported for the central Arctic, affects the long-term volume trend (Schweiger et al., 2011). Schweiger et al. (2011) conclude that the PIOMAS-based volume trend is lower than the actual one. Given that similar systematic errors in effective sea ice thickness are found for the Greenland Sea (Fig. 2), it is likely that the derived Greenland Sea sea ice volume trend is underestimated. The PIOMAS Fram Strait sea ice volume flux can also be affected by these systematic errors. The model studies show three major positive peaks in the Fram Strait sea ice volume flux since 1979: 1981-1983, 1989-1990, 1994-1995 (Arfeuille et al., 2000; Lindsay and Zhang, 2005). The anomaly in 1989-1990 was caused by an increase in the thickness of the transported sea ice, while the anomaly in 1994-1995 was due to an intensification of southward sea ice drift (Arfeuille et al., 2000). The reduction of Arctic multiyear ice fraction during the late 1980s – early 1990s (Comiso, 2002; Rigor and Wallace, 2004; Yu et al., 2004; Maslanik et al., 2007) are in line with this finding. The sea ice volume flux through Fram Strait derived from PIOMAS shows the peaks in 1981-1985 and 1994-1995, but does not capture the anomaly of 1989-1990 (Fig.4c). During this period there is no significant shift in the PIOMAS effective sea ice thicknesses in the Fram Strait which is likely caused by the PIOMAS systematic errors which smoothed the differences in thickness between thick and thin ice. Since 1993, the PIOMAS Fram Strait sea ice volume flux correlates well with the observation-based fluxes (Fig. 3). The main sources of relative errors between the Fram Strait volume flux estimates can

be related to the different choice of methodologies, data-sets and gates used to derive sea ice volume fluxes (Tab.1, Fig.1) Lower PIOMAS-based sea ice volume flux can be attributed to the discussed above general PIOMAS tendency to underestimate sea ice thickness. Fig. 1b shows that for the entire meridional 82 °N gate, which is the main gate for sea ice import to the Greenland Sea, the PIOMAS effective sea ice thickness is lower compared to the CS2 effective thickness. In addition, the NSIDC sea ice drift shows lower speed compared to the OSI SAF drift used in Ricker et al. (2018). A combination of lower drift speed with thinner ice thickness might be the reason of the largest offset (Tab.2, Fig. 3) between the PIOMAS-based Fram Strait sea ice volume fluxes and those derived in Ricker et al. (2018).

## 5.2   Link to the variability of ocean temperature and atmospheric forcing

The revealed decrease in sea ice volume in the Greenland Sea goes in parallel with an increase in the ice volume inflow through Fram Strait. As the sea ice volume flux through Denmark Strait does not show any significant change, this indicates a simultaneous intensification of the processes of ice melt and reduction in sea ice formation in the sea. The latter is supported by the highest negative trends in the sea ice area (Fig. 1, expressed in SIC trend) in the area of the Odden tongue between 73 and 77°N

The interannual variations in sea ice area were previously linked to variations in air temperature (Comiso et al., 2001). The results of our paper permitted to speculate, that ocean temperature may be important in controlling Odden formation (see also Shuchman et al. (1998); Germe et al. (2011)). E.g. the reduction of Odden tongue occurrence in 2000s (Latarius and Quadfasel, 2010) might be partially driven by the increase in upper ocean heat content (Fig.5b)

The atmospheric heat convergence over the Greenland Sea is estimated as the sum of atmospheric heat fluxes across the northern, southern, eastern and western boundaries of the Greenland Sea (positive fluxes are in the study region), using ERA-Interim reanalysis. On average, from October to April next year, we obtained always negative atmospheric heat convergence over the Greenland Sea (1000 to 900 GPa) of -120 TW on average, varying from -170 to -90 TW. The sign is consistent with winter typical winds from the Arctic or Greenland (see, for example, Germe et al. (2011)), being warmed while passing over the region. The negative atmospheric heat convergence is roughly balanced by the integral heat release from the ocean to the atmosphere over the same area on the order of +130±40 TW, assuming the regional mean winter heat release by the ocean of 150±50 W m$^{-2}$ (Moore et al., 2015).

The heat convergence has tendency to decrease in absolute value from 1993 to 2016 by about 4 TW, accompanied by a rise of the area-mean winter air temperature by about 1°C. The oceanic southwards heat advection through 77.5°N in the upper 200-m layer increases by 1 TW. The source of the atmospheric warming lies possibly in the northwest, in the south-eastern Fram Strait, – a known region of high oceanic heat flux into the atmosphere (see, for example, Dukhovskoy et al. (2006)).

We argue that at least the overall sea ice volume loss from 1993 to 2016 is governed by the ocean. The surplus of the amount of the heat, released by the ocean at end of the study period, is more than twice of that necessary for bringing up the observed sea ice volume loss, even when accounting for the detected increase in the sea ice volume import through Fram Strait. Heat loss to the atmosphere and the neighboring ocean areas should take up the rest of the heat. In particular, the observed increase

of ocean temperature over the Greenland Sea (Fig. 5b) may be a reason for a corresponding increase in the air temperature, used for explaining negative trends in the sea ice area (Comiso et al., 2001).

The observed trends are due to both, the increase in temperature of the AW in the MIZ, as well as an increase in winter MLD in the area, bringing more AW to the surface. A significant vertical extent of the warm subsurface AW layer, going down to 500-700 m depth (Håvik et al., 2017), results in a higher ocean heat release for a stronger mixing for the observed MLD in the MIZ. A similar mechanism was suggested for the Nansen Basin of the Arctic Ocean, where an enhanced vertical mixing through the pycnocline is thought to decrease the sea ice area in the basin (Ivanov and Repina, 2018).

In turn, the subsurface AW in the EGC is fed by the recirculation of the surface water of the WSC, an extension of the Norwegian Atlantic Front Current (NwAFC) and the Norwegian Atlantic Slope Current (NwASC). The recirculation is mostly driven by eddies (Boyd and D'Asaro, 1994; Nilsen et al., 2006; Hattermann et al., 2016). The interannual variations in the vertical mixing intensity between the AW, the PW and the modified AW, returning from the Arctic through the southern Fram Strait, as well as variations in ocean-atmosphere exchange in that area leads to interannual variability of the AW advected by the EGC into the Greenland Sea (Langehaug and Falck, 2012). All the processes intensify during highly dynamic winter conditions. Nevertheless, interannual correlation of the summer upper ocean water temperature (0-200 m), spatially averaged over the MIZ area, with that in the upper WSC is 0.8-0.9. Further south, correlation of interannual variations of the MIZ temperature with that of the NwAFC (NwASC) or with the heat flux across the Svinøy section are low. The decrease is due to damping of the advected heat anomalies in the Norwegian Sea by eddy heat transport and ocean-atmosphere exchange (Asbjørnsen et al., 2019). Besides differences in local forcing, regional atmospheric forcing over the northwestern Barents Sea regulates the interannual variations of the heat re-distribution between the WSC and the Barents Sea (Lien et al., 2013), further decreasing the correlations.

Nevertheless, in a long run (during four recent decades), temperature at the WSC, the NwAFC, NwASC and the heat flux across Svinøy section all show positive trends (Fig. 4, 5). This is confirmed by a number of studies (Alekseev et al., 2001c; Piechura and Walczowski, 2009; Beszczynska-Möller et al., 2012). The trends form a part of the long term oscillation of water temperature in the Norwegian Current (Yashayaev and Seidov, 2015).

Pressure fields in the nothern north Atlantic are mainly governed by NAO and the East Atalantic patterns (Woollings et al., 2010; Moore and Renfrew, 2012; Foukal and Lozier, 2017). Bioth patterns affect the wind stress curle, largely regulating ocean circulation in the Nordic Seas. During the positive NAO phase, the cyclonic atmospheric circulation over the Nordic Seas intensifies (Skagseth et al., 2008; Germe et al., 2011). This leads to stronger northerly winds along the Greenland shelf, as well as stronger southerly winds along the Norwegian coast, which results in a more intensive cyclonic oceanic circulation in the Nordic Seas (Schlichtholz and Houssais, 2011). Several regional studies, based on in situ data, demonstrate a higher intensity of oceanic transport of volume and heat along the AW path towards the Fram Strait during the positive NAO phase (Raj et al., 2018; Walczowski, 2010; Chatterjee et al., 2018). Thus, change from strongly negative to strongly positive NAOI results in the NwASC volume inflow to the Nordic Seas to increase by 50%, as well as the oceanic heat flux (Skagseth et al., 2004, 2008; Raj et al., 2018). We obtained a significant correlation between NAOI and oceanic heat advection with the Norwegian current at Svinoy (0.5 for the heat flux integrated over the upper 500-m layer). The link between the AW transport by the WSC, as

well as the cyclonic circulation in the Greenland Sea, and NAO phase is also obtained from observations and numerical models (Walczowski, 2010; Chatterjee et al., 2018). Correlation of NAOI with southward heat flux in the Fram recirculation is also positive, but not significant (0.3). The intensity of the flux may be damped by non-linear dependence due to the AW enters the resirculation as eddy shedding. Additionally, observations demonstrate that the positive NAO phase drives a stronger ice drift
through Fram Strait (Vinje and Finnekåsa, 1986; Koenigk et al., 2007; Giles et al., 2011; Köhl and Serra, 2014), a stronger EGC (Blindheim et al., 2000; Kwok, 2000), and a typically lager extension of Odden ice tongue (Shuchman et al., 1998; Germe et al., 2011). The stronger PW transport also dams the AW anomalies, entering into the study region.

NAO phase is showed to be the main driver for interannual variations of sea ice volume flux to the Greenland Sea (Germe et al., 2011; Ricker et al., 2018). The simultaneous long-term (1974-1997) intensification of the AW inflow in the Nordic Seas
across the Faroe-Shetland Ridge, and of eastwards advection of PW to the southwestern Norwegian Sea, as a response to NAO forcing has been noted in several studies (see, for example, Blindheim et al. (2000); Yashayaev and Seidov (2015). The long-term variations in the NAO index go in parallel with those in the Atlantic Multidecadal Oscillation (AMO), at least during the latest 70 years (Yashayaev and Seidov, 2015). This suggests that the positive phase of NAO corresponds, in the long-term tendency, to the positive phase of AMO, i.e. the higher water temperature in the North Atlantic. Both tendencies lead to higher
heat fluxes into the Nordic Seas.

From the beginning of the 1970s the winter NAO index is growing. From 1979 to 2016 it is mostly positive (Fig. 7), although an overall winter trend can be separated into an increase from 1979 to 1994, a rapid drop from 1995 to 1996 and an increase from 1996 to 2016. The NAO index drop in 1995-1996 coincides with a drop in regional sea ice volume loss and a decrease in the WSC water temperature (Fig.4 b,d). This can be related to the minimum heat flux through the Svinøy section
in 1994 (Fig. 4,d). The time needed for water properties to propagate from Svinøy to the Fram Strait with the NwAC is on the order of 1.5-2 years (Walczowski, 2010).

Summer NAO index does not govern the interannual variations of the atmospheric system, as well as in the oceanic ones (circulation in the Nordic Seas intensifies in winter and is thought to bring more AW to the recirculation region compared to that in summer). Consistent with other studies of seasonal interannual variations of current intensity in the region, our
results suggest that these are winter variations of the AW transport that bring up the interannual variations of the subsurface water temperature in the MIZ of the Greenland Sea. The decreasing summer NAO index from 1979, may be responsible for a somewhat stronger tendency in the SIV loss in winter, compared to summer (Fig. 4a,b).

Summing up, the positive phase of NAO intensifies the whole current system of the Nordic Seas, simultaneously intensifying sea ice flux through Fram Strait and the northward heat flux with the AW to the Nordic Seas. In this paper we demonstrated that
the intensification of the AW heat inflow contributes to variations of the sea ice volume in the Greenland Sea. This supplements previous results, showing that the AW inflow dominates the oceanographic conditions over the upper Greenland Sea, except for the shelf area (e.g. Alekseev et al. (2001b); Marnela et al. (2013)).

In spite of the stronger ice melt, the upper ocean salinity in MIZ, as well as along the EGC and in the NwAC, has increased during recent decades (Fig. 5d). We relate salinification in the MIZ area of the upper Greenland Sea to a stronger flux of the
AW and more intensive winter mixing. These effects override the additional freshwater input from the ice melt. Oppositely,

during freshening of the upper Greenland Sea, the Great salinity anomaly 1966-1972, more ice was observed in the MIZ region – the Odden ice tongue was pronounced (Rogers and Hung, 2008). This confirms the reverse relation between the sea ice extent and the MIZ salinity in the Greenland Sea and their dependence on interannual variations of the intensity of the AW advection.

Another possibly not independent mechanism is linked to the intensity of the deep convection in the Greenland Sea (Fig. 7).

A more intense convection, governed by thermohaline characteristics of the upper Greenland Sea, the sea ice extent and the intensity of ocean-atmosphere heat and freshwater exchange (Marshall and Schott, 1999; Moore et al., 2015), lowers the sea-level in the Greenland Sea (Gelderloos et al., 2013; Bashmachnikov et al., 2019). This in turn increases the cyclonic circulation in the region. This effect works together with NAO forcing. Deep convection in the Greenland Sea shows a consistent increase from about 1000 m in the beginning of the 1990s to about 1500-2000 m during 2008-2010, after which a certain tendency

to decrease is noted (Bashmachnikov et al., 2019). The on-going increase in salinity of the upper Greenland Sea (Fig. 5d) during the recent decades favours deeper convection (see also Lauvset et al. (2018); Brakstad et al. (2019)). Satellite altimetry data show that, during the same period, the area-mean cyclonic vorticity over the Nordic Seas has grown by about 10%. The circulation increase is also consistent with the detected intensification of the AMOC after its minimum in the 1980s (Rahmstorf et al., 2015). However, during the latest decade a stagnation or a possible reversal of the tendency is observed (Smeed et al.,

15  2014).

## 6   Conclusions

Using PIOMAS sea ice volume data we derived trends in the mean annual, winter and summer sea ice volume (SIV) in the Greenland Sea and the sea ice volume flux (SIF) through Fram Strait from 1979 to 2016. Taking into account the SIV inflow and outflow through Fram and Denmark Straits, the thermodynamic SIV loss within the Greenland Sea was derived. We found

an increase in monthly SIV loss by 9.4 $km^3$ per decade. From 1979 to 2016 the overall SIV loss comprises $\sim 270$ $km^3$, in spite of an increase in SIF of $\sim 280$ $km^3$ during the same time period. However, those PIOMAS-based trends should be treated cautiously. The absence of positive anomaly in PIOMAS-based SIF in 1989-1990 indicate that the PIOMAS underestimate thickness of thick sea ice in the Fram Strait and in the Greenland Sea. The biases might lead to a weaker long-term SIF trend, while the SIV trend may be stronger.

Our analysis of the upper ocean water properties in the marginal sea ice (MIZ) zone of the EGC, shows a notable increase of the Atlantic Water (AW) temperature below the pycnocline, as well as of winter mixed layer depth from 1993 to 2016. These changes result in a higher sea-surface heat release, providing twice the amount of the heat needed for bringing up the observed SIV loss. This suggests that, the long-term variations of the heat flux entering the Nordic Seas, advected northwards with the NwAC as the AW and, further on, with the WSC into the MIZ largely contribute to the corresponding long-term SIV variations

in the Greenland Sea. The analysis of marginal sea ice zone (MIZ) ocean parameters showed an increase in mixed layer depth (MLD) and its temperature from 1993 to 2016. The estimated amount of additional oceanic heat released from 1993 to 2016 surplus the amount of heat necessary for bringing up the observed SIV loss. Therefore, we state that the AW advection into the MIZ largely contributes to the SIV loss. We suggest that the simultaneous tendencies in the long-term increase of SIF and of

the AW transport are both linked to a higher intensity of atmospheric circulation during the positive NAO phase, and, possibly, to the positive AMO phase, often linked to the intensification of the AMOC since the 1980s. Not being independent, both mechanisms finally lead to a decrease of SIV in the western Greenland Sea.

*Acknowledgements.* The research was supported by RSF, project No. 17-17-01151.

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

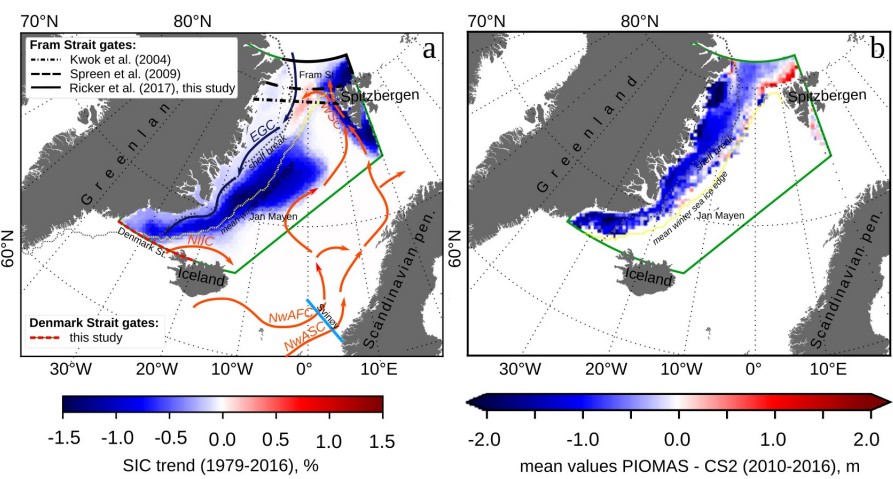

**Figure 1.** The study region is marked with the green box: a - linear trends in the mean October-April NSIDC sea ice concentration (SIC) over the period 1979-2016 (Comiso, 2015). The black lines show gates used for estimation of the sea ice volume flux through Fram Strait. Mean winter sea ice edge is shown in dash yellow, the shelfbreak (500-m isobath) is shown in dash grey. EGC is the East Greenland Current, NIIC – the North Icelandic Irminger Current, NwAFC – the Norwegian Atlantic Front Current, NwASC – the Norwegian Atlantic Slope Current, WSC – the West Spitsbergen Current; b - mean difference between mean PIOMAS and CS2 effective sea ice thickness (m) for October- April, 2010-2016.

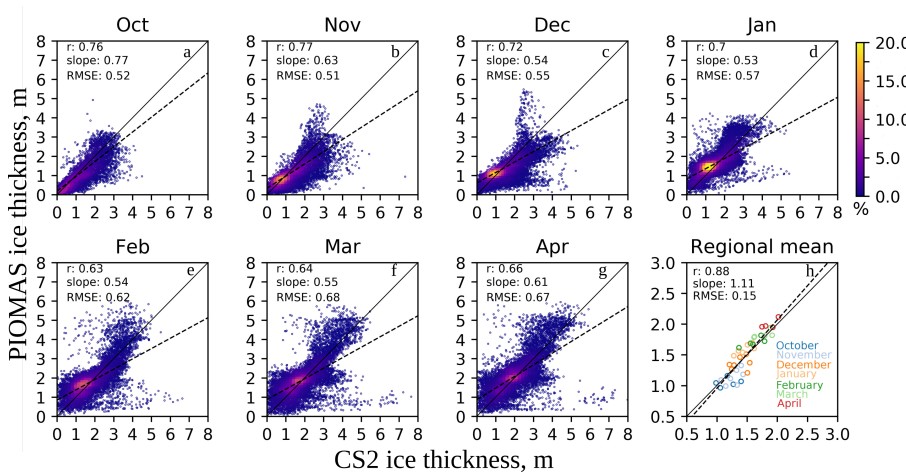

**Figure 2.** Density scatter plots of PIOMAS and CS2 monthly effective sea ice thickness (m) in the Greenland Sea, October-April 2010-2016: (a-g) - each point corresponds to one grid-cell sea ice thickness; (h) mean monthly sea ice thickness over the ice covered area of the Greenland Sea for all inter-compared snapshots. The color of the points in panel h corresponds to a month. The dashed lines show the linear regression fit and the solid lines are 45° angles. The correlation coefficients (r), the slope of the linear regressions and the root-mean-square error (RMSE) are given in the upper left corner.

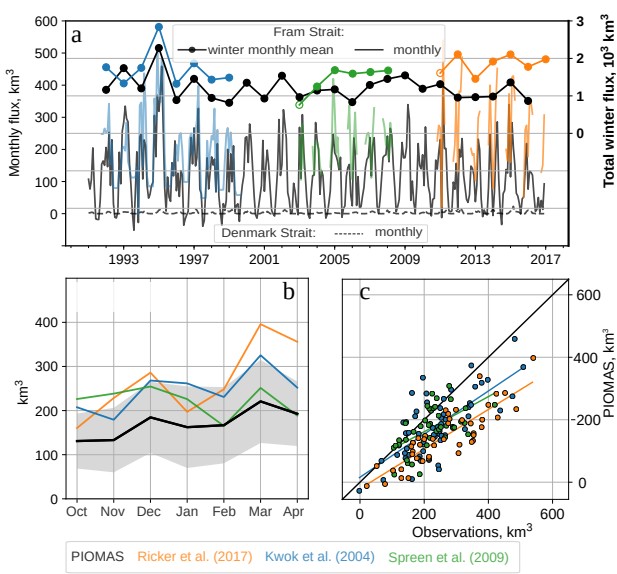

**Figure 3.** Sea ice volume fluxes (km³): a - time series of PIOMAS and observation-based monthly sea ice volume fluxes through Fram and the Denmark Straits, 1991-2016 (note that the total winter fluxes are referenced to the right scale). Empty circles indicate seasons with an incomplete winter cycle: b - winter intra-annual cycle sea ice volume flux through Fram Strait, averaged over the period of the observations and over 1991-2016 for PIOMAS data-set. The gray background color correspond to one standard deviation interval from the PIOMAS mean; c - scatter diagram of monthly mean PIOMAS sea ice volume fluxes through Fram Strait versus monthly mean observations.

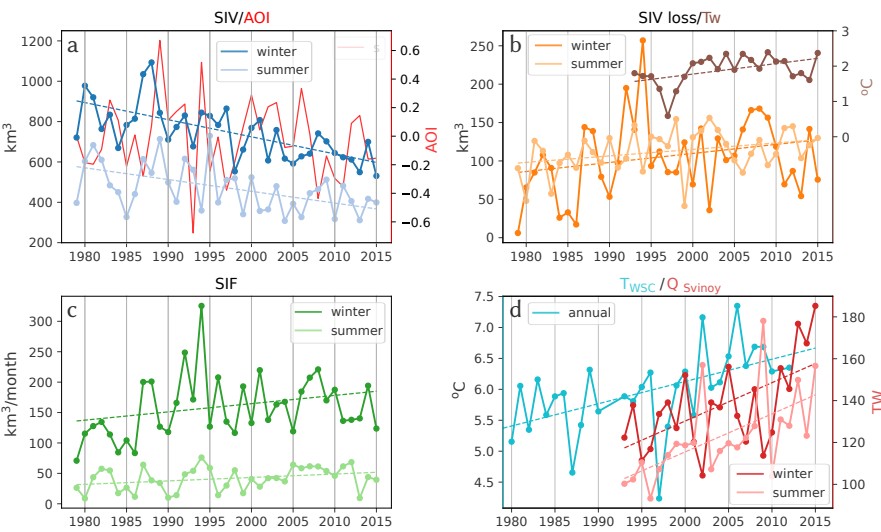

**Figure 4.** Time series of winter (December-April) and summer (May-November) and annual ice-ocean-atmosphere characteristics in the Greenland Sea: (a) monthly mean PIOMAS sea ice volume (SIV, km$^3$) and monthly summer AO index (AOI), (b) monthly mean PIOMAS sea ice volume loss (SIV loss, km$^3$) and mean September water temperature in MIZ (Tw,$^\circ$C), (c) monthly mean sea ice volume flux through Fram Strait (SIF, km$^3$/month) (d) annual mean water temperature in the West Spitsbergen Current ($T_{WSC}$, $^\circ$C) and monthly mean ocean heat flux ($Q_{Svinoy}$, TW) through Svinøy section (see Fig. 1).

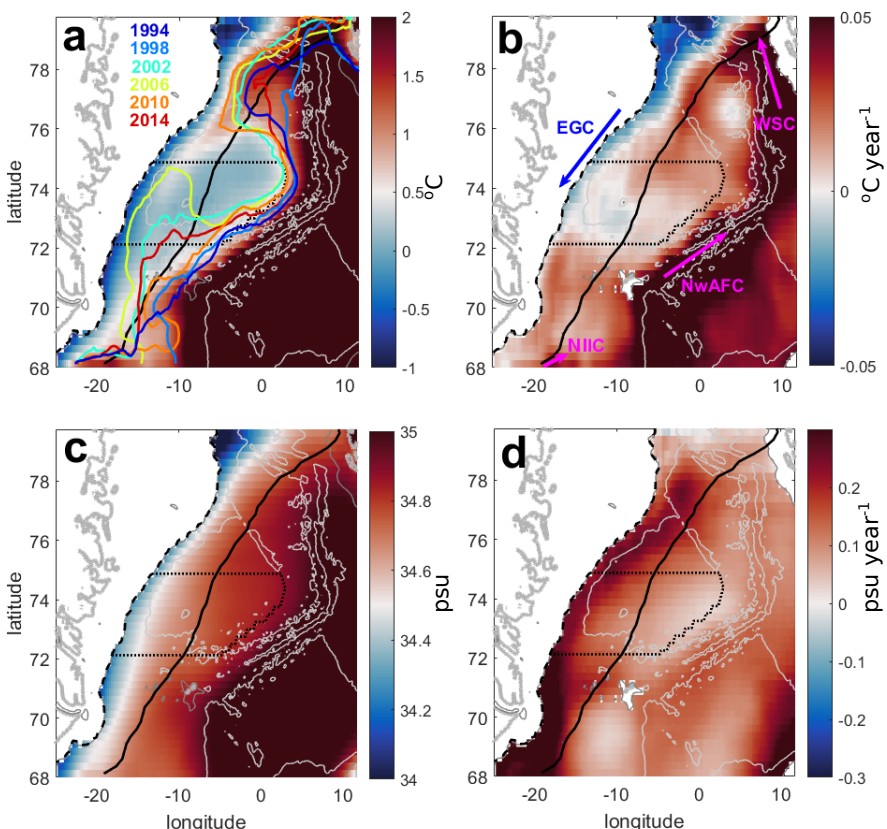

**Figure 5.** Marginal sea ice zone (enclosed in black lines) and themohaline water properties averaged in the upper 50-m layer during cold season (October-April). a - time-mean (1993-2016) temperature (°C) in MIZ and location of 2°C isotherm in November for selected years; b - linear temperature trend (°C year$^{-1}$) in the upper 50 m-layer from 1993 to 2016; c - time-mean (1993-2016) salinity in MIZ; d) linear salinity trend in the upper 50-m layer from 1993 to 2016. In plate (b) EGC is the East Greenland Current, NwAFC – the Norwegian Atlantic Front Current, NIIC – the North Icelandic Irminger Current, WSC – the West Spitsbergen Current. Dotted lines in panels (b) and (d) mark the region, where Odden tongue is observed.

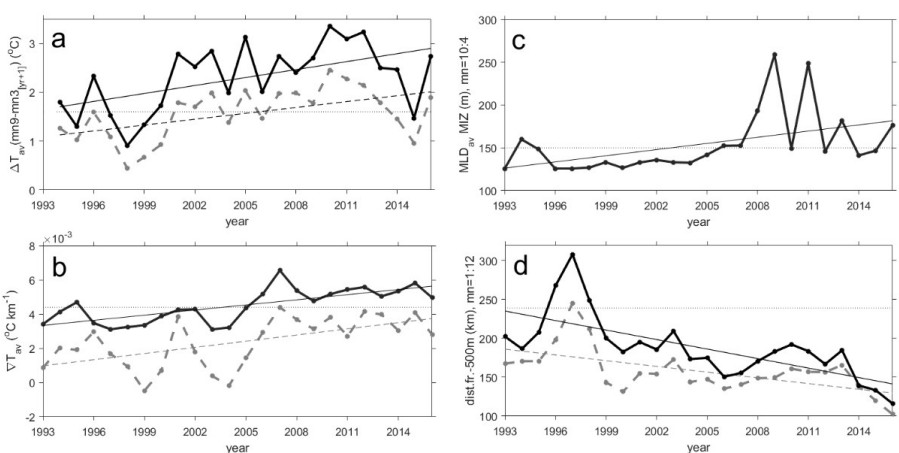

**Figure 6.** Interannual variations of water properties, averaged over the MIZ area. (a) Temperature drop (°C) from maximum in September to minimum in April next year; (b) annual mean temperature gradient across the MIZ (°C km$^{-1}$); (c) the mixed layer depth (m), averaged over the cold season; (d) annual mean distance of the 3°C isotherm from the shelf break (km). In panels (a), (b) and (d) solid black line – data averaged over the upper 50-m layer, dashed gray line – over the upper 200-m layer. In panel (d) 3°C isotherm is shown for the 50-m means and 2°C – for the 200-m means.

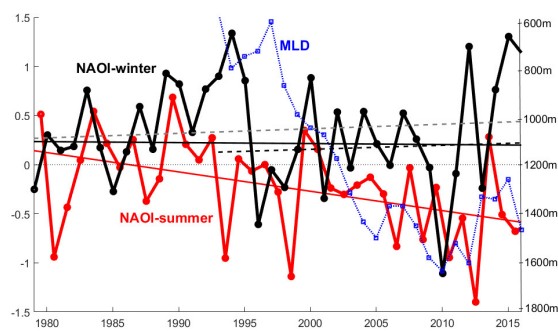

**Figure 7.** Cold season NAO index (black, November-April) and warm season NAO index (red, May-October) with linear trends. Additionally plotted are the trends of cold season NAO index since 1993 (black dashed line, October-April) and for winter season (gray dashed line, January-April). The blue line shows maximum MLD in the Greenland Sea derived from ARMOR data-set (see Bashmachnikov et al. (2019) for details).

**Table 1.** The list of data sources used for estimates of sea ice volume flux through Fram Strait: sea ice concentrations (SIC), sea ice thicknesses (SIT), sea ice drift velocities (SID) and the time periods of the estimates.

| Study | SIC | SIT | SID | Period |
|---|---|---|---|---|
| Kwok et al. (2004) | ULS moorings | ULS moorings | Kwok and Rothrock (1999) | 1991-2002 |
| Spreen et al. (2009) | ASI AMSR-E | ICESat | IFREMER | 2003-2008 |
| Ricker et al. (2018) | OSI SAF SIC + sea ice type product | AWI Cryosat-2 | OSI SAF | 2010-2017 |
| this study | - | PIOMAS | NSIDC Pathfinder v3 | 1979-2017 |

**Table 2.** Statistics of monthly PIOMAS versus satellite-based estimates of the sea ice volume fluxes through Fram Strait: Pearson correlation coefficient (cor. coef), variance relative to PIOMAS (var. rel.), bias, relative percentage difference (RPD), root mean square error (RMSE).

| Study | cor.coef. | mean slope | var. rel.,% | bias | RPD,% | RMSE,km$^3$ |
|---|---|---|---|---|---|---|
| Kwok et al. (2004) | 0.70 | 0.71 | 98 | 47 | 66 | 75 |
| Spreen et al. (2009) | 0.60 | 0.61 | 97 | 33 | 45 | 56 |
| Ricker et al. (2018) | 0.84 | 0.66 | 162 | 107 | 88 | 108 |

var. rel.,% $= (100\% * var_{obs})/var_{PIOMAS}$

bias $= obs. - PIOMAS$

**Table 3.** Trends in monthly mean characteristics in the Greenland Sea calculated over annual (September-August), winter (October-April) and summer (May-September) periods: sea ice volume (SIV, $km^3$ $year^{-1}$), sea ice volume loss (SIV loss, $km^3$ $year^{-1}$), sea ice flux through Fram Strait (SIF Fram, $km^3$ $year^{-1}$), water temperature in MIZ (Tw, $^\circ$C $year^{-1}$) and in the West Spitsbergen Current (TWSC, $^\circ$C $year^{-1}$), heat flux across the Svinøy section ($Q_{Svinoy}$, TW $year^{-1}$). $r^2$ - coefficient of determination, STD - standard deviation (m), p-value - probability value.

| parameter | season | trend | $r^2$ | STD | p-value |
|---|---|---|---|---|---|
| SIV, $km^3$ $year^{-1}$ | annual | -7.24 (-1.15%) | 0.42 | 1.48 | <0.01 |
| | winter | -8.48 (-1.35%) | 0.44 | 1.66 | <0.01 |
| | summer | -5.82 (-0.93%) | 0.26 | 1.72 | <0.01 |
| SIV loss, $km^3$ $year^{-1}$ | annual | 0.94 (0.88%) | 0.09 | 0.52 | 0.08 |
| | winter | 1.18 (1.10%) | 0.06 | 0.83 | 0.17 |
| | summer | 0.84 (0.79%) | 0.10 | 0.45 | 0.07 |
| SIF Fram, $km^3$ $month^{-1}$ $year^{-1}$ | annual | 0.96 (0.88%) | 0.09 | 0.53 | 0.08 |
| | winter | 1.36 (1.25%) | 0.08 | 0.82 | 0.10 |
| | summer | 0.56 (0.52%) | 0.09 | 0.32 | 0.08 |
| Tw, $^\circ$C $year^{-1}$ | annual | 0.015 (1.50%) | 0.23 | 0.007 | 0.04 |
| | winter | 0.008 (0.01%) | 0.05 | 0.007 | 0.29 |
| | summer | 0.026 (3.00%) | 0.29 | 0.008 | <0.01 |
| $Q_{Svinoy}$, TW $year^{-1}$ | annual | 1.84 (1.39%) | 0.48 | 0.41 | <0.01 |
| | winter | 1.83 (1.38%) | 0.35 | 0.54 | <0.01 |
| | summer | 1.82 (1.37%) | 0.36 | 0.53 | <0.01 |
| $T_{WSC}$, $^\circ$C $year^{-1}$ | annual | 0.036 (0.60%) | 0.30 | 0.30 | <0.01 |