# Peer review of "Sea ice volume variability and water temperature in the Greenland Sea"

_The Cryosphere, 2019_

## Referee Comment (RC1) · Anonymous Referee #1 · 31 Jul 2019

Review of

Sea ice volume variability and water temperature in the Greenland Sea

by Selyuzhenok, V., et al.

Summary: In this paper the authors investigate the temporal development of the sea-ice volume in the Greenland Sea and attribute their observations about the sea-ice volume changes to changes in the oceanographic conditions. On the sea-ice volume side the paper employs PIOMAS model computations. These computations involve local ice growth as well as im- and export out of the Greenland Sea. PIOMAS data are assessed with independent data before deriving conclusions about the sea-ice volume change. On the oceanographic side the authors use the ARMOR data set, a

compilation of quality controlled hydrographic observations of salinity and temperature. The authors complement this data set with additional estimates of these parameters from a regression analysis. By means of a regional sea-ice mass balance analysis and an analysis of the increase of the ocean heat content, the shoaling of warmer water masses and increases in mixed layer depth the authors conclude that the increasing amount of sea-ice volume imported into the Greenland Sea through Fram Strait is melted completely by the ocean.

I rate this as an important contribution to the scientific literature and suggest that the paper is going to be published in "The Cryosphere". The topic would possibly also find many readers in Journal of Geophysical Research - Oceans.

Before publication and finalization of the paper I ask the authors to comply to a number of concerns I point out below - first some general comments, then some specific comments, followed by some editoral remarks and suggestions. Note that the major concerns are listed in the general comments. The specific comments mostly pertain to (possible) misunderstandings from my side or issues which I did not understand from the text. I ask the authors for a careful revision of the paper - also with respect to English grammar, usage of symbols and units and the figures. I found the paper difficult to read once in a while.

General comments: GC1: Four questions/comments I have after reading the introduction, which I suggest you to comment on in the paper: 1) Do fresh water export (through Fram Strait) variations influence sea-ice production on and off the Greenland shelf? 2) How much of the sea ice drifting south along the Greenland coast on the shelf is advected into the open Greenland Sea, i.e. off the shelf, and how is this related to the wind? 3) What are the water masses encountered in the Greenland Sea on and off the shelf? 4) PIOMAS is your work horse. Even though PIOMAS seems to have an excellent performance it should be kept in mind that this is a model with some inherent difficulties to describe the actual physical properties. Therefore it could add excellence to your paper by stating that you are aware of potential biases (as you

will show below) in the model parameters, and by making the point that you are less interested in absolute values but rather in long-term variations and trends (and there is no reason why the model should have any drift over time, i.e. one could expect a bias in the sea-ice thickness of 1 m in 1979 to be of the same magnitude in 2009 under the same environmental conditions).

GC2: The CS-2 data set is taken as if it is the truth. There are two concerns which need to be mentioned in the data-set description and again mentioned in the context of your inter-comparison between PIOMAS and CS-2 sea-ice thickness. 1) The CS-2 sea-ice thickness retrieval requires snow depth information which is taken from a climatology. Hence any inter-annual variation in sea-ice thickness might not be due to an actual variation in sea-ice thickness but due to a variation in the match between the snow-depth climatology and the actual snow depth. 2) By the same token: The snow-depth climatology used is not valid outside the Arctic Ocean. Snow depths outside the Arctic Ocean are based on an extrapolation which, e.g. in the Hudson Bay provide negative snow depths.

GC3: This concern goes to Section 3.2. I have a few comments / questions here which I ask the authors to explain better and/or comment in their paper. 1) I would strongly recommend to assign an ice mass balance GAIN to a POSITIVE value of "MB" and an ice mass balance LOSS to a NEGATIVE value of "MB" and not the way done currently. It is confusing the way written. 2) Did you take into account how long sea ice stays in your region of interest? Or in other words: How long does a group of ice floes entering the Greenland Sea at Fram Strait need to travel the distance to Denmark Strait? Could this impact your estimates? 3) How did you compute the regional sea-ice volume? What is the region over which you compute the sea-ice volume? 4) Please carry out a unit check. Which physical units do V, QF and MB have? Do these fit together? 5) You combine the difference in the regional sea-ice volume of two consecutive months, e.g. January and February, with the sea-ice volume flux difference at the northern (QF) and southern (QD) end of your region of interest for February. I assume that the time

for which the sea-ice volume data are "valid" are Jan 15 and Feb 15, i.e. the middle of the respective month, integrating over Jan 1 to 31 and Feb. 1 to 28. For which time period is the sea-ice volume flux estimate valid? To me February implies that it is also derived for February and is hence valid for Feb 1 to Feb 28. Please describe what you combined in more detail because to me the balance seems not closed the way it is computed / written. It seems to me that you are combining different time periods.

GC4: A lot of the interpretation of the data is / needs to be based on the ARMOR data set period which begins in 1993 and ends in 2016. On the other hand, the main results obtained with PIOMAS with respect to sea-ice volume and sea-ice volume fluxes and sea-ice mass balances are for the period 1978/79 through 2017, hence a substantially longer period. The paper would benefit from adding a careful consideration and discussion of the considerably different trends in the sea-ice volume related variables for the shorter ARMOR period in comparison to the longer period. Conclusions might change.

GC5: The period considered starts in winter 1978/79 and hence at a time when Is-Odden events occurred quite regularly. The paper lacks a discussion of the results with respect to the Is-Odden variability and, in particular, about the practical absense of the Is-Odden since about 2004 (?). In addition, the paper lacks a discussion about the validity of the usage of an average MIZ area in a highly dynamic region where, thanks to the Is-Odden, sea-ice edges can be located substantially further off-shelf than suggested by the MIZ area chosen. Particularly in the context of Equation 7 usage of an actually varying MIZ might change the picture. Finally, the period also covers the so-called ice-surge years 1989-1991 when a lot of the really thick and old ice exited the Arctic Ocean through the Fram Strait. A discussion of whether this is visible in the results or not (and why not) would also nicely complement this paper - perhaps even more than the relatively hypthetical considerations about NAO-Index links with water mass properties, circulation changes, and mixed layer depth variations.

#############################

Specific comments:

Page 1 - Line 18: From where is "oceanic buoyancy advected to the sea"? Which sea?

Page 2 - Line 2: "by solid ice transport" –> do you refer to sea-ice transport? Then I suggest to name it like this and then add something like "melting outside the Arctic Ocean"

- Line 10: Did Ricker et al. (2018) also exclude extreme negative NAO events? If not then please re-formulate the sentence accordingly.

- Lines 13-15: Please make sure you write sea-ice volume flux where you refer to volume flux and sea-ice area flux where you refer to area flux. Here it remains unclear what "sea ice flux" is. - Line 16/17: I am not sure the statement about the sea-ice production holds the way written, because "sea-ice production" is not just about sea-ice area but also about sea-ice thickness and/or volume. I did not find any hint about sea-ice volume in Germe et al. (2011). It is a tricky region. Perhaps you could split this statement into two parts: one related to the sea-ice on the shelf which particularly in the northern part (i.e. between Fram Strait and 75 degN) experiences a lot of fractioning and lead openings in which sea-ice forms quickly and to considerable thicknesses while the other one related to the off-shelf new ice formation in the Is-Odden tongue area, which is mostly thin, grease and pancake ice, sea ice. I agree with you that the largest variability is observed in the Is-Odden region but, to my knowledge, we also simply don't know anything about the variability of sea-ice production on the Greenland Sea shelf.

- Lines 32/33: "Shorter time series" <–> Figure 3c in Spreen et al. (2009) does not go along well. I suggest to rewrite this statement.

Page 3 - Section 2.1 general: Please provide information such as grid resolution and type, time step (6-hourly?, daily?), etc. with which you used the PIOMAS data.

- Lines 7-10: Please be more specific with the data sets assimilated into PIOMAS, e.g.

which algorithm the sea-ice concentrations are based upon, what the origin of the sea-surface temperature data set used and what kind of NCEP/NCAR data is used? Is the latter from re-analysis?

- Line 17/18: I suggest to use "inter-comparison" instead of "cross-validation". What you carry out is not a validation - mainly because you don't have the true sea-ice thickness at hand. The same applies to later usage of this term.

- Line 19: While you describe the CS-2 data in Section 2.2 you don't describe the ULS data (which you state here to be used for the "cross-validation" of the sea-ice volume)

- Line 25/26: What kind of a grid is this? "spatial resolution" –> "grid resolution".

- Line 29: I find your variable notation quite confusing and not to the point (here and again later in your paper). Suggestion: SIC –> C, HI –> I, HIE –> I_eff , i.e. with "eff" as a subscript. You could drop the "i" in the subscript and simply write in the text that you carry out this computation for every grid cell.

Page 4 - Lines 12/13: How are the vertical density profiles computed? Are these part of the ARMOR data set or did you compute them on your own? Are the mentiond current velocities relevant for your paper? Are these available with the same grid resolution?

- Lines 18-20: It is not entirely clear to me from how many profiles (?) with which average inter-profile distance (?) data contribute to the time series used. What is meant by "the core"?

- Line 21: Would it do any harm on the data set to also include data from May? That way you would comply with your earlier definition of summer: May through September.

- Lines 24-29: Which sea-ice drift data set is used? Is this quantity provided by PI-OMAS? You have introduced the effective sea-ice thickness already before and can delete the second sentence here, changing "sea-ice thickness" to "effective sea-ice thickness" in the first sentence. Did Sumata et al. (2014/2015) also include PIOMAS and/or the sea-ice drift data set you used in their inter-comparison studies?

Page 5: - Equations 2 to 4 and related text: Following up with my comment to Equation 1 I suggest that you also here change the notation. It seems that you need to use super-scripts to indicate the source of the data, i.e. IˆCS2_eff for the effective sea-ice thickness from CS-2 (see Eq. 1) and IˆPIOMAS_eff for the effective sea-ice thickness from PIOMAS. On which grid is this computation carried out? If l = 25 km = constant distance between grid cells (or grid cell centers?), then it needs to be a grid such as the EASE-grid? Please be more specific here. Furthermore, usage of $D_x$ and $D_y$ suggests that your drift data set indeed only contains drift components relative to the grid (which?) on which the data set is provided and does NOT contain the true u (West-East positive) and v (South-North positive) motion components? May I nevertheless suggest that you change "D" to something like "v" for velocity or, even better, "u" and "v" (of course keeping the sub-scripts x and y)? If you then also replace "l" by "d" for distance then equations 2 and 3 might be more understandable at first glance.

- Lines 9-11: I suggest to term this sea-ice volume flux component QD. I suggest to refer to Figure 1 for illustration of the location of this gate. Is QD defined positive when leaving the Greenland Sea?

- Lines 11-15: It might make sense to put these lines into a new paragraph, starting with "In order ...". I don't understand what you did here. Did you read the figures of the sea-ice volume fluxes from the papers or did you carry out the entire computations again on your own or did you copy the figures? Please be more specific in what you did. Please also stress that in case of Spreen et al. (2009) you only used the ICESat data part.

- Line 17: "formed due to thermodynamically" ?? please re-phrase

- Lines 28-30: Did you use density or potential density? You text is confusing here.

Page 6 - Line 2: "tested point" –> perhaps better: "tested depth"?

- Lines 4-7: Please motivate your choice of defining the MIZ. I am asking because

the inter-annual variation of the MIZ certainly results in actually much larger or much smaller areas to be considered. Particularly for winters before 2004, when the Is Odden was observed more often than after 2003, this definition would mean that the MIZ is defined for a much smaller region than actually occupied.

- Lines 11-15: Please write where this transect Q is located. If Q is located along a latitude, isn't $d_x$ constant? I understood that the ARMOR data set as 1/4 degree resolution, so that neighboring data points are separated by the distance corresponding to 1/4 of a degree at the latitude of Q. If not - how is $d_x$ computed? In Equation 6, I suggest to use a small "v" for the current speed and instead of the subscript "w" use "water" to avoid confusion with the vertical velocity component which is usually termed "w". Are density and specific heat of water constants or do these vary with temperature? Is 1030 kg/mˆ3 a valid value for the Greenland Sea? $d_z$ denotes the "processed depth level" but the index "i" in $Q_i$ and $T_i$ denotes the i-th grid cell? Perhaps it makes sense to re-write Equation 6 with two integral signs, one over dx and one of dz? Please write the motivation to use $T_{ref}$ = -1.8degC (because you want to estimate the role of this heat flux in melting sea ice).

- Line 26/27: This tail grows over time and is most pronounced in April. Are you able to assign a particular area in your region of interest to this tail?

Figure 1: - Why do you show data for the period September-April? You defined winter further above as October-April. This is confusing. - Which sea-ice concentration data set is used in Fig. 1 a? NSIDC offers a multitude of different data sets. - Did you interpolate the PIOMAS data onto the CS-2 grid or vice versa? - The color bar used as legend in Figure 1 b is empty. Please correct. - If possible I would enlarge the figure. - Caption: "isobash" –> "isobath"; state the time period (months, years) for which Fig. 1 b) is computed.

Figure 2: - Again the question one which grid this comparison is carried out - I don't understand how the data points in Figure 2 h) are computed. It says area-mean ... but

I find several points per month, as if several sub-areas were used. - While the color coding of Figure 2 a) to g) and its usage in Figure 2 h) is nice, the scatterplots in a) to g) would benefit from color-coding the probability of a respective SIT data pair to occur. That way one cannot not use the color anymore in Fig 2 h) but there you could use different symbols and only provide ONE region mean value and express the variability of the area-mean monthly SIT by error bars denoting plus/minus one standard deviation for both data sets. - Caption: I note that image i) is not existent. That part of the caption should be deleted. - Please note the unit of the RMSE given in the scatterplots.

Page 7 - Line 6: "start decreasing" –> well, you might not want to exaggerate this finding, it is just for 2016 and 2017.

- Lines 10-12: I guess your statements about the inter-annual and intra-annual variations in sea-ice volume flux hold - particularly in the light that PIOMAS is known to under-estimate thickness for thick sea ice and therefore not unexpectedly show a slight negative bias in the Fram Strait sea-ice volume flux compared to the other data sets. <–> But I am much less confident with the results about the sea-ice volume for the reasons laid out in GC2 and because Fig. 1 b) has very small areas where the difference PIOMAS minus CS-2 SIT is acceptably low. Positive and negative sea-ice thickness differences along your gate in the Fram Strait tend to cancel each other out and therefore the sea-ice volume flux agreement is good (By the way: There the CS-2 SIT data set is potentially much more credible than, e.g. at 78deg N). The large bias at the Denmark Strait possibly is not to relevant because of the small flux value anyways. But the majority of the Greenland Sea shows a substantial bias between PIOMAS and CS-2 and you need to discuss whether this bias (if it is real) is relevant for your findings or not.

Figure 3: - I believe it is sufficient to show the mean monthly values for the three satellite / ULS data sets. One can see whether they are within the error margin of PIOMAS or not. If you want to provide the standard deviations of the three other data sets then you could do this in a Table, don't you think so. In any case Fig. 3 b) would become more

[Figure]

readable without the dotted lines. - I am a bit confused about the different time scales. In Fig. 3 a) you show PIOMAS for 1991 to 2017 but in Fig. 3 b) your computations are based in one year less (2016)? - I have to admit that I don't like that the grey shaded area denotes the standard deviation over the entire period. Did you by chance play around with the data to see how this shaded area looks like when using exactly the same periods as used for the observations? Only in that case a check whether the observations fall into the shaded area or not makes sense. - The legend under Fig. 3 b) says Ricker et al. 2017 instead of 2018. - Fig. 3 c), y-axis: check unit. - Please enlarge the entire figure.

Lines 14-28 and Figure 4 and Table 3: - Please describe whether the seasonal (i.e. summer and winter) values shown in Figure 4 are total values, i.e. May+June+July+August+September, or mean monthly values for these months). I assume the latter. Possibly I overlooked something of this description in the text? - Please explain why in Figure 4 (see caption) you re-define winter to Dec.-Apr and summer to May-Nov. while earlier in the paper you use Oct.-Apr. for winter and May-Sep. for summer; also for Table 3 you seem to have used the latter two periods. - Why do you refer the winter and summer trends to the annual mean sea-ice volume (lines 15-17)? Wouldn't it have been more straightforward to relate the seasonal trends to the respective seasonal mean values? - I suggest to enlarge Figure 4 as a whole. That way you would be able to replace the "a", "w" and "s" in the annotation of the different colored lines by "annual", "winter" and "summer" and make the Figure as a whole more readable - because in this case you can also resolve the ambiguity in the annotation with "a" which so far means "AOI" in image a) but "annual" in image d). - You forgot to describe what is shown in Figure 4 c). I assume these are the mean seasonal monthly mean sea-ice volume fluxes through Fram Strait? In general the caption of Figure 4 needs a revision since it should contain information about what "a", "w", and "s" mean. The unit of TWSC should possibly be just °C. For the ocean heat flux you might want to add "Q_Svinoy" in the caption as well as at the right y-axis annotation and use the currently present "TW" as the unit. - I note that you display annual values in Table 3

but refer to decadal values in the text. It might be good to harmonize this and change the values in Table 3 to decadal values as well. - "unexpectedly goes along with an increase in the monthly ice volume flux through" –> "coincides with an increased sea-ice volume import through" - Since in Line 20 you state a significance level it might be good to do this for the trends in the total Greenland Sea sea-ice volume as well; these are even more significant it seems. - Table 3, caption: "summer (March-September)" –> "summer (May-September)" - Line 21: I don't understand where the 112.8 kmˆ3 / decade come from. If I add up 12 times the monthly sea-ice import per decade (of 9.6 kmˆ3) then I end up with 115.2 kmˆ3 / decade - in case this is what you wanted to do. - Line 22: "Fig 2" –> I guess this needs to be Fig. 3 a) - Lines 23-28: Please spend a bit more time and effort to describe what we see in Figure b) and relate it to Equation 5. I also suggest to exchange images b) and c). You could write that for quite a number of years the sea-ice volume loss is larger in summer than winter - which is not surprizing as summer is the main melting season. Fig. 4 c) kind of shows the left difference of Equation 5. Would it make sense to show an additional image in which you show the the right difference, i.e. the mean difference of the sea-ice volume of consecutive months? Such an additional image could aid in the interpretation of Fig. 4 b).

Table 3: - What is rˆ2? - What is the unit of the STD and for which period / over which data is it computed?

Page 8 - Line 2: "downwards" –> "with depth"?

- You use the upper 50-m layer and the upper 200-m layer when showing and explaining your results. Why two different thick water layers? Please motivate / explain in the text or change.

- Line 8: "over the 200-m layer" –> "over the upper 200-m layer"

- What is the reason to show the November 2°C isotherms? Why not December or February?

- Line 10 and Figure 5 b): Please be consistent with what you show. In the text you speak about "linear temperature trends". In the caption of Figure 5 you write "linear change in temperature" and the title of Figure 5 b) says dT2016-1993 which could be interpreted as a plain difference between 2016 and 1993. Please correct and/or modify accordingly. If Fig. 5b) indeed shows a trend then you need to change the unit.

- Figure 5 in general: I suggest to remove all Figure titles and put the respective information in the annotation of the legend and the caption.

- Line 11: You refer to the MIZ only and therefore "western" needs to be "eastern".

- How realistic is the cooling in the northern part of the MIZ?

- Line 12&14 and Figure 5 d): Same comment as for Line 10 and Figure 5 b)

- Line 12: "Fig. 4d" –> "Fig. 5d"

- Lines 13 and 16: Add "layer" behind "200-m"

- Line 16: "and over the MIZ area"? Would "in the MIZ area" be better? As far as I understood you, you concentrate on the MIZ, don't you?

- Lines 17/18: "From ..." –> this is one way to interpret this figure. Another way would be to interpret the early years' small temperature decreases from Sep. to Mar. as a negative anomaly; it is unfortunate that you don't have data before 1993. You could refer in this context to Figure 5b and Figure 4d, right?

- Lines 19-22: "The heat ..." –> I am not sure I understand what you want to state here. First of all, isn't it normal that the heat stored during summer & fall is released during winter? Secondly, an increasing (as you postulated) cooling from September to March (Fig. 6 a) can indeed by caused by an intensification of the vertical mixing and hence a more efficient ocean-atmosphere heat exchange. Also, it could be caused by a higher autumn water temperature but also by a lower March water temperature. What I am missing here is an attempt to relate the observed differences to the extent of the Is-

[Figure]

Odden. Its formation and presence has a profound impact on the upper layer water mass properties. I would delete the Line 19/20 sentence part "decreasing the ...". This is a hypothesis.

- Line 24: add "(not shown)" behind "in winter". For Figure 6 b) one could also postulate a step change between 1993-2006 and 2007-2015.

- Lines 25/26: For the discussion of Fig. 6 b) you refer to Fig. 6 d); I'd see a much better association between Fig. 5 a) and 5 d) in the sense that the dip / peak around 1997/98 could be an anomaly.

- Line 28: add "are" before "observed"

- Lines 30-32: What explains the peaks in winters 2008/09 and 2010/11 in Fig. 6 c)? These are possibly the main reasons for the observed increase in MLD.

Page 9 - Line 1: These September temperature values are not shown somewhere, are they?

Equations 7 and 8: - Please spend a subscript "water" to the density in Equation 7 and replace the subscript "L" in Equation 8 by "ice". - Replace "dq" by "dQ" in Equation 8. - In the text you write 1.8°C for 2016, in Equation 7 you used 2.0°C. Please correct.

- Lines 12/13: I don't agree with the way you estimate the sea-ice volume loss for the 24-year period. That trend you use (possibly from Table 3) is computed over the entire period, starting in winter 1978/79 and not for the period 1993-2015. Fig. 4 b) clearly shows that if one would compute a trend for the 1992/93 through 2014/15 winter time period it might be negative. Also, you use 12 months while in Equation 7 you insert the winter MLD change. It seems hence doubtful to use the entire year. It might therefore make sense to revise this estimate.

- Line 13: "of ice needed to fuse" ? –> delete?

- Lines 13-15: Would it make sense to also mention that a large fraction of your MIZ

area is potentially not covered by sea ice anyways? Would it also make sense to mention that new ice formation in the Is-Odden area but also otherwise in your MIZ area counter-acts this heat release? Would it make sense to also mention that the heat not necessarily needs to reach the surface but stays aways from the sea ice at some depth? My feeling is that one should not overlook the assumptions made.

- Line 22: "multiyear" –> Do you refer to multiyear ice here? In that case write it accordingly.

- Line 23: Whom do you mean with "The authors"?

- Lines 23/24: This is a global statement, perhaps too global. PIOMAS under-estimates thicker ice thickness and over-estimates thinner ice thickness. Please discuss this in more detail because, yes, the thick ice in the Greenland Sea has become thinner but at the same time the Is-Odden feature with a lot of thin ice has vanished.

- Lines 25/26: "compared to know from literature fluxes" –> "compared to flux values known from literature"

- Lines 29/30: Fig. 2 i) does not exist. I guess this needs to be Fig. 1 b). "is lower compared to" –> I'd say this applies to 2/3 of the meridional gate. Don't forget the zonal part of the gate where the differences are opposite. Don't forget also GC2 in this context. "the NSIDC sea ice drift" –> needs to be introduced in the data section. Version 2 is quite old, by the way. State of the art is Version 4.

Page 10 - Lines 2-7: As an outlook you could add that it might make sense to separately, in PIOMAS, look at the changes in sea-ice formation in the true MIZ, i.e. the actually ice covered area and not just the average MIZ as defined by you, and in the consolidated ice covered part on the shelf. There are many leads created in the wider Fram Strait area in which thin ice grows quickly and which is advected southward on the shelf, continuing to grow.

- Line 7: "intensification of in sea ice melt" ?

- Line 24: "through to be mostly driven" ?

- Lines 25-27: Please rewrite this sentence. It is confusing. Which "inconsistency"? Which "peculiarities"? "delution"? Does Polar Water have an influence on your area?

Page 11 - Line 5: "NAO phase increases of the intensity" ?

- Line 17: Fig. 4 f) needs to be Fig. 4 d).

- Line 33: "Governed by ..." This sentence is difficult to read; please re-formulate.

Figure 7: Here different winter and summer periods than in the rest of the paper are used. Why? Please motivate, change, or delete.

Page 12 - Line 19: Why "Therefore"?

Editoral stuff: General: - I found "northern winds" and "northerly winds". Please use one term. - Check for "Oddin" - I found "accessed" in case where "assessed" should be used, e.g. Page 3, Line 17 or Page 5, Line 10. - It might enhance the flow of your paper if you always use the same term for the same parameter. Example: use "effective sea-ice thickness" all the time and not "effective ice thickness" - You have an issue with using "though" instead of "through". Please check.

Page 1 - Line 16: "The 2/3 of" –> "Two third of ..."

Page 2 - Lines 11-13: there are some issues with blanks and parentheses. Please check.

Page 4 - Line 19: WSC needs to be explained. "quire" –> "quite"?

Page 5 - Line 9: "months" –> "month" - Line 13: "while in other" –> "while in the other"

Page 6 - Line 3: Check references mentioned here - Line 25: Put "Schweiger et al., 2011" in ()

Page 7 - Line 23: "significantly" –> "significant" "sea ice balance of the sea" –> "sea-ice mass balance of the Greenland Sea"

Page 8 - Line 10: "Yan" –> "Jan"

Page 10 - Line 17: "brining" –> "bringing"

---

## Referee Comment (RC2) · Anonymous Referee #2 · 20 Aug 2019

Summary:

This study investigates the temporal development of sea ice volume in the Greenland Sea between 1979 and 2016 based on the PIOMAS model. Changes in sea ice volume, as well as import and export of sea ice are used to compute the evolution of total sea ice volume loss in the region. The authors find that the sea ice volume has decreased through the period even though the import of sea ice has increased. They explain this development by increased melting in the region as a result of higher ocean heat content in the marginal ice zone (MIZ) in the Greenland Sea. Hydrographic data from the ARMOR data set is used to show that the temperature in the MIZ has increased as a result of warmer Atlantic Water flowing into the region and due to increased mixed-layer depths that entrain more heat from the water column below.

[Figure]

I found this study interesting to read. The description of long-term variations in sea ice volume in the Greenland Sea and its link to changes in mixed-layer properties in the MIZ make this study an important contribution to the literature on Arctic and Subarctic sea ice variability. Hence, I recommend this paper to be published in "The Cryosphere".

However, I do think there are a number of issues that the authors have to address before the paper is ready for publication. My main concerns are listed under general comments. Then follows several specific comments and technical corrections, many of which are related to unclear text and English grammar.

General comments:

The development of the sea ice volume in the Greenland Sea is investigated, but how is the Greenland Sea defined? The red box in Fig. 1 marks the entire Nordic Seas, which consists of the Norwegian Sea in the east and the Greenland + Iceland Seas as well as the east Greenland shelf in the west. I would rather say that you study the sea ice volume in the Nordic Seas or western Nordic Seas with a focus on the marginal ice zone. The inconsistent use of "the Greenland Sea", "the Nordic Seas", and "the Greenland-Norwegian region" etc. makes the paper a bit confusing to read and it is not clear to me over which region you actually computed the sea ice volume.

The authors start by introducing the Greenland Sea as an important area for deep convection and that the intensity of convection is controlled by buoyancy fluxes, in particular the input of freshwater (and sea ice). However, little is said about the observed changes in local sea ice formation, the retreat of the ice edge, winter-time heat loss, and their combined effect on convection in the Greenland Sea which has varied substantially over the past four decades. See e.g. Visbeck et al. (1995); Marshall and Schott (1999); Moore et al. (2015); Brakstad et al. (2019).

Some statements about the amount of available data in the MIZ (in the ARMOR data set) are required. How does the generally sparse data coverage along the east Greenland shelf affect your results? It would also be good to compare your mixed-layer

properties with observations (i.e. Nilsson et al., 2008; Pawlowicz, 1995; Brakstad et al., 2019). All of these papers show ocean surface temperatures well below 0°C during winter (in the MIZ and in the center of the Greenland Sea). This contradicts what you describe on Page 8 – Line 6-7, that the temperature is always above 0°C leading to sea ice melt. Furthermore, you have used the mean 15% sea ice concentration contour from 1979 to 2016 to define the MIZ. The position of the ice edge has varied substantially during this period (i.e. Moore et al, 2015). How does that affect your results?

It is interesting that the warming of the Greenland Sea and the MIZ can account for the sea ice volume loss in the area plus the increased sea ice export through Fram Strait. However, as noted also in the specific comments, information about the role of the atmosphere is missing. This is crucial in order to obtain a more complete picture of the drivers for the observed development of the sea ice volume. As it stands, you assume that the atmosphere plays a minor role (Page 9 – Line 15 & Page 10 – Line 9). It is possible to quantify the fraction of heat released to the atmosphere, and the role of increased atmospheric temperature, using an atmospheric reanalysis product. I think that considering the atmosphere as well would make your conclusions more solid.

I find the link between long-term variations in sea ice volume and the NAO a bit speculative. On page 11 – line 2 you write that several studies have shown that during positive NAO phase, the intensity of ocean heat flux to the Nordic Seas increases by 50%. However, neither of the studies referred to (i.e. Skagseth et al., 2004 and Raj et al., 2018) examines the oceanic heat flux/ heat transport into the Nordic Seas (rather velocity and volume transport). When Raj et al. (2018) discuss the increase of 50% they are talking about an increase in volume transport. What about variations in temperature of the inflowing Atlantic Water? Based on the studies you refer to, I find the link between NAO and temperature/heat content in the MIZ exaggerated. Either focus less on the NAO link, or refer to literature that show the link more clearly, or investigate the link more thoroughly in this paper.
Specific comments:

Page 1 - Line 16: What do you mean by "this region"? The Greenland Sea, the Nordic Seas, or the North Atlantic? I do not think any of these papers state that 2/3 of the deep AMOC originates from the Greenland Sea.

Page 2 - Line 1: Approximately 50% of the freshwater anomaly at the surface or of the entire water column? Also, what do you mean by "the Norwegian-Greenland region". The Nordic Seas? Changes in salinity of the northward flowing Atlantic Water are also important (ie. Lauvset et al., 2018; Mork et al., 2019).

Page 2 – Line 6: Another very relevant reference for sea ice flux through Fram Strait, and for comparison with your results, is Smedsrud et al. (2017).

Page 2 – Line 14: Please clarify what you mean by "even stronger linked to the Arctic Dipole pattern". In addition, you should briefly introduce the Arctic Dipole pattern, as it may not be clear to all readers what this is.

Page 2- Line 17: The Odden sea ice tongue has not been formed in the Greenland Sea since the early 2000s (ie. Moore et al., 2015). Since then, sea ice has been close to absent in the center of the Greenland Sea.

Page 3 – Line 4: The detected variations of what?

Page 3 – Line 25: How is monthly sea ice thickness from the Cryosat-2 satellite data-set obtained?

Page 4 – Line 9: What do you mean by different weights? Please elaborate.

Page 4 – Line 11: Include reference to the method used in the World Ocean Atlas data-set.

Page 4 – Line 20-21: Interannual variations of what? In addition, replace " - the months the most densely covered with data" with "which are the months with densest data coverage".

Page 5 – Line 10-11: Denmark Strait is between Greenland and Iceland, not all the way to 36E! Please use a different term for your meridional section (a section along the Greenland Scotland Ridge?), or separate it into several sections (ie. one west and one east of Iceland).

Page 5 –Line 17: What do you mean by "due to thermodynamically within the Greenland Sea"? Please clarify.

Page 5 – Line 29: How were the density profiles filtered?

Page 6 – Line 3: How were you able to compare your MLDs with Kara et al. (2003)? None of their figures show MLDs in the Nordic Seas. de Boyer Montégut et al. (2004) are also looking at global mixed layers. I think it would be better to compare with observed MLDs from the Greenland and Iceland seas (Brakstad et al., 2019 and Våge et al., 2015, respectively).

Page 7 – Line 15-16: How does the negative trend in sea ice volume compare to those found in Moore et al. (2015) and Onarheim et al. (2018)?

Page 7 – Line 33: Unclear. Please expand. Atlantic-origin water in the EGC is capped by fresh/cold Polar Water and sea ice during winter, which will inhibit ventilation of the Atlantic Water. Våge et al. (2018) show that due to the retreat of the ice edge the last decades, Atlantic Water has been and is more likely to be ventilated in the EGC. However, we do not know if this takes place "regularly".

Page 8 – Line 1-2: The temperature (and salinity) of the Atlantic Water in the EGC is not increasing downstream. Please clarify what you mean by "increasing southeastwards".

Page 8 – Line 4: "West Islandic Current" is not typically used. Rather use "North Icelandic Irminger Current". A better ref. here would be Jónsson and Valdimarsson (2005) or Hansen et al. (2008).

Page 8 – Line 6-7: As stated in the general comments, you need to compare your data with observations and discuss the temperature uncertainty due to limited data in the

MIZ. Temperatures of 0.1-0.2°C in winter seems unrealistically high.

Page 8 – Line 14: Perhaps you should show the mixed-layer depth for comparison with previous work (ie. Brakstad et al., 2019 and Våge et al., 2015)

Page 8 – Line 17: Clarify what you mean by "overall year mean increase of temperature".

Page 8 – Line 19-22: These lines are confusing and hard to read. What do you mean by "decreasing the interannual trends to insignificant"? Please be more specific.

Page 8 – Line 28-29: Bondevik (2011) is gray literature (no peer review). I would encourage you to refer to peer reviewed literature. In addition, add "are" before "observed".

Page 8 – Line 28-30: Explain how this increases ice melt.

Page 8 – Line 30-32: As stated in the general comments: How does your definition of the MIZ and the data coverage in the MIZ affect the results?

Page 8 – Line 34-35: This corroborates the results of Lauvset et al. (2018) who examined the relationship between hydrography (and MLD) in the Greenland Sea and the temperature/salinity of the northward flowing Atlantic Water.

Page 9 – Line 6-7: The 20% depend on how you define the Greenland Sea.

Page 9 – Line 7: "additional heat release": In addition to what?

Page 9 – Line 13-14: It would be interesting to quantify the fraction of heat released to the atmosphere. This should be possible using atmospheric reanalyses.

Page 9 – Line 15-16: What about increasing atmospheric temperature?

Page 9 – Line 28: Clarify what you mean by "the discussed above general PIOMAS tendency"

Page 9 – Line 29: Figure 2i does not exist.

Page 10 – Line 6: This sentence is not in agreement with Page 8 – Line 6-7 where you state that no sea ice formation occur and that the surface temperature is always >0. Here you write that sea ice is formed locally and that the atm. play a role.

Page 10 – Line 11-12: "almost twice of" what? Please clarify.

Page 10 – Line 25-27: These two sentences are very confusing. Which inconsistency? What local peculiarities? Do you need these sentences at all? If so, please re-phrase and be more specific.

Page 10 – Line 30: Where did you obtain data (heat fluxes) from the Svinøy section? Please include reference.

Page 11 – Line 2: Raj et al. (2018) show a 50% increase in volume transport not oceanic heat flux. (See general comment).

Page 11 – Line 12: You have not really discussed any eastward advection of Polar Water to the southwestern Norwegian Sea. How does this relate to your results? Please elaborate.

Page 11 – Line 23-24: This sentence contradicts line 19, where you state that the summer NAO is not important?

Page 11 – Line 25: What do you mean by "main currents in the Greenland Sea"? Be more specific.

Page 12 – Line 5-7: Maybe better to refer to Brakstad et al. (2019), Lauvset et al. (2018), and Latarius and Quadfasel (2016) that all look at interannual changes in MLD in the Greenland Sea during your period. Lauvset et al. (2018) and Brakstad et al. (2019) both discuss the role of increased salinity on the mixed-layer depth.

Page 12 – Line 9: Smeed et al. (2014) show a weakened AMOC.

Page 12 – Line 20: "govern" is too strong. Line 23-24: "Atlantic Water advection into the MIZ largely contributes to the SIV loss" is more appropriate.

Page 12 – Line 28: In the last paragraph: The link to NAO is speculative, and you have not shown this link in this paper.

Technical corrections:

Page1 - Line 16: Replace "The 2/3" with "Two thirds"

Page 1 - Line 18: What do you mean by "to the sea"? Into the Greenland Sea?

Page 2 – Line 1: Replace "through the Fram Strait" with "through Fram Strait". (Also the case for Page 2 - Line 6, 9 and 10 etc.)

Page 2 – Line 9: Should be "drive" not "drives"

Page 2- Line 11: The entire reference here should be within parenthesis. "(Kwok et al., 2004)" not "Kwok et al. (2004)". Also the case for "Schweiger et al. (2011)" on Page 6 - line 25 in example. Please go through all references and make sure they are consistent.

Page 2 – Line 34: Replace "Oddin" with "Odden".

Page 3 – Line 15: Singular vs plurals: Use either "the spatial pattern of PIOMAS ice thickness agrees" or "the spatial patterns of PIOMAS ice thickness agree".

Page 3 – Line 15: Remove comma after "those".

Page 3 – Line 25: Should be "provides" not "provide"

Page 3- Line 26: Insert "the" before "CS2 data-set".

Page 4- Line 3: Insert "the" before "ARMOR data-set".

Page 4 – Line 7: Insert "depth" before "levels".

Page 4- Line 9: Replace "all observed in situ" with "all in situ observations".

Page 4 – Line 18: Remove comma before "used" and after "paper".

Page 4 – Line 19: Replace "quire" with "quite"

Page 4 – Line 21: Use "a" instead of "the" in "kriging with the 30-km window".

Page 4 – Line 25: Remove comma after "Note".

Page 5 – Line 9: Remove "s" in "months".

Page 5 – Line 10: Denmark Strait should be with capital S.

Page 5 – Line 11: Replace "access" with "assess".

Page 5 – Line 13: Should be "were adopted" not "was adopted".

Page 5 – Line 13: Add "the" before "other".

Page 5 – Line 14: Replace "also is" with "is also".

Page 5 – Line 15: Should be "data-sets" not "data-set".

Page 5 – Line 27: Add "the" before "ARMOR data-set".

Page 6 – Line 3: Remove "de Boyer". It is written twice.

Page 6 – Line 19: Should be "underestimates" instead of "underestimate".

Page 6 – Line 20: Remove "the" before CS2. Also the case on line 21.

Page 6- Line 20: Remove "s" in "values".

Page 6 – Line 21: Remove "the" before "Spitsbergen". Also the case on line 23.

Page 6 – Line 23-24: Either use "PIOMAS tend to overestimate" or "PIOMAS overesti-mates".

Page 6 – Line 24: Remove "thickness".

Page 6 – Line 26: "discrepancies" should be singular => "discrepancy".

Page 6 – Line 30: Remove "the" before "PIOMAS".

[Figure]

Page 6 – Line 31: Replace "all are" with "are all".

Page 6 – Line 31: Add "of" after "correlation".

Page 6 – Line 31: Add "the" before "Ricker et al. (2018) data"

Page 7 – Line 16: Replace "comprises" with a more appropriate term ("was"?).

Page 7 – Line 22: Remove "for" before "about".

Page 7 – Line 23: Should be "significant effect" rather than "significantly effect". Also replace "the sea" with "the Greenland Sea".

Page 8 – Line 1: Add "the" before "climatology".

Page 8 – Line 9: "approaches" is an odd choice of tense when you talk about something that happened from 1990s to 2000s. Replace with "approached" or "propagated towards".

Page 8 – Line 10: It should be "Jan Mayen" not "Yan Mayen".

Page 8 – Line 11: Replace "western" with "eastern". In addition, do you mean "Frontal Current" instead of "Front Current" (same for Page 10 – Line 23)?

Page 8 – Line 12: The "tendencies" are shown in figure 5d. Replace "Fig. 4d" with "Fig. 5d".

Page 8 – Line 23: nearly doubles from 1993 to ?

Page 9 – Line 12: Remove "the" after exceeds. It is written twice.

Page 9 – Line 22: Remove "thickness" after "thick ice".

Page 9 – Line 25: Should be "appears" not "appear". Also, replace "lower compared to know from literature fluxes" with "lower than those estimated by previous studies" or something similar.

Page 9 – Line 27: Remove "the" before "data".

Page 10 - Line 1: Remove "the" before "sea ice volume".

Page 10 – Line 13: Replace "uptake" with "take up".

Page 10 – Line 17: "brining" should be "bringing".

Page 10 – Line 18: "later" should be "layer".

Page 10 – Line 19: Write "Nansen Basin" with capital B.

Page 10 – Line 29: Replace "Further" with "Farther".

Page 10 – Line 30: "Svinoy" should be "Svinøy". Also the case on Page 10 - line 34 and Page 11 – line 16 and 17 etc.

Page 10 – Line 31: Remove comma after "Barents Sea".

Page 10 – Line 34: Remove "in" after "confirmed by".

Page 11 – Line 1: Use capital S in "Nordic Seas". Also the case for line 10 and 20.

Page 11 – Line 5: Remove "of" after "NAO phase increases".

Page 11 – Line 10: "Fram Strat" should be "Fram Strait".

Page 11 – Line 11: Replace "through" by "across" and use capital R in "Faroe-Shetland Ridge".

Page 11 – Line 12: Inconsistent capitalization of "water". Here you write "Polar Water", while in line 6 you use "Atlantic water". Please be consistent throughout the paper.

Page 11 – Line 28: Replace "is" with "was" after "more ice".

Page 11 – Line 29: Add "the" before "Odden ice tongue".

Page 12 – Line 7: Remove "the" after "favours".

Page 12 – Line 22: "MID" should be "MLD".

Page 12 – Line 23: Add "heat" before "necessary".

Page 12 – Line 25: "Froe-Shetland ridge" should be "Faroe-Shetland Ridge". This sentence is also incomplete. Please re-phrase.

Figure 1: The color in the right color bar is missing.

Figure 2: In the figure caption you describe panel (i) – "difference between mean PI-OMAS and CS2 effective ice thickness", but panel "i" is not included in the figure (only panels a-h).

Figure 4: Please write out what the legends "w", "s", and "a" mean.

Figure 5: The color bar in panel "d" has the wrong units. The panel shows change in salinity, but have units of °C.

Figure 6: In the figure caption: Remove parenthesis after "cold season".

Figure 7: Is there missing a second y-axis for the normalized maximum MLD? If not, I do not understand what the values -1 to 1.5 in normalized maximum MLD mean. Please explain.

Table 3: Explain all columns. (i.e. what is correlated in the column r2?)

References:

Brakstad et al (2019): Water Mass Transformation in the Greenland Sea during the Period 1986–2016. J. Phys. Oceanogr., 49, 121–140. https://doi.org/10.1175/JPO-D-17-0273.1

Hansen et al. (2008): The Inflow of Atlantic Water, Heat, and Salt to the Nordic Seas Across the Greenland–Scotland Ridge. In: Dickson R.R., Meincke J., Rhines P. (eds) Arctic–Subarctic Ocean Fluxes. Springer, Dordrecht. https://doi.org/10.1007/978-1-4020-6774-7_2

Jónsson and Valdimarsson (2005): The flow of Atlantic water to the North Icelandic

[Figure]

Shelf and its relation to the drift of cod larvae, ICES Journal of Marine Science, Volume 62, Issue 7, Pages 1350–1359, https://doi.org/10.1016/j.icesjms.2005.05.003

Latarius and Quadfasel (2016): Water mass transformation in the deep basins of the Nordic Seas: Analyses of heat and freshwater budgets. Deep Sea Research Part 1: Oceanographic Research Papers, Volume 114, Pages 23-42. https://doi.org/10.1016/j.dsr.2016.04.012

Lauvset et al. (2018): Continued warming, salinification and oxygenation of the Greenland Sea gyre, Tellus A: Dynamic Meteorology and Oceanography, 70:1, 1-9, DOI: 10.1080/16000870.2018.1476434

Marshall and Schott (1999): Open-ocean convection: Observations, theory, and models. Rev. Geophys., 37, 1–64, https://doi.org/10.1029/98RG02739.

Moore et al. (2015): Decreasing intensity of open-ocean convection in the Greenland and Iceland seas, Nature Climate Change, 5, 877, https://doi.org/https://doi.org/10.1038/nclimate2688

Mork et al. (2019): Recent Warming and Freshening of the Norwegian Sea Observed by Argo Data. J. Climate, 32, 3695–3705,https://doi.org/10.1175/JCLI-D-18-0591.1

Nilsson et al. (2008): Liquid freshwater transport and Polar Surface Water characteristics in the East Greenland Current during the AO-02 Oden expedition. Progress in Oceanography, volume 78, Issue 1, Pages 45-57. https://doi.org/10.1016/j.pocean.2007.06.002

Onarheim et al. (2018): Seasonal and Regional Manifestation of Arctic Sea Ice Loss. J. Climate, 31, 4917–4932,https://doi.org/10.1175/JCLI-D-17-0427.1

Pawlowicz (1995): A note on seasonal cycles of temperature and salinity in the upper waters of the Greenland Sea Gyre from historical data. Journal of Geophysical Research, vol. 100, No. C3, Pages 4715-4726.

[Figure]

Smedsrud et al. (2017): Fram Strait sea ice export variability and September Arctic sea ice extent over the last 80 years, The Cryosphere, 11, 65-79, https://doi.org/10.5194/tc-11-65-2017, 2017.

Smeed et al. 2014 Observed decline of the Atlantic Meridional Overturning Circulation 2004–2012.Ocean Science, 10 (1). 29-38. https://doi.org/10.5194/os-10-29-2014

Visbeck et al. (1995): Preconditioning the Greenland Sea for deep convection: Ice formation and ice drift. J. Geophys. Res., 100, 18 489–18 502, https://doi.org/10.1029/95JC01611.

Våge et al. (2015): Water mass transformation in the Iceland Sea. Deep Sea Research Part I: Oceanographic Research Papers 101, 98-109. https://doi.org/10.1016/j.dsr.2015.04.001

Våge et a. (2018): Ocean convection linked to the recent ice edge retreat along east Greenland. Nature Communications, volume 9, Article number: 1287. DOI: 10.1038/s41467-018-03468-6

---

## Author Comment (AC1) · 7 Oct 2019

Referee #1

Dear referee, thank you very much for your thorough review. It has helped us to substantially improve the manuscript. Please find detailed responses to each of your comments below.

**General comments:**

**General Comment 1**: Four questions/comments I have after reading the introduction, which I suggest you to comment on in the paper:

**1)** Do fresh water export (through Fram Strait) variations influence sea-ice production on and off the Greenland shelf?

**2)** How much of the sea ice drifting south along the Greenland coast on the shelf is advected into the open Greenland Sea, i.e. off the shelf, and how is this related to the wind?

**3)** What are the water masses encountered in the Greenland Sea on and off the shelf?

**4)** PIOMAS is your work horse. Even though PIOMAS seems to have an excellent performance it should be kept in mind that this is a model with some inherent difficulties to describe the actual physical properties. Therefore it could add excellence to your paper by stating that you are aware of potential biases (as you will show below) in the model parameters, and by making the point that you are less interested in absolute values but rather in long-term variations and trends (and there is no reason why the model should have any drift over time, i.e. one could expect a bias in the sea-ice thickness of 1 m in 1979 to be of the same magnitude in 2009 under the same environmental conditions).

**Response:**
**1)** In our study we consider only the area off the Greenland shelf. In this region, according to Fig. 5 (c,d), water salinity increases by 0.1-0.25 in the upper 50-m mixed layer over all the "MIZ" zone used. This is in spite of a larger ice transport through Fram, and we explain this in the paper by a larger concurrent Atlantic water transport into the Greenland Sea. The salinity increase by 0.2, leads to a drop of the water freezing temperature by 0.1. Using Cp=3900 J/(kg $^{\circ}$C), water density=1030 kg/m$^3$ and the MIZ area =2.3*10$^{11}$ m$^2$, we find the additional heat needed to be applied due to the salinity drop to be about 1*10$^{17}$ J. This is 3 order of magnitude less than the additional heat released by the ocean (2*10$^{20}$ J) and the salinity variations can be neglected.

**2)** The advection of sea ice drifting south along the eastern Greenland coast indeed has an influence on interannual sea ice variability of the interior Greenland Sea. We have not found any quantitative estimate in the literature. However, a qualitative linkage between wind pattern and sea ice advection from the eastern Greenland coast is described in Germe et al. (2011). According to Germe et al. (2011), in the region the wind varies with the NAO phase. During the negative NAO phase, a reduction of the northerly wind, permits a more intensive westward Ekman drift of sea ice into the Greenland Sea interior. This information is now included in Introduction (page 3 lines 4 — 6). These may slightly increase the ice volume off shelf due to transport form the shelf. These variations in ice advection are incorporated into PIOMAS through dependence of ice concentration of wind and ocean current drag. Thus, they are included in our mass balance estimates.

**3)** The paragraph below, describing the water masses in the sea, is added to the Introduction (page 2, lines 6— 17):

"The upper 500 m in the western Greenland Sea is formed by mixing the Polar Water (PW) with temperature, close to freezing and salinity from 33 to 34 and the Atlantic Water (AW) with temperature over 3 ◦ C and salinity around 34.9 recirculating in the southern part of the Fram Strait (Moretskij and Popov, 1989; Langehaug and Falck, 2012; Jeansson et al., 2017). The maximum PW content quickly decreasing in the off shelf direction is found in the upper 200 m of the Greenland shelf (Håvik et al., 2017). The AW is found below the PW. Its core is observed in the seawards branch of the EGC, trapped by the continental slope. The centrals parts of the Greenland Sea represents a mixture of the AW and the PW with the Greenland Intermediate Water (with temperature -0.4 – -0.8 ◦ C and salinity ∼34.9). The core of the Greenland Intermediate Water is found at 500-1000 m. The Greenland Sea Deep Water (with temperature -0.8 – -1.2◦C and salinity ∼34.9) is found below 1000 m. The latter two water masses are formed by advection of the intermediate and deep water, coming from the Arctic Eurasian basin through the Fram Strait, mixed with the recirculating Atlantic Water by winter convection (Moretskij and Popov, 1989; Alekseev et al., 1989; Langehaug and Falck, 2012). The convection depth in the Greenland Sea often exceeds 2000 m (Latarius and Quadfasel, 2016; Bashmachnikov et al., 2019)."

**4)** We agree that the missing information on PIOMAS potential biases is important for understanding the results. There are indications of sea ice thinning since 1980s (e.g. Lindsay and Zhang 2005). A reduction of the sea-ice thickness in the Fram Strait was observed in 2003–2012 (Renner et al. 2014). As the PIOMAS bias depends on the ice thickness, the error sign and magnitude will differ in different parts of the region and with time. This issue was addressed in the Discussion (Sec. 5.1). Taking into account you comment, we also changed the data description (page 4, lines 5-8):

"The spatial patterns of PIOMAS ice thickness agrees well with those, derived from in situ and satellite data. The model overestimates the thickness of thin ice and underestimates the thickness of thick ice. Such systematic differences might affects long-term trends in thickness and volume (Schweiger et al., 2011). There is an indication that the PIOMAS shows a conservative sea ice volume trend (1979-2010)."

**General Comment 2**: The CS-2 data set is taken as if it is the truth. There are two concerns which need to be mentioned in the data-set description and again mentioned in the context of your inter-comparison between PIOMAS and CS-2 sea-ice thickness. 1) The CS-2 sea-ice thickness retrieval requires snow depth information which is taken from a climatology. Hence any inter-annual variation in sea-ice thickness might not be due to an actual variation in sea-ice thickness but due to a variation in the match between the snow-depth climatology and the actual snow depth. 2) By the same token: The snow-depth climatology used is not valid outside the Arctic Ocean. Snow depths outside the Arctic Ocean are based on an extrapolation which, e.g. in the Hudson Bay provide negative snow depths.

**Response:** We fully agree that the uncertainties of the CryoSat-2 sea ice thickness retrieval need to be discussed in more detail. Indeed, the modified Warren Climatology, which is used to convert freeboard into sea ice thickness, is not applicable in the Fram Strait. Therefore the snow depth used for the thickness retrieval in Fram Strait is based on an extrapolation of the climatology. On the other hand, ice flows that pass the Fram Strait, comming from the Central Arctic, are advected very fast within one month (up to 500 km/month) . Therefore, we would not expect a significant difference in snow depth between 82°N and 78°N. Nevertheless, the fact that a climatology is used here, means that interannual variations in snow depth are not captured, and can therefore cause interannual biases in the sea ice thickness retrieval. We have added a paragraph in section 2.2 for clarification (page 4 lines 25-31):

"Uncertainties of CS2 ice thickness increase below 78∘N due to sparse orbit coverage (Ricker et al., 2014). The CS2 retrieval is based on sea ice freeboard measurements that are converted into sea ice thickness assuming hydrostatic equilibrium. Estimates of snow depth, required for the conversion, are based on the modified Warren climatology (Warren et al., 1999; Ricker et al., 2014). This climatology is not defined in the Fram Strait or Greenland Sea, therefore, snow depth estimates are extrapolated. Moreover, interannual variability in snow depth is not captured by the climatology, which can potentially cause biases in the final sea ice thickness retrieval. In addition, high drift speeds can also cause biases in the ice thickness retrieval due to the timeliness of the satellite passes within one month. The typical uncertainty is in the range of 0.3 - 0.5 m, but may potentially reach higher values."

**General Comment 3**: This concern goes to Section 3.2. I have a few comments / questions here which I ask the authors to explain better and/or comment in their paper.

**1)** I would strongly recommend to assign an ice mass balance GAIN to a POSITIVE value of "MB" and an ice mass balance LOSS to a NEGATIVE value of "MB" and not the way done currently. It is confusing the way written.
**2)** Did you take into account how long sea ice stays in your region of interest? Or in other words: How long does a group of ice floes entering the Greenland Sea at Fram Strait need to travel the distance to Denmark Strait? Could this impact your estimates?
**3)** How did you compute the regional sea-ice volume? What is the region over which you compute the sea-ice volume?
**4)** Please carry out a unit check. Which physical units do V, QF and MB have? Do these fit together?

**5)** You combine the difference in the regional sea-ice volume of two consecutive months, e.g. January and February, with the sea-ice volume flux difference at the northern (QF) and southern (QD) end of your region of interest for February. I assume that the time for which the sea-ice volume data are "valid" are Jan 15 and Feb 15, i.e. the middle of the respective month, integrating over Jan 1 to 31 and Feb. 1 to 28. For which time period is the sea-ice volume flux estimate valid? To me February implies that it is also derived for February and is hence valid for Feb 1 to Feb 28. Please describe what you combined in more detail because to me the balance seems not closed the way it is computed / written. It seems to me that you are combining different time periods.

**Response:**
**1)** The notation and corresponding formula were change according your recommendation.

**2)** On average it take 3-4 month for sea ice to travel from the Fram Strait to the Denmark Strait (Mironov, 2004). Once the sea ice entered the region, the its volume added for in the regional volume balance (V(m+1)-Vm). For the interannual variations discussed, the travel time from the Fram Strait to the Denmark Strait does not impact the estimates.

**3):** Thank you, this information was missing. We added the following sentence to the text (page 7, lines 2-5):
"The regional sea ice volume was calculated for the area limited by 82∘N and 66∘N latitudes and boarder on the east shown in Figure 11a (green box). We slightly extended the eastern boundary of the Greenland Sea to the south-east, compared to its classical definition in order to include the entire area of the Odden ice tongue formation."

**4):** Thank you, there was a time variable missing in the equation (5). It is now corrected: V, MB are in $km^3$, QF is in $km^3$ $month^{-1}$.

**5):**  The regional sea ice volume and the sea ice fluxes in the computations are estimated using the same sea ice thickness data, averaged over the same month. The balance is then correctly obtained for the integral ice volume over the Greenland Sea. Here we neglect higher frequency (intra-monthly) variations. The point of this comment might be that after ice enters the northern part of the region, it might take time for it to travel to the central areas of the Greenland Sea, where it efficiently melts. However, the results of our study are obtained for interannual (cold-season-mean) variations. Averaged over the cold season and taking into account the ice travel time of 3-4 months (see above), we may consider the process of ice inflow and that of ice melt to be simultaneous on these time scales.

**General Comment 4:** A lot of the interpretation of the data is / needs to be based on the ARMOR data set period which begins in 1993 and ends in 2016. On the other hand, the main results obtained with PIOMAS with respect to sea-ice volume and sea-ice volume fluxes and sea-ice mass balances are for the period 1978/79 through 2017, hence a substantially longer period. The paper would benefit from adding a careful consideration and discussion of the considerably different trends in the sea-ice volume related variables for the shorter ARMOR period in comparison to the longer period. Conclusions might change.

**Response:**  Thank you, good point. We have calculated the trend in sea-ice variables since 1995 in order to exclude an anomalous sea ice volume flux thorough the Fram Strait in 1994, which would affect the linear trend. For this shorter period the trends SIV and SIF lose their statistically significance, as the lengths of the time series, used for the computations, are now much shorter. Nevertheless, the magnitudes of the trends remain close to those derived for 1979-2015 (see table below), which indicates that the two results are comparable. As the long-term trends (1979-2015) represents the changes in sea-ice parameters with a higher formal statistical accuracy, we keep this result in the paper. However, the linkage between ocean and sea ice is analyzed based on a shorter time series (1993-2015), limited by the ocean data-base.

Trends (1995-2015) in monthly mean characteristics in the Greenland Sea calculated over annual (September-August), winter (October-April) and summer (March-September) periods: sea ice volume (SIV, km$^3$/year), sea ice volume loss (SIV loss, km$^3$/year), sea ice flux though the Fram Strait (SIF Fram, km$^3$/year), water temperature in MIZ (Tw, °C/year) and in the West Spitsbergen Current (TWSC, °C/year), heat flux across the Svinoy section ($Q_{Svinoy}$, TW/year).

| parameter | season | trend | $r^2$ | STD | p-value |
|---|---|---|---|---|---|
| SIV, km$^3$/year | annual | -8.80 (-0.89%) | 0.26 | 2.31 | 0.03 |
| | winter | -8.12 (-1.25%) | 0.27 | 3.19 | 0.02 |
| | summer | -6.82 (-1.05%) | 0.15 | 3.75 | 0.09 |
| SIV loss, km$^3$/year | annual | 0.57 (0.53%) | 0.02 | 0.92 | 0.54 |
| | winter | 1.01 (0.94%) | 0.03 | 1.46 | 0.50 |
| | summer | -0.36 (-0.33%) | 0.01 | 1.07 | 0.74 |
| SIF Fram, km$^3$/year | annual | 0.89 (1.11%) | 0.05 | 0.91 | 0.34 |
| | winter | 0.81 (1.02%) | 0.02 | 1.37 | 0.56 |
| | summer | 0.85 (1.05%) | 0.08 | 0.67 | 0.22 |

**General Comment 5:**

**1)** The period considered starts in winter 1978/79 and hence at a time when Is-Odden events occurred quite regularly. The paper lacks a discussion of the results with respect to the Is-Odden variability and, in particular, about the practical absense of the Is-Odden since about 2004 (?).

**2)** In addition, the paper lacks a discussion about the validity of the usage of an average MIZ area in a highly dynamic region where, thanks to the Is-Odden, sea-ice edges can be located substantially further off-shelf than suggested by the MIZ area chosen. Particularly in the context of Equation 7 usage of an actually varying MIZ might change the picture.

**3)** Finally, the period also covers the so-called ice-surge years 1989-1991 when a lot of the really thick and old ice exited the Arctic Ocean through the Fram Strait. A discussion of whether this is visible in the results or not (and why not) would also nicely complement this paper - perhaps even more than the relatively hypthetical considerations about NAO-Index links with water mass properties, circulation changes, and mixed layer depth variations.

**Response:**

**1)** The interannual variation in Odden occurrence is linked to the local surface temperature, local wind and on the large scale - to variations in NAO phase (e.g. Germe et al. 2011, Rogers and Hung 2008, Comiso 2001, Shuchman et al. 1998). The idea that the ocean may be important in modulating the formation of Odden tongue was proposed by Visber et al. (1995). Germe et al. (2011) showed that the occurrence of the Odden feature is not linked neither to the regional sea-ice variability, nor to the Fram Strait sea-ice areal flux. We also did not find any link between the sea-ice variables and the time series Odden occurrence from literature. On the other hand, the increase in the ocean heat content between 1993-2016 are visible in the area of Odden formation. At this stage, we only provide an addition argument in favor of further quantification and understanding of the oceanic influence on the Odden formation. We added few sentences to the discussion (page 13 lines 11-14):

"The interannual variations in sea ice area were previously linked to variations in air temperature (Comiso et al., 2001). The results of our paper permitted to speculate, that ocean temperature may be important in controlling Odden formation (see also Shuchman et al. (1998); Germe et al. (2011)). E.g. the reduction of Odden tongue occurrence in 2000s (Latarius and Quadfasel, 2010) might be partially driven by the increase in upper ocean heat content (Fig.5b)."

and (page 10 lines 27-34):

"With a stronger melting of sea ice at the seawards part of the MIZ, together with the ice volume loss, we should observe a sea ice area loss. This is consistent with Germe et al. (2011). In particular, positive water temperature trends over the eastern part of the Odden region suggest an overall decrease of the Odden formation by the end of the study period. The mean temperature trends over the Odden region (the area within the dotted line in Fig.15b) is $0.08 \circ$ C per year, i.e. there is an area-mean increase by $1.8 \circ$ C from 1993 to 2016. This exceeds the mean ocean temperature increase, averaged in the MIZ area (Eq.7), which includes the northern shelf break regions with negative temperature trends. Therefore, the estimates of the heat available for the ice melt, based on the values presented in Eq.(7), should be considered as the lower limit of the heat release within the Odden region."

**2):** In order to justify the validity of average MIZ, we added the following information to the text (page 7 lines 21 -34):

"The position of the real MIZ strongly varies in time and along the EGC, being a function of local direction and intensity of sea ice transport by wind and current, variation in the characteristics of ice transport from the Arctic and interaction of ice floes, local ice thermodynamics, etc. Presence of melting sea ice, in turn, affects the upper ocean and air temperatures. A warmer winter ocean warms

up the air, which can further be advected over the sea ice causing its melt away from the sea ice edge. Furthermore, an anomalously warmer ocean may prevent (or delay) formation of a new ice. All these distant factors certainly affect the MIZ position. However, if we estimate ocean temperature variations only along the actual MIZ, we do not account for these effects. The considerations above show that defining the oceanic region directly and indirectly affecting the ice volume in the sea is not straightforward. In this study we define interannual variations of ocean temperature in a fixed region, which is defined as an area enclosed between the 500-m isobath, marking the Greenland shelf break, and the mean winter location of the sea ice edge (Fig. 11). Using the fixed region also assures compatibility of interannual temperature variations. For the computations, the sea ice edge was defined as the 15% mean winter NSIDC sea ice concentration for 1979-2016. For brevity we further, somewhat deliberately, call this region the MIZ area. We further will see that temperature trends remain positive and of the same order of magnitude all over the western Greenland Sea, except for a few limited areas along the shelf break. This assure robustness of the results to the choice of the study region."

As the trends in Figure 5b are all positive and of the same order of magnitude, some reasonably sizable variations in the position of the eastern boundary of the study region makes no difference to the result. The following discussion is added (page 10 lines 10-34):

[revised manuscript text omitted]

**Specific comments:**

**Comment:** Page 1 - Line 18: From where is "oceanic buoyancy advected to the sea"? Which sea?

**Response:** The sentence is re-phrased: "as well as oceanic buoyancy advection into the region."

**Comment:** Page 2 - Line 2: "by solid ice transport" –> do you refer to sea-ice transport? Then I suggest to name it like this and then add something like "melting outside the Arctic Ocean"

**Response:** The sentence is re-phrased.

**Comment:** - Line 10: Did Ricker et al. (2018) also exclude extreme negative NAO events? If not then please re-formulate the sentence accordingly.

**Response:** Thank you. Ricker et al. (2019) did not exclude extreme negative NAO events. The sentences are re-phrased:
"There is a moderate correlation (0.62) between between NAO index (excluding extreme negative NAO events) and winter sea-ice area flux through the Fram Strait over 24 years of satellite observations (1978-2002) (Kwok et al. 2004). A higher correlation (0.70) between NAO index and winter sea-ice volume flux based CS2 data (2010-2017) is reported by Ricker et al. (2018)."

**Comment:** - Lines 13-15: Please make sure you write sea-ice volume flux where you refer to volume flux and sea-ice area flux where you refer to area flux. Here it remains unclear what "sea ice flux" is.

**Response:** Thank you, corrected

**Comment:** - Line 16/17: I am not sure the statement about the sea-ice production holds the way written, because "sea-ice production" is not just about sea-ice area but also about sea-ice thickness and/or volume. I did not find any hint about sea-ice volume in Germe et al. (2011). It is a tricky region. Perhaps you could split this statement into two parts: one related to the sea-ice on the shelf which particularly in the northern part (i.e. between Fram Strait and 75 degN) experiences a lot of fractioning and lead openings in which sea-ice forms quickly and to considerable thicknesses while the other one related to the off-shelf new ice formation in the Is-Odden tongue area, which is mostly thin, grease and pancake ice, sea ice. I agree with you that the largest variability is observed in the Is-Odden region but, to my knowledge, we also simply don't know anything about the variability of sea-ice production on the Greenland Sea shelf.

**Response:** Thank you, the sentence is re-phrased:

"The sea ice production in the Greenland Sea takes place east of the shelf between 71-75$^o$N and north of 75$^o$N withing the highly dynamic pack ice transported southwards along the Greenland coast. The latter fills in cracks and leads and can reach considerable thickness. While the sea ice forming east of the shelf is mostly thin newly-formed ice."

**Comment:** - Lines 32/33: "Shorter time series" <–> Figure 3c in Spreen et al. (2009) does not go along well. I suggest to rewrite this statement.

**Response:** Thank you, clarified this in the text:

"A combined time series of of sea-ice volume flux through the Fram Strait (1990-1996 (Vinje et al. 1998; 1991-1999 (Kwok et al., 2004) and 2003-2008 (Spreen et al., 2009)) shows a shift towards lower fluxes in the early 2000s compared to 1990s. However, the later study of Ricker et al. (2018) reveals that the sea-ice volume flux in 2010-2017 is similar to that in 1990s. Due to different uncertainties in the data used by the cited authors and to different methodologies used in those studies, it not possible to merge their results to get uninterrupted data-set for the entire period from 1990 to 2017. Although individual studies do not present significant trends in the volume flux, the overall tendency remains unknown."

**Comment:** Page 3 - Section 2.1 general: Please provide information such as grid resolution and type, time step (6-hourly?, daily?), etc. with which you used the PIOMAS data.

**Response:** We added this information:
"The original monthly PIOMAS sea ice thickness data were re-gridded to 25 km EASE-2."

**Comment:** - Lines 7-10: Please be more specific with the data sets assimilated into PIOMAS, e.g. which algorithm the sea-ice concentrations are based upon, what the origin of the sea-surface temperature data set used and what kind of NCEP/NCAR data is used? Is the latter from re-analysis?

**Response:** We have slightly changed the sentences, the detailed information can be found in the referred literature:

"It assimilates NSIDC (National Snow and Ice Data Center) near-real time sea daily ice concentration, daily surface atmospheric forcing and the sea-surface temperature in the ice-free areas from NCEP (National Centers for Environmental Prediction)/NCAR (National Center for Atmospheric Research) reanalysis (Zhang et al., 2003, Schweiger et al., 2011)."

**Comment:** - Line 17/18: I suggest to use "inter-comparison" instead of "cross-validation". What you carry out is not a validation - mainly because you don't have the true sea-ice thickness at hand. The same applies to later usage of this term.

**Response:** We agree that "inter-comparison" is a better term. Corrected.

**Comment:** - Line 19: While you describe the CS-2 data in Section 2.2 you don't describe the ULS data (which you state here to be used for the "cross-validation" of the sea-ice volume)

**Response:** Thank you, the presence of ULS data in the text is confusing. It was used for sea ice volume flux estimation in Kwok et al., 2004. To avoid confusion we removed "ULS" from the text and refer to the data set as "observation-based".

**Comment:** - Line 25/26: What kind of a grid is this? "spatial resolution" –> "grid resolution".

**Response:** Changed:
"The CS2 data-set provide monthly average sea ice thickness on EASE-2 grid with 25x25 km spatial resolution from 2010 to 2017."

**Comment:** - Line 29: I find your variable notation quite confusing and not to the point (here and again later in your paper). Suggestion: SIC –> C, HI –> I, HIE –> I_eff , i.e. with "eff" as a subscript. You could drop the "i" in the subscript and simply write in the text that you carry out this computation for every grid cell.

**Response:** Corrected.

**Comment:** Page 4 - Lines 12/13: How are the vertical density profiles computed? Are these part of the ARMOR data set or did you compute them on your own? Are the mentiond current velocities relevant for your paper? Are these available with the same grid resolution?

**Response:** The routine computations of water density is done using UNESCO 1981 equation of state of the seawater. We use current velocities for computation of oceanic heat advection through selected sections. The currents are gridded into the same spatial grid as the T-S data. To avoid ambiguity we re-phrased the sentence as:

"The oceanic heat fluxes are estimated using currents from the ARMOR data-set with the same spatial and temporal resolution. The current velocities at various depth levels are obtained by extrapolating the sea-surface current from the satellite altimetry, downwards using the thermal wind relations. The vertical density profiles, used for the computations, are assessed from the previously obtained temperature and salinity profiles (Mulet et al., 2012)."

**Comment:** - Lines 18-20: It is not entirely clear to me from how many profiles (?) with which average inter-profile distance (?) data contribute to the time series used. What is meant by "the core"?

**Response:** The entire paragraph is re-written:

"Long-term series of monthly gridded water temperature is obtained from "The Climatological Atlas of the Nordic Seas and Northern North Atlantic" (Korablev 2007}. The data-base merges together data from ICES (International Counsel for Exploration of the Sea), from IMR (Institute of the Marine Research), from a number of international projects (ESOP, VEINS, TRACTOR, CONVECTION, etc.), as well as from Soviet Union cruises in the study region. However, there are

too few observations in the EGC before the 2000s. In this paper we use long-term temperature time series in the much better sampled upper WSC at 78ºN, west of East-Fjord (Fig. 1). The depth averaged water temperature at 100-200 m is used, as this layer is dominated by the Atlantic Water and it is not directly affected by heat exchange with the atmosphere all year round. This results in the highest temperature at these depths during cold season. Even this region was sampled in a quite irregular manner, with a lower sampling frequency in winter. Since 1979, the average number of samples was 161 per year, varying from, on average, 2-5 per year from November to May to 20-35 per year from June to October. The data-gaps in the time series were filled in by kriging with the 30-km window. The interannual variations presented in this study were averaged over the months the most densely covered with data (June to September)."

**Comment:** - Line 21: Would it do any harm on the data set to also include data from May? That way you would comply with your earlier definition of summer: May through September.

**Response:** The data in May are too scares and were not included in the mean values. This certainly does not affect the observed interannual trends.

**Comment:** - Lines 24-29: Which sea-ice drift data set is used? Is this quantity provided by PIOMAS? You have introduced the effective sea-ice thickness already before and can delete the second sentence here, changing "sea-ice thickness" to "effective sea-ice thickness" in the first sentence. Did Sumata et al. (2014/2015) also include PIOMAS and/or the sea-ice drift data set you used in their inter-comparison studies?

**Response:** We use NSIDC Pathfinder v 3 data It is now mentioned in the text. Sumata et al. (2014, 2015) used the version 2 of the NSIDC data-set. The redundant sentences about the effective ice thickness is removed.

**Comment:** Page 5: - Equations 2 to 4 and related text: Following up with my comment to Equation 1 I suggest that you also here change the notation. It seems that you need to use super-scripts to indicate the source of the data, i.e. I^CS2_eff for the effective sea-ice thickness from CS-2 (see Eq. 1) and I^PIOMAS_eff for the effective sea-ice thickness from PIOMAS. On which grid is this computation carried out? If l = 25 km = constant distance between grid cells (or grid cell centers?), then it needs to be a grid such as the EASE-grid? Please be more specific here. Furthermore, usage of D_x and D_y suggests that your drift data set indeed only contains drift components relative to the grid (which?) on which the data set is provided and does NOT contain the true u (West-East positive) and v (South-North positive) motion components? May I nevertheless suggest that you change "D" to something like "v" for velocity or, even better, "u" and "v" (of course keeping the sub-scripts x and y)? If you then also replace "l" by "d" for distance then equations 2 and 3 might be more understandable at first glance.

**Response:** We agree that in this form the understanding of equations requires some time. Nevertheless, we decided to leave the equations like they are in order to keep it consistent with Ricker et al. 2018. The calculations are performed at the EASE-2 grid. As it mentioned in the data description, the CS2 is originally on EASE-2 grid and the PIOMAS data was converted to EASE-2 grid.

**Comment:** - Lines 9-11: I suggest to term this sea-ice volume flux component QD. I suggest to refer to Figure 1 for illustration of the location of this gate. Is QD defined positive when leaving the Greenland Sea?

**Response:** The gates are now illustrated in Figure 1. We also introduced QD in the text:

" A similar methodology was used to assess the sea-ice volume flux through the Denmark Strait (QD) along the meridional section (66ºN and 35ºW – 20ºE). The positive sign of QD corresponds to the sea ice volume outflow from the Greenland Sea."

**Comment:** - Lines 11-15: It might make sense to put these lines into a new paragraph, starting with "In order ...". I don't understand what you did here. Did you read the figures of the sea-ice volume fluxes from the papers or did you carry out the entire computations again on your own or did you copy the figures? Please be more specific in what you did. Please also stress that in case of Spreen et al. (2009) you only used the ICESat data part.

**Response:** Thank you, it is supposed to be a new paragraph. The monthly fluxes from Kwok et al. (2004) and Spreen et al. (2009) are presented as tables in the corresponding papers. The flux from Ricker et al. (2018) was provided by the author. From Spreen et al. (2009) we use monthly-mean flux derived using weighted ICESat thickness data. The interested reader can refer to the cited literature for details.

**Comment:** - Line 17: "formed due to thermodynamically" ?? please re-phrase

**Response:** Thank you, corrected:
" lost or gained due to due to freezing or melt.."

**Comment:** - Lines 28-30: Did you use density or potential density? You text is confusing here.

**Response:** For computation of the vertical density gradients we use potential density. We made the corresponding changes in the text of this paragraph:

"The method is similar to that used by Pickart et al. (2002), but is applied to the vertical profiles of the potential density gradients. Before processing, the potential density profiles were filtered to remove the small-scale noise. The gravitationally unstable segments were artificially mixed to neutral stratification. The MLD is defined as the depth where the vertical density gradient exceeds its two local standard deviations within a 50-m window, centered at the tested depth."

**Comment:** Page 6 - Line 2: "tested point" –> perhaps better: "tested depth"?

**Response:** Thank you, corrected.

**Comment:** - Lines 4-7: Please motivate your choice of defining the MIZ. I am asking because the inter-annual variation of the MIZ certainly results in actually much larger or much smaller areas to be considered. Particularly for winters before 2004, when the Is Odden was observed more often than after 2003, this definition would mean that the MIZ is defined for a much smaller region than actually occupied.

**Response:** This comment virtually repeats the **General Comment 5 2)**. We added the required information to the text (please see the answer to the **General Comment 2)** above).

**Comment:** - Lines 11-15: Please write where this transect Q is located. If Q is located along a latitude, isn't d_x constant? I understood that the ARMOR data set as 1/4 degree resolution, so that neighboring data points are separated by the distance corresponding to 1/4 of a degree at the latitude of Q. If not - how is d_x computed? In Equation 6, I suggest to use a small "v" for the current speed and instead of the subscript "w" use "water" to avoid confusion with the vertical velocity component which is usually termed "w". Are density and specific heat of water constants or do these vary with temperature? Is 1030 kg/m^3 a valid value for the Greenland Sea? d_z denotes the

"processed depth level" but the index "i" in Q_i and T_i denotes the i-th grid cell? Perhaps it makes sense to re-write Equation 6 with two integral signs, one over dx and one of dz? Please write the motivation to use T_ref = -1.8degC (because you want to estimate the role of this heat flux in melting sea ice).

**Response:** The computed total oceanic heat flux is indeed an integral over the section, where $d_x=1/4°$ and $d_z$ varies from 10 m with depth (as presented in the original ARMOR data-set we corrected the equation 6 to:
"

$$Q = \iint \left[ \rho c_p (T - T_{ref}) v \right] dxdz \quad (6)$$

where $\rho=1030$ kg m$^{-3}$ is the mean sea water density; $c_p = 3900$ J kg$^{-1}$ $°$C$^{-1}$ is specific heat of sea water; T is sea water temperature, $T_{ref} =-1.8\ °$C is the "reference temperature", v is current velocity perpendicular to the transect. The choice of the reference temperature is conditioned by study of the role of heat fluxes on melting sea ice."

**Comment:** - Line 26/27: This tail grows over time and is most pronounced in April. Are you able to assign a particular area in your region of interest to this tail?

**Response:** The referred tail grows from October to April with an increase of the fraction of thick ice in the region. The 'tail' values mainly fall in the dark blue area in Figure 1b.

**Comment:** Figure 1: - Why do you show data for the period September-April? You defined winter further above as October-April. This is confusing. - Which sea-ice concentration data set is used in Fig. 1 a? NSIDC offers a multitude of different data sets. - Did you interpolate the PIOMAS data onto the CS-2 grid or vice versa? - The color bar used as legend in Figure 1 b is empty. Please correct. - If possible I would enlarge the figure. -Caption: "isobash" –> "isobath"; state the time period (months, years) for which Fig. 1b) is computed.

**Response:**
- Thank you, it is a typo. We show the October-April trend in SIC.
- We used NSIDC Bootstrap Sea Ice Concentrations from Nimbus-7 SMMR and DMSP SSM/I-SSMIS, Version 2. The reference is added to the Figure 1 caption.
- The PIOMAS was interpolated to the CS2 grid. A clarifying sentence was added to the data description: "The original monthly PIOMAS sea ice thickness data were re-gridded to 25 km EASE-2."
- Thank you, the figure is updated.
- Unfortunately the figure can not be enlarged since we used the journal standard two-column figure in the Latex template. Probably typesetting solves this.
- Thank you, the typo is corrected and the time period is added to Figure 1 b.

**Comment:** Figure 2: - Again the question one which grid this comparison is carried out - I don't understand how the data points in Figure 2 h) are computed. It says area-mean ... but I find several points per month, as if several sub-areas were used. - While the color coding of Figure 2 a) to g) and its usage in Figure 2 h) is nice, the scatterplots in a) to g) would benefit from color-coding the probability of a respective SIT data pair to occur. That way one cannot not use the color anymore in Fig 2 h) but there you could use different symbols and only provide ONE region mean value and express the variability of the area-mean monthly SIT by error bars denoting plus/minus one standard deviation for both data sets. - Caption: I note that image i) is not existent. That part of the caption should be deleted. - Please note the unit of the RMSE given in the scatterplots.

**Response:** The data comparison is performed om 25 km EASE-2 grid. It is now clear from the description of the data. Figure 2 h shows all monthly "snapshots" from November 2010 to December 2016. Therefore, there are several values for each calendar months.

Following the reviewer suggestion, we change the color-code in the Figure. However, we did not use month-average values and error bars for panel h. In our opinion, the current plot is more informative.

**Comment:** Page 7 - Line 6: "start decreasing" –> well, you might not want to exaggerate this finding, it is just for 2016 and 2017.

**Response:** -  Thank you, we removed this line.

**Comment:** - Lines 10-12: I guess your statements about the inter-annual and intra-annual variations in sea-ice volume flux hold - particularly in the light that PIOMAS is known to under-estimate thickness for thick sea ice and therefore not unexpectedly show a slight negative bias in the Fram Strait sea-ice volume flux compared to the other data sets. <–> But I am much less confident with the results about the sea-ice volume for the reasons laid out in GC2 and because Fig. 1 b) has very small areas where the difference PIOMAS minus CS-2 SIT is acceptably low. Positive and negative sea-ice thickness differences along your gate in the Fram Strait tend to cancel each other out and therefore the sea-ice volume flux agreement is good (By the way: There the CS-2 SIT dataset is potentially much more credible than, e.g. at 78deg N). The large bias at the Denmark Strait possibly is not to relevant because of the small flux value anyways. But the majority of the Greenland Sea shows a substantial bias between PIOMAS and CS-2 and you need to discuss whether this bias (if it is real) is relevant for your findings or not.

**Response:** We agree that there is a substantial bias between PIOMAS and CS2 in the region. As the reviewer has mentioned CS2 data has also rather high uncertainties at these latitudes. Nevertheless, Figure 2 shows that there is a high correlation between the two data-sets on month-to-month (Fig. 2A-g), as well as on year-to-year (Fig. 2h) time scales. Therefore, we believe that the relative interannual sea-ice volume changes are captured by PIOMAS and the data allows estimation of a conservative SIV trend. This is in agreement with conclusions of Schweiger et al. (2011) who performed a detailed investigation of PIOMAS uncertainties. The systematic PIOMAS error and its influence in the trends is discussed in Section 5.1

**Comment:** Figure 3: - I believe it is sufficient to show the mean monthly values for the three satellite / ULS data sets. One can see whether they are within the error margin of PIOMAS or not. If you want to provide the standard deviations of the three other data sets then you could do this in a Table, don't you think so. In any case Fig. 3 b) would become more readable without the dotted lines. - I am a bit confused about the different time scales. In Fig. 3 a) you show PIOMAS for 1991 to 2017 but in Fig. 3 b) your computations are based in one year less (2016)? - I have to admit that I don't like that the grey shaded area denotes the standard deviation over the entire period. Did you by chance play around with the data to see how this shaded area looks like when using exactly the same periods as used for the observations? Only in that case a check whether the observations fall into the shaded area or not makes sense. - The legend under Fig. 3 b) says Ricker et al. 2017 instead of 2018. - Fig. 3 c), y-axis: check unit. - Please enlarge the entire figure.

**Response:**
-Figure 3a shows PIOMAS time series up to 2016. The typo in caption is corrected.
- In Figure 3 b we removed the standard deviation curves for the observation-based data set. The PIOMAS standard deviation remains. We agree that the PIOMAS seasonal cycle computed for the entire period between 1991-2016 and its standard deviation is not directly comparable to the seasonal cycle of observation-based data. However, following Spreen et al. (2009) and Ricker et al.

(2018), we present this figure to give an impression of how well the different seasonal cycles fit to each other. Below we plotted the PIOMAS seasonal cycles for the same time periods as the observation-based data-sets. There is some general similarity with Figure 3 b: Kwok et (2004) fits fairly well to the PIOMAS curves, Spreen et al. (2009) results fit better during the second half of the year, the results by Ricker et al. (2018) show the same seasonal cycle, but are above the PIOMAS estimates.

The typo in the legend is corrected.

[Figure]

**Comment:** Lines 14-28 and Figure 4 and Table 3: - Please describe whether the seasonal (i.e. summer and winter) values shown in Figure 4 are total values, i.e. May+June+July+August+September, or mean monthly values for these months). I assume the latter. Possibly I overlooked something of this description in the text?

**Response:** The values are monthly means averaged over winter, summer and the whole season. We clarified it in the text and in the caption for Figure 4.

**Comment:** - Please explain why in Figure 4 (see caption) you re-define winter to Dec.-Apr and summer to May-Nov. while earlier in the paper you use Oct.-Apr. for winter and May-Sep. For summer; also for Table 3 you seem to have used the latter two periods.

**Response:** Thank you, this is a typo migrated from an earlier version of the manuscript. It is now corrected.

**Comment:** - Why do you refer the winter and summer trends to the annual mean sea-ice volume (lines 15-17)? Wouldn't it have been more straightforward to relate the seasonal trends to the respective seasonal mean values?

**Response:** We relate winter and summer trends to the long-term annual mean value in order to show their relative importance in comparison to the overall sea-ice volume in the region.

**Comment:** - I suggest to enlarge Figure 4 as a whole. That way you would be able to replace the "a", "w" and "s" in the annotation of the different colored lines by "annual", "winter" and "summer" and make the Figure as a whole more readable - because in this case you can also resolve the ambiguity in the annotation with "a" which so far means "AOI" in image a) but "annual" in image d).

**Response:** The figure size is set to the journal standard of a two-column figure. We replaced 'w', 's' and 'a' by 'winter', 'summer' and 'annual' in all panels.

**Comment:** - You forgot to describe what is shown in Figure 4 c). I assume these are the mean seasonal monthly mean sea-ice volume fluxes through Fram Strait?

**Response:** Thank you, we added the missing description for panel c) .

**Comment:** In general the caption of Figure 4 needs a revision since it should contain information about what "a", "w", and "s" mean. The unit of TWSC should possibly be just ◦C. For the ocean heat flux you might want to add "Q_Svinoy" in the caption as well as at the right y-axis annotation and use the currently present "TW" as the unit.

**Response:** Thank you, it is now corrected.

**Comment:** - I note that you display annual values in Table 3 but refer to decadal values in the text. It might be good to harmonize this and change the values in Table 3 to decadal values as well. - "unexpectedly goes along with an increase in the monthly ice volume flux through" –> "coincides with an increased sea-ice volume import through"

**Response:** Thank you, the sentence is re-phrased. We leave the annual trends in the Table 3 since the conversion to the decadal scale is straightforward.

**Comment:**- Since in Line 20 you state a significance level it might be good to do this for the trends in the total Greenland Sea sea-ice volume as well; these are even more significant it seems.
**Response:** The level of significance was added to the text.

**Comment:**- Table 3, caption: "summer (March-September)"–> "summer (May-September)"

**Response:** Thank you, corrected.

**Comment:**- Line 21: I don't understand where the 112.8 km^3 /decade come from. If I add up 12 times the monthly sea-ice import per decade (of 9.6 km^3) then I end up with 115.2 km^3 / decade - in case this is what you wanted to do.

**Response:** Thank you, your estimate is correct. The wrong value migrated from an older version of the manuscript.

**Comment:**- Line 22: "Fig 2" –> I guess this needs to be Fig. 3 a)

**Response:** Thank you, corrected.

**Comment:**- Lines 23-28: Please spend a bit more time and effort to describe what we see in Figure b) and relate it to Equation 5. I also suggest to exchange images b) and c). You could write that for quite a number of years the sea-ice volume loss is larger in summer than winter - which is not surprizing as summer is the main melting season. Fig. 4 c) kind of shows the left difference of Equation 5. Would it make sense to show an additional image in which you show the the right difference, i.e. the mean difference of the sea-ice volume of consecutive months? Such an additional image could aid in the interpretation of Fig. 4 b).

**Response:** We extended the description for Figure 4 b:

"For about a half of the years during the study period, sea ice volume loss in summer is higher than that in winter. However, there are a few years (1992, 1994, 2004-2007) when winter sea ice volume loss significantly exceeds the summer one. During these years an increased sea ice volume flux thought the Fram Strait is detected (Fig. 4c)."

Concerning the right part of Equation 5, it would not show much more additional information. It is clear from Fig. 4 b and c, that variations in SIV are defined by the Fram Strait sea ice volume flux component.

**Comment:** Table 3: - What is r^2? - What is the unit of the STD and for which period / over which data is it computed?

**Response:** Now this information is provided in the Table heading:
r^2 - coefficient of determination, STD - standard deviation (m), p-value - probability value.

**Comment:** Page 8 - Line 2: "downwards" –> "with depth"?
**Response:** Thank you, corrected.

**Comment:** - You use the upper 50-m layer and the upper 200-m layer when showing and explaining your results. Why two different thick water layers? Please motivate / explain in the text or change.

**Response:** The obtained results are the same, whether we use the upper 50-m layer or the upper 200-m layer. We added the following text to make the choice of the layers clearer for a reader :

"The sea ice is affected by the heat in the upper mixed layer, the depth of which varies on synoptic, seasonal and interannual time scales. Our analysis shows that the obtained tendencies are largely independent from the choice of the water layer, at least within the upper 200 m of the water column. In further analysis we present results for the upper 50 m layer (the typical summer mixed layer in the MIZ) and the upper 200 m layer (the typical winter mixed layer in the MIZ)."

For consistency, we further added lines on characteristics of both, 50 m and 200 m layers in Figure 6. There is no principal difference between the results.

**Comment:** - Line 8: "over the 200-m layer" –> "over the upper 200-m layer"
**Response:** Thank you, corrected.

**Comment:** - What is the reason to show the November 2∘C isotherms? Why not December or February?

**Response:** We clarified this in the text. The following text is added:

"Water temperature in November reflects the heat fluxes accumulated during the warm period. It shows the background conditions formed by the beginning of winter cooling, when sea ice start forming localy. However, the performed tests show that the tendency of the isotherm to approach the shelf break is consistent for different isotherms (from 1 to 3 °C), for different layer thickness (50 to 200 m) and for different months. The difference is only observed for winter months, when the whole upper 200-m mixed layer effectively releases heat and the interannual trends become insignificant."

**Comment:** - Line 10 and Figure 5 b): Please be consistent with what you show. In the text you speak about "linear temperature trends". In the caption of Figure 5 you write "linear change in temperature" and the title of Figure 5 b) says dT2016-1993 which could be interpreted as a plain difference between 2016 and 1993. Please correct and/or modify accordingly. If Fig. 5b) indeed shows a trend then you need to change the unit.

**Response:** Thank you, this was a typo in the captions, now corrected. The correct captions for Fig. 5 (b,d) are is "linear temperature trends (°C year$^{-1}$)" and "linear salinity trends (year$^{-1}$)".

**Comment:** - Figure 5 in general: I suggest to remove all Figure titles and put the respective information in the annotation of the legend and the caption.

**Response:** We removed the titles above the panels to avoid confusion.

**Comment:** - Line 11: You refer to the MIZ only and therefore "western" needs to be "eastern". - How realistic is the cooling in the northern part of the MIZ?

**Response:** Thank you, this is corrected.
The cooling in the northern part of the MIZ is consistent with the stronger sea ice and Polar Water transport. The upper ocean cooling in the western Fram Strait is also derived from the model study (Chatterjee et al., 2018)

**Comment:** - Line 12&14 and Figure 5 d): Same comment as for Line 10 and Figure 5 b)
**Response:** Figure 5 in updated.

**Comment:** - Line 12: "Fig. 4d" –> "Fig. 5D"
**Response:** The sentence is removed.

**Comment:** - Lines 13 and 16: Add "layer" behind "200-m"
**Response:** Thank you, this is corrected.s

**Comment:** - Line 16: "and over the MIZ area"? Would "in the MIZ area" be better? As far as I understood you, you concentrate on the MIZ, don't youd?
**Response:** Thank you, this is corrected.

**Comment:** - Lines 17/18: "From ..." –> this is one way to interpret this figure. Another way would be to interpret the early years' small temperature decreases from Sep. to Mar. as a negative anomaly; it is unfortunate that you don't have data before 1993. You could refer in this context to Figure 5b and Figure 4d, right?

**Response:** Thank you, we added the references to the Figures 5 and 4.

"The temperature increases during all seasons, but the strongest increase is detected in autumn (by 0.5 and 0.6ºC over the 24 years). The winter convection efficiently uplifts heat to the sea surface. The heat accumulated in summer is mostly released during winter. Figure 4d suggests the results can be extrapolated back to, at least, 1980, as the slope of the trend lines in temperature of advected Atlantic Water for 1980-1992 is practically the same as for the period discussed above."

**Comment:** - Lines 19-22: "The heat ..." –> I am not sure I understand what you want to state here. First of all, isn't it normal that the heat stored during summer & fall is released during winter? Secondly, an increasing (as you postulated) cooling from September to March (Fig. 6 a) can indeed by caused by an intensification of the vertical mixing and hence a more efficient ocean-atmosphere heat exchange. Also, it could be caused by a higher autumn water temperature but also by a lower March water temperature. What I am missing here is an attempt to relate the observed differences to the extent of the Is-Odden. Its formation and presence has a profound impact on the upper layer water mass properties. I would delete the Line 19/20 sentence part "decreasing the ...". This is a hypothesis.

**Response:** We agree that this is a standard situation. However, we talk about temperature trends in the upper 200-m layer, and it is not obvious that all additional advected heat in the layer will be release through the sea-surface. In the end of this paragraph, we wanted to highlight this result. The heat naturally goes to the atmosphere or to the ice melt. However, we do not have in-situ measurements to prove this with computations. We changed the end of the paragraph to:

"We observe a growing difference between September and March temperatures (Fig. 6a) together with a decrease of temperature interannual trends to insignificant in winter. The growing difference in temperature is observed in spite of the equal winter and summer trends in the heat inflow with the NwAC (see $T_w$ and $Q_{Svinoy}$ in Tab.3). Therefore, in the MIZ region, all additional heat, accumulated in the upper 200-m layer during summer, is uplifted to the sea surface by winter convection, preventing ice formation in the ice-free areas or melting the ice in the ice-covered ones. "

**Comment:** - Line 24: add "(not shown)" behind "in winter". For Figure 6 b) one could also postulate a step change between 1993-2006 and 2007-2015.

**Response:** Thank you, corrected. The step change is characteristic for this particular case, which reflects mostly winter situation. For summer or autumn, the trends do not show a step, but are rather monotonous. In Figure 6, we now have both, the results for the upper 50m and for the upper 200m layers.

**Comment:** - Lines 25/26: For the discussion of Fig. 6 b) you refer to Fig. 6 d); I'd see a much better association between Fig. 5 a) and 5 d) in the sense that the dip / peak around 1997/98 could be an anomaly.

**Response:** The reviewer possibly means that the peak in 1996-1998 forms the trend. However, the negative trend will persist if we remove this peak, which is easily seen from Figure 6d. However, to show the configuration of the isotherms during different years we also refer to Fig. 5a:

"This goes together with a decrease of the annual mean distance of the 2 or 3ºC isotherm to the shelf break (Fig. 6d): from 120 km in 1993 to 50 km in 2016 (see also Fig.5a)."

**Comment:** - Line 28: add "are" before "observed"
**Response:** Thank you, corrected.

**Comment:** - Lines 30-32: What explains the peaks in winters 2008/09 and 2010/11 in Fig. 6 c)? These are possibly the main reasons for the observed increase in MLD.

**Response:** If these years are removed, the trend remains, but the difference along the trend between mean MLD in 1993 and 2016 will be 30 m, instead of 50 m. This corresponds to an overall increase in winter vertical mixing in the Greenland Sea, where the intensity of deep convection increases from around 1000 m in the beginning of the 1990s to over 1500-2000 after the mid-2000s (Bashmachnikov et al., 2019).
The question by the referee is very difficult to answer, as the intensity of vertical mixing depends on a number different factors. This requires a full separate study. We may note that 2009 and 2011 were the years with an anomalously high density in the Greenland Sea north of Jan Mayen. We also note a low oceanic advection of heat during 2008 (the third highest MLD) and 2011 and anomalously high heat fluxes to the atmosphere due to a small extent of ice-cover during winter 2009. All these factors are favourable for the detected deeper mixing in MIZ during the mentioned years.

**Comment:** Page 9 - Line 1: These September temperature values are not shown somewhere, are they?

The time series of mean September water temperature in the 200-m layer is added to Fig. 4b.

**Comment:** Equations 7 and 8: - Please spend a subscript "water" to the density in Equation 7 and replace the subscript "L" in Equation 8 by "ice". - Replace "dq" by "dQ" in Equation 8.
- In the text you write 1.8∘C for 2016, in Equation 7 you used 2.0∘C. Please correct.
**Response:** Thank you, corrected.

**Comment:** - Lines 12/13: I don't agree with the way you estimate the sea-ice volume loss for the 24-year period. That trend you use (possibly from Table 3) is computed over the entire period, starting in winter 1978/79 and not for the period 1993-2015. Fig. 4 b) clearly shows that if one would compute a trend for the 1992/93 through 2014/15 winter time period it might be negative. Also, you use 12 months while in Equation 7 you insert the winter MLD change. It seems hence doubtful to use the entire year. It might therefore make sense to revise this estimate.

**Response:** Thank you, for the comparison of SIV loss and an increase in oceanic heat release, only winter months has to be taken into account. We corrected the estimates. We agree that out calculations would have had more weight if all trends were computed for the same period. If we shorten the period for sea-ice variable analysis to 1993-2015, the trend in SIV loss becomes negative, but not statistically significant (e.g. trend in winter SIV loss equals -1.19 km$^3$/year, p-value equals 0.47). Note, that if we exclude the season of extreme SIV loss in 1994 and compute trends from 1995 (see the answer to General Comment 4), the magnitude of the trends becomes very close to the long-term ones. Although we do not have the data on MIZ temperature and MLD before 1993, the indications of changes in ocean state since 1979 can be seen from temperature of West Spitzbergen Current.

**Comment:** - Line 13: "of ice needed to fuse" ? –> delete?
**Response:** Thank you, corrected.

**Comment:** - Lines 13-15: Would it make sense to also mention that a large fraction of your MIZ area is potentially not covered by sea ice anyways? Would it also make sense to mention that new ice formation in the Is-Odden area but also otherwise in your MIZ area counter-acts this heat release? Would it make sense to also mention that the heat not necessarily needs to reach the surface

but stays aways from the sea ice at some depth? My feeling is that one should not overlook the assumptions made.

**Response:** Some of the suggestion by the reviewer follows directly the discussion of Eq. 8 on page 12 lines 5-8 (some heat is directly released to the atmosphere and do not interact with ice, or goes to ice melt). Also during ice formation the upper ocean becomes more saline, which enhances the convection and increases heat release towards the sea-surface. However, sea ice melt may inhibit the heat release by increasing the haline stratification near the sea-surface. These issues are added to the discussion:

"Certainly, not all heat released by the upper ocean in the MIZ area goes to the ice melt. An unknown fraction of heat is directly transferred to the atmosphere through open water, ice leads or is advected away from the MIZ area by ocean currents and eddies. Melting the ice may additionally increase haline stratification at the lower boundary of the ice, preventing ocean heat contacting with the ice cover. However, the estimates above suggest that, the autumn warming of the upper MIZ region, limited from below by the winter mixed layer, is able to release the amount heat far exceeding the amount, sufficient for the observed reduction of SIV in the region."

**Comment:** - Line 22: "multiyear" −> Do you refer to multiyear ice here? In that case write it accordingly.
**Response:** Thank you, corrected.

**Comment:** - Line 23: Whom do you mean with "The authors"?
**Response:** Re-phrased.

**Comment:** - Lines 23/24: This is a global statement, perhaps too global. PIOMAS under-estimates thicker ice thickness and over-estimates thinner ice thickness. Please discuss this in more detail because, yes, the thick ice in the Greenland Sea has become thinner but at the same time the Is-Odden feature with a lot of thin ice has vanished.
**Response:** We added the text regarding PIOMAS uncertainties in response to you General Comment 5 3)

**Comment:** - Lines 25/26: "compared to know from literature fluxes" −> "compared to flux values known from literature"
**Response:** Thank you, corrected.

**Comment:** - Lines 29/30: Fig. 2 i) does not exist. I guess this needs to be Fig. 1 b). "is lower compared to" −> I'd say this applies to 2/3 of the meridional gate. Don't forget the zonal part of the gate where the differences are opposite. Don't forget also GC2 in this context. "the NSIDC sea ice drift" −> needs to be introduced in the data section. Version 2 is quite old, by the way. State of the art is Version 4.

**Response:** Thank you, the reference to the figure is corrected. We also clarified in the text that the meridional gates are the main gated for sea ice import to the region. For flux calculation we used the NSIDC Pathfinder Version 3 (product, which is now introduced in the method description.

**Comment:** Page 10 - Lines 2-7: As an outlook you could add that it might make sense to separately, in PIOMAS, look at the changes in sea-ice formation in the true MIZ, i.e. the actually ice covered area and not just the average MIZ as defined by you, and in the consolidated ice covered part on the shelf. There are many leads created in the wider Fram Strait area in which thin ice grows quickly and which is advected southward on the shelf, continuing to grow.

**Response:** We agree that formation of sea ice in crack and leads might have a large contribution to the energy balance. However such study requires a data of higher resolution than the PIOMAS has.

**Comment:** - Line 7: "intensification of in sea ice melt" ?
**Response:** Thank you, corrected.

**Comment:** - Line 24: "through to be mostly driven" ?
**Response:** Thank you, corrected.

**Comment:** - Lines 25-27: Please rewrite this sentence. It is confusing. Which "inconsistency"? Which "peculiarities"? "delution"? Does Polar Water have an influence on your area?
**Response:** The phrase is re-written as:
"The interannual variations in the vertical mixing intensity between the Atlantic water, the Polar water and the modified Atlantic water, returning from the Arctic in the southern Fram Strait, as well as variations in ocean-atmosphere exchange in that area leads to interannual variability of the Atlantic water advected by the EGC into the Greenland Sea (Langehaug & Falck, 2012)."

**Comment:** Page 11 - Line 5: "NAO phase increases of the intensity" ?
**Response:** Thank you, corrected.

**Comment:** - Line 17: Fig. 4 f) needs to be Fig. 4 d).
**Response:** Thank you, corrected.

**Comment:** - Line 33: "Governed by ..." This sentence is difficult to read; please re-formulate.
**Response:** The sentence is re-formulated:
A more intense convection, governed by thermohaline characteristics of the upper Greenland Sea, the ice extent and the intensity of ocean-atmosphere heat and freshwater exchange (Marshall and Schott, 1999; Moore et al., 2015), lowers the sea-level in the Greenland Sea (Gelderloos et al., 2013; Bashmachnikov et al., 2019). This in turn increases the cyclonic circulation in the region.

**Comment:** Figure 7: Here different winter and summer periods than in the rest of the paper are used. Why? Please motivate, change, or delete.
**Response:** The figure is changed in accordance with recommendations.

**Comment:** Page 12 - Line 19: Why "Therefore"?
**Response:** Changed to "This siggests that.."

Editoral stuff:

**Comment:**- I found "northern winds" and "northerly winds". Please use one term.
**Response:** Thank you, corrected.

**Comment:**- Check for "Oddin"
**Response:** Thank you, corrected.

**Comment:**- I found "accessed" in case where "assessed" should be used, e.g. Page 3, Line 17 or Page 5, Line 10.
**Response:** Thank you, corrected.

**Comment:**- It might enhance the flow of your paper if you always use the same term for the same parameter. Example: use "effective sea-ice thickness" all the time and not "effective ice thickness"
**Response:** Thank you, corrected.

**Comment:**- You have an issue with using "though" instead of "through". Please check.
**Response:** Thank you, corrected.

**Comment:**Page 1 - Line 16: "The 2/3 of" –> "Two third of …"
**Response:** Corrected.

**Comment:**Page 2 - Lines 11-13: there are some issues with blanks and parentheses. Please check.
**Response:** Thank you, checked and corrected.

**Comment:**Page 4 - Line 19: WSC needs to be explained. "quire" –> "quite"?
**Response:** Thank you, corrected.

**Comment:**Page 5 - Line 9: "months" –> "month" - Line 13: "while in other" –> "while in the other"
**Response:** Thank you, corrected.

**Comment:**Page 6 - Line 3: Check references mentioned here - Line 25: Put "Schweiger et al., 2011" in ()
**Response:** Thank you, corrected.

**Comment:** Page 7 - Line 23: "significantly" –> "significant" "sea ice balance of the sea" –> "sea-ice mass balance of the Greenland Sea"
**Response:** Thank you, corrected.

**References:**
(see also references in the updated version of the manuscript)

Renner, A.H., Gerland, S., Haas, C., Spreen, G., Beckers, J.F., Hansen, E., Nicolaus, M. and Goodwin, H., 2014. Evidence of Arctic sea ice thinning from direct observations. *Geophysical Research Letters, 41*(14), pp.5029-5036.

Mironov, E.U., 2004. Ice conditions in the Greenland and Barents seas and their long-term forecast.

---

## Author Comment (AC2) · 7 Oct 2019

Referee #2

Dear referee,
Thank you for reviewing our manuscript. The provided references and comments has helped us to improve the text. We addressed all you comments below.

General comments:
**Comment:** The development of the sea ice volume in the Greenland Sea is investigated, but how is the Greenland Sea defined? The red box in Fig. 1 marks the entire Nordic Seas,which consists of the Norwegian Sea in the east and the Greenland + Iceland Seas aswell as the east Greenland shelf in the west. I would rather say that you study the seaice volume in the Nordic Seas or western Nordic Seas with a focus on the marginalice zone. The inconsistent use of "the Greenland Sea", "the Nordic Seas", and "the Greenland-Norwegian region" etc. makes the paper a bit confusing to read and it is notclear to me over which region you actually computed the sea ice volume.

**Response:** According to classification of the International Hydrographic Organization (IHO) the Greenland Sea extends from the Fram Strait to the Denmark Strait. Its eastern boundary goes along the western coast of Spitsbergen, from the south-eastern point of Spitsbergen to Jan Mayen and further south to the north-eastern extreme of Island. The term Island Sea, presently often used to define the southern part of the Greenland Sea from Jan Mayen to Island, is not a part of the standard oceanographic classification of ocean basins. We do not use this term in this paper.
In the previous version of Figure 1 the Norwegian Sea was included in the study region. In the new version the eastern boundary is corrected (see green boundary in the new version of Fig. 1). We slightly extend the eastern boundary of the Greenland Sea south-eastwards, compared to its classical definition in IHO in order to include the in the study region the entire area of the Odden ice tongue. We also agree that "the Greenland-Norwegian region" is stylistically bad and replaced it with "The Nordic Seas".

**Comment:** The authors start by introducing the Greenland Sea as an important area for deep convection and that the intensity of convection is controlled by buoyancy fluxes, in particular the input of freshwater (and sea ice). However, little is said about the observed changes in local sea ice formation, the retreat of the ice edge, winter-time heat loss, and their combined effect on convection in the Greenland Sea which has varied substantially over the past four decades. See e.g. Visbeck et al. (1995); Marshall andSchott (1999); Moore et al. (2015); Brakstad et al. (2019).

**Response:** We have added the proposed references to the text.. Further, the possible effect of deep convection on the advective process are briefly addressed in the Discussion. However, our study only marginally touches these questionable issues.

**Comment:** Some statements about the amount of available data in the MIZ (in the ARMOR dataset) are required. How does the generally sparse data coverage along the east Greenland shelf affect your results?

**Response:** The number of vertical profiles in the Greenland Sea between 1993 and 2016 vary from 50 to 300 per year, on average 150 casts per year. ARMOR dataset also favors from additional use of satellite sea-surface data, particularly relevant for our study. It is, however, difficult to assess the accuracy of the data, as ARMOR assimilates all available in-situ casts.

**Comment:** It would also be good to compare your mixed-layer properties with observations (i.e. Nilsson et al., 2008; Pawlowicz, 1995; Brakstad etal., 2019). All of these papers show ocean surface temperatures well below $0^{\circ}$C during winter (in the MIZ and in the center of the Greenland Sea). This contradicts what you describe on Page 8 – Line 6-7, that the temperature is always above $0\circ$C

leading to sea ice melt. Furthermore, you have used the mean 15% sea ice concentration contour from 1979 to 2016 to define the MIZ. The position of the ice edge has varied substantially during this period (i.e. Moore et al, 2015). How does that affect your results?

**Response:** In order to justify the validity of average MIZ, we added the following information in the text (page 7 lines 21-34):

"The position of the real MIZ strongly varies in time and along the EGC, being a function of local direction and intensity of sea ice transport by wind and current, variation in the characteristics of ice transport from the Arctic and interaction of ice floes, local ice thermodynamics, etc. Presence of melting sea ice, in turn, affects the upper ocean and air temperatures. A warmer winter ocean warms up the air, which can further be advected over the sea ice causing its melt away from the sea ice edge. Furthermore, an anomalously warmer ocean may prevent (or delay) formation of a new ice. All these distant factors certainly affect the MIZ position. However, if we estimate ocean temperature variations only along the actual MIZ, we do not account for these effects. The considerations above show that defining the oceanic region directly and indirectly affecting the ice volume in the sea is not straightforward. In this study we define interannual variations of ocean temperature in a fixed region, which is defined as an area enclosed between the 500-m isobath, marking the Greenland shelf break, and the mean winter location of the sea ice edge (Fig. 11). Using the fixed region also assures compatibility of interannual temperature variations. For the computations, the sea ice edge was defined as the 15% mean winter NSIDC sea ice concentration for 1979-2016. For brevity we further, somewhat deliberately, call this region the MIZ area. We further will see that temperature trends remain positive and of the same order of magnitude all over the western Greenland Sea, except for a few limited areas along the shelf break. This assure robustness of the results to the choice of the study region."

**Comment:** It is interesting that the warming of the Greenland Sea and the MIZ can account for the sea ice volume loss in the area plus the increased sea ice export through FramStrait. However, as noted also in the specific comments, information about the role of the atmosphere is missing. This is crucial in order to obtain a more complete picture of the drivers for the observed development of the sea ice volume. As it stands, you assume that the atmosphere plays a minor role (Page 9 – Line 15 & Page 10 – Line 9).It is possible to quantify the fraction of heat released to the atmosphere, and the role of increased atmospheric temperature, using an atmospheric reanalysis product. I think that considering the atmosphere as well would make your conclusions more solid.

**Response:** We fully agree that solid conclusion about the oceanic input to the sea ice volume loss in the region can not be drawn without a proper analysis of the atmospheric data. However, the scope of this study, as stated in the last paragraph of the introduction, is to explore the linkage between sea ice and ocean. A consideration of the atmosphere requires a separate investigation, as the atmospheric heat content at the sea surface highly depends on the oceanic one (the sum of sensible and latent heat fluxes in the region is one of the main components of the lower atmosphere heat balance and is directed from the ocean to the atmosphere all year round). In this paper, we find an indication that the estimated increase in ocean heat content can solely be responsible for the additional sea ice volume loss. However, we do not state in the conclusions, that ocean is the only contributor to the sea ice loss.

**Comment:** I find the link between long-term variations in sea ice volume and the NAO a bit speculative. On page 11 – line 2 you write that several studies have shown that during positive NAO phase, the intensity of ocean heat flux to the Nordic Seas increases by 50%. However, neither of the studies referred to (i.e. Skagseth et al., 2004 and Raj etal., 2018) examines the oceanic heat flux/ heat transport into the Nordic Seas (rather velocity and volume transport). When Raj et al. (2018) discuss the increase of 50% they are talking about an increase in volume transport. What about

variations in temperature of the inflowing Atlantic Water? Based on the studies you refer to, I find the link between NAO and temperature/heat content in the MIZ exaggerated. Either focus less on the NAO link, or refer to literature that show the link more clearly, or investigate the link more thoroughly in this paper.

**Response:** This partly repeats the comments by reviewer 1. In fact, the increase of the volume flux leads to an increase of the heat flux in this region. We re-worked section 5.2, elaborating on the linkage between SIV and NAO and added a number of references. Please, see page 14 lines 12-35 -page 15 lines 1-11) in the new version of the manuscript.

Specific comments:

**Comment:** Page 1 - Line 16: What do you mean by "this region"? The Greenland Sea, the NordicSeas, or the North Atlantic? I do not think any of these papers state that 2/3 of thedeep AMOC originates from the Greenland Sea.

**Response:** We change the phrase to: "More than half of the deep AMOC water originated from the Greenalnd Sea (Yashayaev et al., 2007; Rhein et al., 2015)."

**Comment:** Page 2 - Line 1: Approximately 50% of the freshwater anomaly at the surface or of theentire water column? Also, what do you mean by "the Norwegian-Greenland region".The Nordic Seas? Changes in salinity of the northward flowing Atlantic Water are alsoimportant (ie. Lauvset et al., 2018; Mork et al., 2019).

**Response:** In the cited works authors talk about the entire water column (Petterson et al. (2006) also adds ice FW flux). We agree that the Atlantic inflow is also important and indirectly is accounted for in the studies cited in the manuscript. We changed the phrase to:

"The freshwater anomaly in the upper Greenland Sea primarily originates from variations in the freshwater flux from the southern Fram Strait, which is formed by mixing of the Atlantic and the Polar water, as well as by solid ice transport (Serreze et al., 2006; Peterson et al., 2006; Glessmer et al., 2014; Lauvset et al., 2018)."

**Comment:** Page 2 – Line 6: Another very relevant reference for sea ice flux through Fram Strait, and for comparison with your results, is Smedsrud et al. (2017).
**Response:** Thank you, we are aware of this study. It is cited in the introduction page 3 lines 18

**Comment:** Page 2 – Line 14: Please clarify what you mean by "even stronger linked to the ArcticDipole pattern". In addition, you should briefly introduce the Arctic Dipole pattern, as itmay not be clear to all readers what this is.
**Response:** We added the phrase, explaining the pattern. For furthert detailes the readers can consult the cited study:
"It is also argued that the interannual variations of the sea ice flux through the Fram Strait is even stronger linked to the Arctic Dipole pattern, that explains a higher fraction of the observed interannual variations in the sea ice area flux than either the AO or the NAO (Wu et al., 2006). The Arctic Dipole pattern is derived as the second sea-level pressure EOF over the Arctic, which has two centers of action: over the Laptev-Kara seas and over the Canadian Archipelago. The pattern represents an important mechanism regulating the ice export through Fram Strait (Wu et al., 2006)."

**Comment:** Page 2- Line 17: The Odden sea ice tongue has not been formed in the GreenlandSea since the early 2000s (ie. Moore et al., 2015). Since then, sea ice has been close to absent in the center of the Greenland Sea.

**Response:** Thank you, we added this information to the text.

**Comment:** Page 3 – Line 4: The detected variations of what?
**Response:** Thank you, corrected to : «the detected variabtions of sea ice mass balance»

**Comment:** Page 3 – Line 25: How is monthly sea ice thickness from the Cryosat-2 satellite data-set obtained?
**Response:** The Cryosat-2 satellite data-set contains monthly average sea ice thinkness informaition sonce Novemner 2010. We now provide references to the data description (Hendricks et al. (2016) and production Ricker et al. (2014). We also added a sentense to the data description (page 4, line 24-25):
"The CS2 retrieval is based on sea ice freeboard measurements that are converted into sea ice thickness assuming hydrostatic equilibrium".

**Comment:** Page 4 – Line 9: What do you mean by different weights? Please elaborate.

**Response:** Gridding is done using the standard Gaussian function, there the weight of each measurement decreases with the distance form the measurement point. However, for the equal distances the in-situ measurements are taken with a higher weights. The procedure is a multi-step complex algorithm, as for any gridded data-set. The details of the method for forming the data set an interested reader can find in the cited study.

We changed the phrase to: "The final monthly mean 3D temperature/salinity distributions are obtained through optimal interpolation of all observed in situ for this month together with the derived "synthetic" profiles, where in-situ profiles, in the vicinity of the point of the observations, are taken with a higher weights (Guinehut et al., 2012)."

**Comment:** Page 4 – Line 11: Include reference to the method used in the World Ocean Atlas data-set.
**Response:** The reference is added: "(as, for example, it is done in the World Ocean Atlas database, https://www.nodc.noaa.gov/OC5/WOD/pr_wod.html)."

**Comment:** Page 4 – Line 20-21: Interannual variations of what? In addition, replace " - the months the most densely covered with data" with "which are the months with densest data coverage".
**Response:** Thank you, corrected.

**Comment:** Page 5 – Line 10-11: Denmark Strait is between Greenland and Iceland, not all theway to 36E! Please use a different term for your meridional section (a section along the Greenland Scotland Ridge?), or separate it into several sections (ie. one west and oneeast of Iceland).
**Response:** There was a typo in the coordinates used to calulate flux thought he Denmark Strait. The gates are now illistrated in Figure 1 a.

**Comment:** Page 5 –Line 17: What do you mean by "due to thermodynamically within the Greenland Sea"? Please clarify.
**Response:**                    The                  sentence                  is                  corrected:
"In order to analyse the sea ice volume lost or gained due to local melt or freezing, we calculated the sea ice mass balance (MB) in the Greenland Sea."

**Comment:** Page 5 – Line 29: How were the density profiles filtered?
**Response:** The phase is changed to: "Before processing, the small-scale noise in the potential density profiles were filtered out with 10-m sliding means."

**Comment:** Page 6 – Line 3: How were you able to compare your MLDs with Kara et al. (2003)? None of their figures show MLDs in the Nordic Seas. de Boyer Montégut et al. (2004) are also looking at global mixed layers. I think it would be better to compare with observed MLDs from the Greenland and Iceland seas (Brakstad et al., 2019 and Vågeet al., 2015, respectively).

**Response:** In the text we refered to the methods suggested in Kara et al. (2003) and de Boyer Montégut et al. (2004). Instead of using figures we programmed the algorithms, described in paper and compared the results. We added the phrase:
"The obtained mean distribution of the MLD, seasonal and interannual variations of the MLD in the central Greenland Sea are consistent with observations (Vage et al., 2015; Latarius & Quadfase, 2016; Brakstad et al., 2019)"

**Comment:** Page 7 – Line 15-16: How does the negative trend in sea ice volume compare to those found in Moore et al. (2015) and Onarheim et al. (2018)?

**Response:** The studies Moore et al. (2015) and Onarheim et al. (2018) show the reduction of sea ice extent. In our study we look at the trends in sea ice volume. In general, the sea ice volume loss can be related to the loss of sea ice extent. The reduction in sea ice extent, including Odden tongue formation are partly described in the introduction and discussed in section 5.3. Now we added the references to Moore et al. (2015) and Onarheim et al. (2018) to the text.

**Comment:** Page 7 – Line 33: Unclear. Please expand. Atlantic-origin water in the EGC is capped by fresh/cold Polar Water and sea ice during winter, which will inhibit ventilation of theAtlantic Water. Våge et al. (2018) show that due to the retreat of the ice edge the last decades, Atlantic Water has been and is more likely to be ventilated in the EGC. However, we do not know if this takes place "regularly".

**Response:** Our estimates of the winter MLD shows this should happen quite regularly. The ice retreat is presumably one of the reasons. The phrases are changed to:
"A relatively warm AW is observed in the East Greenland Current (EGC), off the Greenland shelf break, below a thin upper mixed layer dominated by the cold PW. Our estimates of winter MLD shows that the AW should be regularly brought to the ocean surface by vertical winter mixing, which is consistent with observations (Håvik et al., 2017; Våge et al., 2018)."

**Comment:** Page 8 – Line 1-2: The temperature (and salinity) of the Atlantic Water in the EGC is not increasing downstream. Please clarify what you mean by "increasing southeastwards".

**Response:** We changed the phrase to: "The presence of the AW is observed in climatology as water temperature (and salinity) in the EGC increasing with depth from about 0 $^{\circ}$C at the sea-surface to 2-4$^{\circ}$C at 500 m."

**Comment:** Page 8 – Line 4: "West Islandic Current" is not typically used. Rather use "North Icelandic Irminger Current". A better ref. here would be Jónsson and Valdimarsson(2005) or Hansen et al. (2008).
**Response:** Thank you, we added the reference to Hansen et al. (2008) and replaced the West Islandic Current by North Icelandic Irminger Current.

**Comment:** Page 8 – Line 6-7: As stated in the general comments, you need to compare your data with observations and discuss the temperature uncertainty due to limited data in the MIZ. Temperatures of 0.1-0.2∘C in winter seems unrealistically high.

**Response:** As reviewer correctly mentioned the data are limited in the region. The ARMOR data are based on in-situ data (where available) and interpolated data elsewhere. The temperature uncertainty is close to zero where the casts or the satellite data were obtained. The uncertainty is unknown in the areas where there are no data. However, temperatures above zero are often observed in winter in the region (Latarius & Quadfase, 2016; Brakstad et al., 2019). Here we remind that, for the reasons presented above, we use the fixed region to derive temperature variations, so the near-surface temperature mentioned here is not always in contact with ice, thus can be close to zero.

**Comment:** Page 8 – Line 14: Perhaps you should show the mixed-layer depth for comparison with previous work (ie. Brakstad et al., 2019 and Våge et al., 2015)
**Response:** In fact we do show it in Figure 6c, to which we refer now:
 "Averaged over the upper 200-m, the typical depth of the winter mixed layer (Fig. 6c), the patterns of the mean distribution and of (a somewhat weaker) tendencies in temperature and salinity closely repeat those in Figure 5."
This value is consistent with Brakstad et al., 2019 and Våge et al., 2015, as now stated earlier in the manuscript.

**Comment:** Page 8 – Line 17: Clarify what you mean by "overall year mean increase of temperature".
**Response:** Changed to : "overall increase of annual mean temperature"

**Comment:** Page 8 – Line 19-22: These lines are confusing and hard to read. What do you meanby "decreasing the interannual trends to insignificant"? Please be more specific.
**Response:** Changed to: "We observe a growing difference between September and March temperatures (Fig. 5a) together with a decrease of temperature interannual trends to insignificant in winter, in spite of equal winter and summer trends in the heat inflow with the NwAC (see T_w and Q_Svinoy in Tab.3).

**Comment:** Page 8 – Line 28-29: Bondevik (2011) is gray literature (no peer review). I would encourage you to refer to peer reviewed literature. In addition, add "are" before "ob-served".
**Response:** Thank you, the typo is corrected. There is only one reference to grey literature and since it is relevant, we decided to keep it.

**Comment:** Page 8 – Line 28-30: Explain how this increases ice melt.
**Response:** We added a clarifing sentence:
" The eddies sweep sea ice and PW off and advect warm AW closer to the ice edge, resulting in increase in bottom and lateral sea ice melt"

**Comment:** Page 8 – Line 30-32: As stated in the general comments: How does your definition ofthe MIZ and the data coverage in the MIZ affect the results?
**Response:** The choice of the fixed region for defining interannual temperature variations is now justified in page 7 . Please, see the response to the related general comment.

**Comment:** Page 8 – Line 34-35: This corroborates the results of Lauvset et al. (2018) who examined the relationship between hydrography (and MLD) in the Greenland Sea and thetemperature/salinity of the northward flowing Atlantic Water.
**Response:** Thank you, we make a link to these study (page 11, lines 23-25):
"Since the winter mixing does not reach the lower limit of the warm Atlantic water at 500-700 m, the deeper the mixing, the more heat is uplifted towards the sea-surface, melting the ice in the MIZ, which is consistent with the findings of Lauvset et al. (2018)."

**Comment:** Page 9 – Line 6-7: The 20% depend on how you define the Greenland Sea.
**Response:** To avoid ambiguity,we replaced "Greenladn Sea" with the "study area"

**Comment:** Page 9 – Line 7: "additional heat release": In addition to what?
**Response:** Here we mean the heat released due to an increse in 200-m layer temperature by $2°C$ between 1993 and 2016. This should be clear from the equation and text above.

**Comment:** Page 9 – Line 13-14: It would be interesting to quantify the fraction of heat released to the atmosphere. This should be possible using atmospheric reanalyses.
**Response:** We agree, but this is not straightforward, as heat is consumed also by different processes (ice melting, mixing in vertical and horizontal). This will require a separate study. Please, also see the response to the related general comment.

**Comment:** Page 9 – Line 15-16: What about increasing atmospheric temperature?
**Response:** This repeats one of the general comments. Please, see the respond above. Here we do not state that the atmosphere does not play a role in the ice volume loss. We only compare the amount of oceanic heat to the lost volume of sea ice.

**Comment:** Page 9 – Line 28: Clarify what you mean by "the discussed above general PIOMAS tendency"
**Response:** We are not sure that we understand this comment. IN the text we mean "to the discussed above general PIOMAS tendency to underestimate sea ice thickness", which is discussed few lines above in the same paragraph.

**Comment:** Page 9 – Line 29: Figure 2i does not exist.
**Response:** Thank you, corrected.

**Comment:** Page 10 – Line 6: This sentence is not in agreement with Page 8 – Line 6-7 where youstate that no sea ice formation occur and that the surface temperature is always >0.Here you write that sea ice is formed locally and that the atm. play a role.

**Response:** There is no contradiction. On p.8 we talk about the climatic seasonal means over the upper 50-m of the whole MIZ, including the warmer south-eastern part of the study region. Here we talk about the sea-surface and episodic formation of the ice tongue over a colder sea-surface (sometimes for a week or two). However, we agree with the reviewer that ice advection should also be important, although in the cited papers this factor was considered less significant.

**Comment:** Page 10 – Line 11-12: "almost twice of" what? Please clarify.
**Response:** The sentence was re-phrased:
"The surplus of the amount of the heat, released by the ocean at end of the study period, is more than twice of that necessary for bringing up the observed sea ice volume loss..."

**Comment:** Page 10 – Line 25-27: These two sentences are very confusing. Which inconsistency? What local peculiarities? Do you need these sentences at all? If so, please re-phraseand be more specific.
**Response:** The sentence was re-phrased: "The interannual variations in the vertical mixing intensity between the AW, the PW and the modified AW, returning from the Arctic through the southern Fram Strait, as well as variations in ocean-atmosphere exchange in that area leads to interannual variability of the AW advected by the EGC into the Greenland Sea (Langehaug and Falck, 2012)."

**Comment:** Page 10 – Line 30: Where did you obtain data (heat fluxes) from the Svinøy section? Please include reference.

**Response:** We computed the heat fluxes through the Svinoy section, using ARMOR dataset.

**Comment:** Page 11 – Line 2: Raj et al. (2018) show a 50% increase in volume transport not oceanic heat flux. (See general comment).
**Response:** We substantialy changed the section and added a number of references. Please see the new version of the manuscript, page 14.

**Comment:** Page 11 – Line 12: You have not really discussed any eastward advection of Polar Water to the southwestern Norwegian Sea. How does this relate to your results? Please elaborate.
**Response:** This region is out of the scope of our main line. We refer here to previous studies.

**Comment:** Page 11 – Line 23-24: This sentence contradicts line 19, where you state that the summer NAO is not important?
**Response:** We state that only winter NAO index should be taken into account for accessing the interannual variations, including those in the intensity of the AW advection. Summer NAO is of little relevance. Many studies in the region take into account only winter NAO index.

**Comment:** Page 11 – Line 25: What do you mean by "main currents in the Greenland Sea"? Be more specific.
**Response:** "In spite of the stronger ice melt, the upper ocean salinity in MIZ, as well as along the EGC, as well as along the NwAC, increases during recent decades (Fig. 5d)."

**Comment:** Page 12 – Line 5-7: Maybe better to refer to Brakstad et al. (2019), Lauvset et al.(2018), and Latarius and Quadfasel (2016) that all look at interannual changes in MLDin the Greenland Sea during your period. Lauvset et al. (2018) and Brakstad et al.(2019) both discuss the role of increased salinity on the mixed-layer depth.
**Response:** Thank you, know we refer to the suggested studies: "The on-going increase in salinity of the upper Greenland Sea (Fig. 5d) during the recent decades favors the deeper convection (see also Lauvset et al., 2018; Brakstad et al.,2019)."

**Comment:** Page 12 – Line 9: Smeed et al. (2014) show a weakened AMOC.
**Response:** Smeed et al. (2014) talks about a relatively small AMOC decline after 2004, on the top of the overall AMOC intensification since the 1970s-1980s (shown also in Smeed et al., 2014). We added a phrase:
"However, during the latest decade, a stagnation or a possible reversal of the tendency is observed (Smeed et al., 2014)"

**Comment:** Page 12 – Line 20: "govern" is too strong. Line 23-24: "Atlantic Water advection into the MIZ largely contributes to the SIV loss" is more appropriate.
**Response:** Thank you, corrected.

**Comment:** Page 12 – Line 28: In the last paragraph: The link to NAO is speculative, and you havenot shown this link in this paper.
**Response:** We agree with the reviewer. We now put this as a plausible hypothesis:
**"**This suggest that the simultaneous tendencies in the long-term increase of SIF and of the Atlantic water transport are both linked to a higher intensity of atmospheric circulation during the positive NAO phase, and, possibly, to the positive AMO phase, often linked to the intensification of the AMOC since the 1980s."

Technical corrections:

**Comment:** Page1 - Line 16: Replace "The 2/3" with "Two thirds"

**Response:** Replaced

**Comment:** Page 1 - Line 18: What do you mean by "to the sea"? Into the Greenland Sea?
**Response:** Re-phrased

**Comment:** Page 2 – Line 1: Replace "through the Fram Strait" with "through Fram Strait". (Also the case for Page 2 - Line 6, 9 and 10 etc.)
**Response:** Replaced

**Comment:** Page 2 – Line 9: Should be "drive" not "drives"
**Response:** 'divers' is the correct form as it is related to the conditions of wind intensification.

**Comment:** Page 2- Line 11: The entire reference here should be within parenthesis. "(Kwok et al., 2004)" not "Kwok et al. (2004)". Also the case for "Schweiger et al. (2011)" on Page 6 - line 25 in example. Please go through all references and make sure they are consistent.
**Response:** Thank you, corrected

**Comment:** Page 2 – Line 34: Replace "Oddin" with "Odden".
**Response:** Thank you, replaced

**Comment:** Page 3 – Line 15: Singular vs plurals: Use either "the spatial pattern of PIOMAS icethickness agrees" or "the spatial patterns of PIOMAS ice thickness agree".
**Response:** Thank you, corrected

**Comment:** Page 3 – Line 15: Remove comma after "those".
**Response:** Thank you, corrected

**Comment:** Page 3 – Line 25: Should be "provides" not "provide"
**Response:** Thank you, corrected

**Comment:** Page 3- Line 26: Insert "the" before "CS2 data-set".
**Response:** The sentence was re-phrased

**Comment:** Page 4- Line 3: Insert "the" before "ARMOR data-set".
**Response:** Thank you, corrected

**Comment:** Page 4 – Line 7: Insert "depth" before "levels".
**Response:** Thank you, corrected

**Comment:** Page 4- Line 9: Replace "all observed in situ" with "all in situ observations".
**Response:** Thank you, replaced

**Comment:** Page 4 – Line 18: Remove comma before "used" and after "paper".
**Response:** Thank you, corrected

**Comment:** Page 4 – Line 19: Replace "quire" with "quite"
**Response:** Thank you, replaced

**Comment:** Page 4 – Line 21: Use "a" instead of "the" in "kriging with the 30-km window".
**Response:** Thank you, corrected

**Comment:** Page 4 – Line 25: Remove comma after "Note".Page 5 – Line 9: Remove "s" in "months".
**Response:** Thank you, corrected

**Comment:** Page 5 – Line 10: Denmark Strait should be with capital S.
**Response:** Thank you, corrected

**Comment:** Page 5 – Line 11: Replace "access" with "assess".
**Response:** Thank you, replaced

**Comment:** Page 5 – Line 13: Should be "were adopted" not "was adopted".
**Response:** Thank you, corrected

**Comment:** Page 5 – Line 13: Add "the" before "other".
**Response:** Thank you, corrected

**Comment:** Page 5 – Line 14: Replace "also is" with "is also".
**Response:** Thank you, corrected

**Comment:** Page 5 – Line 15: Should be "data-sets" not "data-set".
**Response:** Thank you, corrected

**Comment:** Page 5 – Line 27: Add "the" before "ARMOR data-set".
**Response:** Thank you, corrected

**Comment:** Page 6 – Line 3: Remove "de Boyer". It is written twice.
**Response:** Thank you, removed

**Comment:** Page 6 – Line 19: Should be "underestimates" instead of "underestimate".
**Response:** Thank you, corrected

**Comment:** Page 6 – Line 20: Remove "the" before CS2. Also the case on line 21.
**Response:** Thank you, removed

**Comment:** Page 6- Line 20: Remove "s" in "values".
**Response:** Thank you, removed

**Comment:** Page 6 – Line 21: Remove "the" before "Spitsbergen". Also the case on line 23.
**Response:** Thank you, removed

**Comment:** Page 6 – Line 23-24: Either use "PIOMAS tend to overestimate" or "PIOMAS overestimates".
**Response:** Thank you, corrected

**Comment:** Page 6 – Line 24: Remove "thickness".
**Response:** We believe that "thickness" is used correctly.

**Comment:** Page 6 – Line 26: "discrepancies" should be singular => "discrepancy".
**Response:** Thank you, corrected

**Comment:** Page 6 – Line 30: Remove "the" before "PIOMAS".
**Response:** Changed

**Comment:** Page 6 – Line 31: Replace "all are" with "are all".
**Response:** Thank you, replaced

**Comment:** Page 6 – Line 31: Add "of" after "correlation".
**Response:** Changed

**Comment:** Page 6 – Line 31: Add "the" before "Ricker et al. (2018) data"
**Response:** Changed

**Comment:** Page 7 – Line 16: Replace "comprises" with a more appropriate term ("was"?).
**Response:** Thank you, replaced

**Comment:** Page 7 – Line 22: Remove "for" before "about".
**Response:** Thank you, corrected

**Comment:** Page 7 – Line 23: Should be "significant effect" rather than "significantly effect". Alsoreplace "the sea" with "the Greenland Sea".
**Response:** Thank you, corrected

**Comment:** Page 8 – Line 1: Add "the" before "climatology".Page 8 – Line 9: "approaches" is an odd choice of tense when you talk about some-thing that happened from 1990s to 2000s. Replace with "approached" or "propagatedtowards".
**Response:** Thank you, corrected

**Comment:** Page 8 – Line 10: It should be "Jan Mayen" not "Yan Mayen".
**Response:** Thank you, corrected

**Comment:** Page 8 – Line 11: Replace "western" with "eastern". In addition, do you mean "Frontal Current" instead of "Front Current" (same for Page 10 – Line 23)?

**Comment:** Page 8 – Line 12: The "tendencies" are shown in figure 5d. Replace "Fig. 4d" with "Fig.5d".
**Response:** Thank you, corrected

**Comment:** Page 8 – Line 23: nearly doubles from 1993 to ?
**Response:** Changed to "from1993 to 2016"

**Comment:** Page 9 – Line 12: Remove "the" after exceeds. It is written twice.
**Response:** Thank you, removed

**Comment:** Page 9 – Line 22: Remove "thickness" after "thick ice".
**Response:** We left the sentence unchanged as this wording is used in the cited study.

**Comment:** Page 9 – Line 25: Should be "appears" not "appear". Also, replace "lower compared to know from literature fluxes" with "lower than those estimated by previous studies" orsomething similar.
**Response:** The entire paragraph is changed in response to another comment.

**Comment:** Page 9 – Line 27: Remove "the" before "data".
**Response:** Thank you, corrected

**Comment:** Page 10 - Line 1: Remove "the" before "sea ice volume".
**Response:** Corrected

**Comment:** Page 10 – Line 13: Replace "uptake" with "take up".
**Response:** Corrected

**Comment:** Page 10 – Line 17: "brining" should be "bringing".
**Response:** Thank you, corrected

**Comment:** Page 10 – Line 18: "later" should be "layer".
**Response:** Thank you, corrected

**Comment:** Page 10 – Line 19: Write "Nansen Basin" with capital B.
**Response:** Thank you, corrected

**Comment:** Page 10 – Line 29: Replace "Further" with "Farther".
**Response:** Thank you, replaced

**Comment:** Page 10 – Line 30: "Svinoy" should be "Svinøy". Also the case on Page 10 - line 34 and Page 11 – line 16 and 17 etc.Page 10 – Line 31: Remove comma after "Barents Sea".
**Response:** Thank you, corrected

**Comment:** Page 10 – Line 34: Remove "in" after "confirmed by".
**Response:** Thank you, corrected

**Comment:** Page 11 – Line 1: Use capital S in "Nordic Seas". Also the case for line 10 and 20.
**Response:** Thank you, corrected

**Comment:** Page 11 – Line 5: Remove "of" after "NAO phase increases".
**Response:** Thank you, corrected

**Comment:** Page 11 – Line 10: "Fram Strat" should be "Fram Strait".
**Response:** Thank you, corrected

**Comment:** Page 11 – Line 11: Replace "through" by "across" and use capital R in "Faroe-Shetland Ridge".
**Response:** Thank you, corrected

**Comment:** Page 11 – Line 12: Inconsistent capitalization of "water". Here you write "Polar Water",while in line 6 you use "Atlantic water". Please be consistent throughout the paper.
**Response:** Thank you, corrected

**Comment:** Page 11 – Line 28: Replace "is" with "was" after "more ice".
**Response:** Thank you, corrected

**Comment:** Page 11 – Line 29: Add "the" before "Odden ice tongue".
**Response:** Thank you, corrected

**Comment:** Page 12 – Line 7: Remove "the" after "favours".
**Response:** Thank you, corrected

**Comment:** Page 12 – Line 22: "MID" should be "MLD".

**Response:** Thank you, corrected

**Comment:** Page 12 – Line 23: Add "heat" before "necessary".
**Response:** Thank you, corrected

**Comment:** Page 12 – Line 25: "Froe-Shetland ridge" should be "Faroe-Shetland Ridge". This sentence is also incomplete. Please re-phrase.
**Response:** The paragraph was removed

**Comment:** Figure 1: The color in the right color bar is missing.
**Response:** Thank you, the figure was updated.

**Comment:** Figure 2: In the figure caption you describe panel (i) – "difference between mean PI-OMAS and CS2 effective ice thickness", but panel "i" is not included in the figure (only panels a-h).
**Response:** Thank you, the caption for panel (I) is removed.

**Comment:** Figure 4: Please write out what the legends "w", "s", and "a" mean.
**Response:** The letters are replaced by full words

**Comment:** Figure 5: The color bar in panel "d" has the wrong units. The panel shows change in salinity, but have units of◦C.
**Response:** Thank you, corrected

**Comment:** Figure 6: In the figure caption: Remove parenthesis after "cold season".
**Response:** Thank you, corrected

**Comment:** Figure 7: Is there missing a second y-axis for the normalized maximum MLD? If not,I do not understand what the values -1 to 1.5 in normalized maximum MLD mean.Please explain.

**Response:** MLD was  notmalized in the standard way:
MLD(normalized)=(MLD-mean(MLD))/std(MLD))
To avoid confusions, the right y-axis now shows  the non-normalized MLD (m).

**Comment:** Table 3: Explain all columns. (i.e. what is correlated in the column r2?)
**Response:** Now all columns are explained. $R^2$ is the coefficient of determintaion. It is a squared coefficient of correlation between the observed values and the ones modeld with the linear trend.

References:

please see the references in the updated version of the manuscript.

---

## Author Comment (AC4) · 7 Oct 2019

The comment was uploaded in the form of a supplement:
https://www.the-cryosphere-discuss.net/tc-2019-117/tc-2019-117-AC4-
supplement.pdf
* * *

---

## Author Comment (AC5) · 7 Oct 2019

The comment was uploaded in the form of a supplement:
https://www.the-cryosphere-discuss.net/tc-2019-117/tc-2019-117-AC5-supplement.pdf
* * *

---

## Referee Report (RR1)

**Second round: review of "Sea ice volume variability and water temperature in the Greenland Sea" by Selyuzhenok, V. et al.**

**Summary:**

The authors have significantly improved the content and structure of the manuscript, but I still have three major comments (listed under general comments) that I think the authors need to elaborate on before the paper is ready for publication.

I have also included several specific comments and technical corrections on the updated text. Most of these are related to unclear text and English grammar.

**General comments:**

I still think that you need to include a statement in the paper about the data availability in the MIZ (especially the generally sparse data coverage on the shelf/shelf break), even though you are not able to address the temperature uncertainty. You can also add, as you mentioned in the previous response, that the additional use of satellite sea-surface data is very relevant and makes your analysis of the ML properties more robust than only including in situ observations, which is a strength of the ARMOR dataset.

Without discussing or investigating the role of increased atmospheric temperature and changes in ocean-atmosphere heat fluxes you cannot really make the conclusions on page 13 – lines 14-15, page 15 – lines 8-9, and page 16 – lines 10-12. You have shown that an increase in AW heat flux CAN contribute to sea ice volume loss, but not that it actually does. The correlation and trend estimates support your hypothesis, but does not imply causation. I understand that a thorough investigation of the atmospheric component requires some work, and could be an entire separate study. However, if you want to make the conclusions above, you cannot avoid to include some discussion on changes in the atmosphere and its potential role for the observed changes in sea ice volume. You do not necessarily need to include any new analysis, but could base a brief discussion on previous studies (ie. Moore et al. 2015 etc.).

The discussion on the link between the NAO index and the ocean circulation is improved in the current version of the paper, but since you already include the winter NAO index in fig. 7: what is the correlation between the winter NAO index and the water temperature at Svinøy? Some studies indicate that the ocean circulation has been decoupled from the NAO in periods during recent decades, and that it is more closely linked to the wind stress curl (Lohmann et al., 2009; Foukal and Lozier, 2017; Asbjørnsen et al., 2019).

**Specific comments:**

Page 1 – Line 16-17: Only refer to Marshall and Schott (1999) or find more appropriate references than Visbeck et al. (1995) and Brakstad et al. (2019) to this line/statement.

Page 1 – Line 18: I do not think that any of these references state that more than 50% of the deep AMOC originates from "the Greenland Sea". It should be "the Nordic Seas" (ie. Chafik and Rossby, 2019). The fraction of contribution from the Greenland Sea to the Nordic Seas overflow water (the dense water that spill south across the Greenland-Scotland Ridge) is still not fully known. Some references looking at the origin of the Nordic Seas overflow waters are: Eldevik et al. (2009), Mastropole et al. (2017), and Jeansson et al. (2017).

Page 2 – Line 1-2: What drives the remaining 35% of the freshwater anomaly? Is this due to variability in the northward flowing AW?

Page 2 – Line 14-16: I would rather state that the Greenland Sea Intermediate Water and the Greenland Sea Deep Water (GSDW) are formed by wintertime convection in the central Greenland Sea. Note also that the GSDW has not been ventilated/formed since before the 1980s.

Page 2 – Line 17: Latarius and Quadfasel (2016) do not show MLDs exceeding 2000m. I think this statement is incorrect (see also Brakstad et al., 2019).

Page 5 – Line 9: It is still not clear from the text what you mean with "different weights". Please add the information from your response to the text (ie: "taken with different weights based on the inverse distance and type of measurement (in situ observations were given higher weights).").

Page 7- Line 19: There is a bias towards deeper mixed layers in the central Greenland Sea. The estimated MLDs in fig. 7 are always deeper than 600m, while both Latarius and Quadfasel (2016) and Brakstad et al. (2019) observe MLDs shallower than 500m in 2003 and 2012 for instance. This should be noted.

Page 8 – Line 7: What do you mean by "is conditioned by study of the role of heat fluxes on melting sea ice''?

Page 10 – Line 3: Tendencies in heat? Please clarify.

Page 10 – Line 8-9: Here you state that the ocean always melt sea ice in the MIZ. This sounds like a contradiction to line 12 on the same page where you state that sea ice is formed locally in winter. I know that you, in the first line (8-9), talk about the mean conditions over the upper 50m in the entire MIZ, but I would rephrase this line to avoid confusion. At least make it clear that the temperature can be well below 0 degrees C in certain regions of the MIZ, and that your average is affected by the defined area of investigation (you average over water temperatures > 1 degrees C in some places (ie. AW in the northeast, fig. 5) which is not representative for the real MIZ characteristics).

Page 11 – Line 25: Lauvset et al. (2018) do not examine the MIZ and heat fluxes toward the sea ice. They investigate how increased salinity and temperature in the northward flowing AW have led to deeper mixed layers in the Greenland Sea gyre. It would be more appropriate to include this reference in the next line (25-26).

Page 13 – Line 31: Clarify what you mean by "All the processes''.

Page 14 – Line 5-11: These lines are not necessary since the same information is repeated/elaborated in the next paragraph. Please remove these lines and make a smooth transition to the discussion of the NAO.

Page 14 – Line 15-16: Which regional studies? Please include reference.

Page 16 – Line 12-15: These lines repeat lines 7-12 and are therefore not necessary. Please remove.

**Technical corrections:**

Generally: Remove ''the'' in ''through the Fram Strait/Denmark Strait''.

Page 1 – Line 3: Replace ''sea ice volume (SIV)'' with ''SIV'' since SIV is defined in line 1 already.

Page 1 – Line 18: Replace ''originated'' with ''originates''.

Page 2 – Line 6: Use ''The'' instead of ''A'' before ''general surface circulation''.

Page 2 – Line 7: Remove comma after ''temperature'' and add comma after ''Polar Water (PW)'' and before ''and the Atlantic Water (AW)''.

Page 2 – Line 9-10: I would first state where the maximum PW is found then that it quickly decreases in the off shelf direction.

Page 2 – Line 11: Remove ''s'' in ''centrals''.

Page 2 – Line 12 and 13:  Do you mean ''Greenland Sea Intermediate Water''?

Page 2 – Line 20: Use ''are'' instead of ''is'' before ''primarily controlled''.

Page 2 – Line 23: Should be ''winds that drive'' not ''drives'' since it refers to "winds". Single vs. plurals.

Page 2 – Line 24: Remove ''between''. It is written twice.

Page 2 – Line 28: Replace ''that'' with ''since it'' before ''explains a higher fraction''.

Page 2 – Line 35: Replace ''is'' with ''was'' after ''the Odden sea ice tongue'' since it is ~20 years since this last occurred.

Page 3 – Line 1: Remove ''and'' before ''northwest of Jan Mayen''?

Page 3 – Line 5: Add ''the'' before ''large''.

Page 3 – Line 15: Replace ''The overall'' with ''An overall''.

Page 3 – Line 16: Add "In particular" or "particularly" to highlight the large changes in the Odden ice tongue area?

Page 3 – Line 21: Add ''the'' before ''1990s''.

Page 3 – Line 22: Add ''is'' before ''not possible''.

Page 3- Line 32: Use ''daily sea ice concentration'' instead of ''sea daily ice concentration''.

Page 3 – Line 33: Remove ''the'' before ''sea-surface temperature''.

Page 4 – Line 7: Remove ''s'' in ''affects'' and insert ''sea ice'' before ''thickness and volume''.

Page 4 – Line 17: Remove ''In the Greenland Sea'' at the beginning of the sentence since it is written at the end of the sentence as well.

Page 4 – Line 30: Remove ''In addition,'' since ''also'' is used later in the sentence.

Page 4 – Line 31: Replace ''timeliness'' with ''timing'' and remove ''the'' before ''satellite passes''.

Page 5 – Line 14: Remove 'the'' before ''satellite altimetry''.

Page 5 – Line 21-23: Replace ''However,'' with ''Since'', add comma after ''2000s'', remove ''. In this paper'', and use capital ''C'' in ''West Spitsbergen Current''.

Page 5 – Line 25: Add ''the'' before ''cold season''.

Page 5 – Line 28: Use ''a'' instead of ''the'' before ''kriging''.

Page 5 – Line 28-29: Replace ''over the months the most densely covered with data'' with ''over the months with densest data coverage''.

Page 6 – Line 20: ''20E'' should be ''20W''.

Page 6 – Line 20: Replace ''the'' with ''a'' before ''sea ice volume''.

Page 6 – Line 21: Replace "access" with "assess".

Page 6 – Line 24: Use "gate locations" instead of "gates locations".

Page 7 – Line 3: Replace "and boarder on the east" with "and by the boarder in the east".

Page 7 – Line 8-9: There is something strange with this sentence. It makes more sense if "to" before "sea ice volume loss" is removed.

Page 7 – Line 17-18: Remove "the" before "weakly", before "Dukhovskoy's", and after "mean distribution of".

Page 7 – Line 25: Remove "a" before "new ice".

Page 7 – Line 25: Replace "All these distant factors" with "All of these factors".

Page 7 – Line 27-28: Replace "the ice volume in the sea" with "the sea ice volume".

Page 7 – Line 28: Replace "define" with "examine" after "In this study we".

Page 8 – Line 6: Insert "and" before "v is current velocity".

Page 8 – Line 10: Remove "in region, PIOMAS" and add comma before "monthly".

Page 8 – Line 12: Should be "values" instead of "value".

Page 8 – Line 24: Replace "off" with "of".

Page 9 – Line 9: Insert "the" before "long-term".

Page 9 – Line 19: Remove "a" before "half of the years".

Page 9 – Line 22: Remove "s" in "trends".

Page 9 – Line 26: Remove "in" before "and sea ice volume flux".

Page 9 – Line 29: Remove "s" in "shows" after "estimates of winter MLD".

Page 10 – Line 6: Add "a" before "maximum".

Page 10 – Line 7: Remove "a" before "somewhat weaker".

Page 10 – Line 9: Remove "the" after "ocean melts".

Page 10 – Line 12: Replace "formed by" with "at", and add "the" before "winter cooling".

Page 10 – Line 12: "localy" should be "locally".

Page 10 – Line 12-17: I would first explain the evolution of the 2degC isotherm (lines 15-17), then explain that this evolution is consistent for different isotherms etc. (lines 12-15). The beginning of the first sentence (line 12) is also a bit confusing and unnecessary. I would rather start this sentence with "The tendency of the isotherm to approach the shelf break is consistent for different isotherms etc.".

Page 10 – Line 16: Remove "s" in "westwards".

Page 10 – Line 25: Remove comma after "area".

Page 10 – Line 26: Rather start this sentence with: "The decreasing temperature in both of these areas is consistent with …."

Page 10 – Line 30: Remove ''s'' in ''trends''.

Page 11 – Line 10: Remove comma before ''accumulated'' and after ''during summer''.

Page 11 – Line 15: Remove ''effect''

Page 11 – Line 17: Remove ''the'' before ''eddy formation'' and replace ''for the'' with ''during''.

Page 11 – Line 18: ''northely'' should be ''northerly''.

Page 11 – Line 18: Replace ''off'' with ''seaward''. Also replace "increase in bottom" with "increased bottom".

Page 11 – Line 23: Remove ''the'' before ''winter mixing''.

Page 11 – Line 25: Replace ''of'' with ''in'' before ''MLD''.

Page 11 – Line 31: Add ''and'' before ''the MIZ area''.

Page 11 – Line 31: Replace ''computations show'' with ''computations give''.

Page 12 – Line 1: Replace ''by'' with ''of''.

Page 12 – Line 3: Remove ''of'' after ice density.

Page 12 – Line 8: Replace ''contacting with'' with ''to reach''. Also remove comma before ''the autumn warming''.

Page 12 – Line 9: Replace ''the amount heat far exceeding the amount, sufficient'' with ''more than enough heat to account''

Page 12 – Line 17: Use ''than'' instead of ''that'' after ''the trend is lower''.

Page 12 – Line 19: Replace ''be also'' with ''also be''.

Page 12 – Line 23: "myltiyear'' should be ""multiyear".

Page 12 – Line 32: Replace "are the main gates" with ''is the main gate''.

Page 13 – Line 6: Remove ''the'' before ''sea ice volume'' and add ''the'' before ''Greenland Sea''.

Page 13 – Line 24: Remove ''for'' before ''in the Nansen Basin''.

Page 14 – Line 14: Replace ''leading to'' with ''which results in''.

Page 14 – Line 18: Remove ''of'' before ''the cyclonic circulation''.

Page 14 – Line 22: Add ''The'' before ''NAO phase''.

Page 14 – Line 25: Add ''of'' before ''PW''.

Page 14 – Line 26: Remove parenthesis before ''Blindheim et al.''.

Page 14 – Line 29: Remove ''a'' before ''higher heat fluxes''.

Page 14 – Line 30: Insert ''the'' before ''1970s''.

Page 14 – Line 34: Replace ''of'' with ''on the'' before ''order of''.

Page 15 – Line 11: Replace ''of'' with ''for'' before ''the shelf area''. Also remove parenthesis before ''Alekseev et al.''.

Page 15 – Line 12-13: Replace ''as well as along the EGC, as well as along the NwAC, increases during recent decades'' with ''as well as along the EGC and in the NwAC, has increased during recent decades''.

Page 15 – Line 13: Remove parenthesis around ''d'' in ''(Fig. 5(d))''. Also the case on line 25.

Page 15 – Line 24: Insert ''the'' before ''1990s''.

Page 15 – Line 28: Insert ''the'' before ''1980s''.

Page 15 – Line 33: Replace ''for'' with ''from'' before ''1979 to 2016''.

Page 16 – Line 1-2: Replace ''It shows'' with ''We found''.

Page 16 – Line 3: Replace ''ice SIF by'' with ''in SIF of''.

Page 16 – Line 5: Do you mean ''thickness of thick sea ice''?

Page 16 – Line 5-6: Replace ''the actual'' with ''a weaker'' and remove ''to be weaker'' after ''SIF trend''. Also replace ''to'' with ''may'' before ''be stronger''.

Page 16 – Line 9: Replace ''value of additional'' with ''amount of the''.

Page 16 – Line 11: Add ''to'' after ''largely contribute''.

Figure 4: In the figure caption: Replace ''and'' with comma after ''(December-April)'' in the first sentence.

Figure 5: In the figure caption: The dotted lines are shown in all panels. Hence, remove "in panels (b) and (d)'' in the last sentence. Also remove comma after "region" in the same sentence.

Figure 7: Please clarify which scale belongs to which graph.

Figure 7: You have plotted MLD in the Greenland Sea, but from the values I assume that you mean within the Greenland Sea gyre/interior basin and not an average over your entire domain/green box? Please clarify.

**References:**

Chafik and Rossby (2019): https://doi.org/10.1029/2019GL082110
Eldevik et al. (2009): https://doi.org/10.1038/ngeo518
Mastropole et al. (2017): https://doi.org/10.1002/2016JC012007
Jeansson et al. (2017): https://doi.org/10.1016/j.dsr.2017.08.013
Asbjørnsen et al. (2019): https://doi.org/10.1029/2018JC014649
Lohmann et al. (2009): https://doi.org/10.1029/2009GL039166
Foukal and Lozier (2017): https://doi.org/10.1002/2017JC012798

---

## Author Response (AR2)

We are grateful for the reviewer for a thorough review of the manuscript and a number of useful suggestions. Below we list the answers to the review.

Summary:
The authors have significantly improved the content and structure of the manuscript, but I still have three major comments (listed under general comments) that I think the authors need to elaborate on before the paper is ready for publication.
I have also included several specific comments and technical corrections on the updated text. Most of these are related to unclear text and English grammar.

General comments:

GC1: I still think that you need to include a statement in the paper about the data availability in the MIZ (especially the generally sparse data coverage on the shelf/shelf break), even though you are not able to address the temperature uncertainty. You can also add, as you mentioned in the previous response, that the additional use of satellite sea-surface data is very relevant and makes your analysis of the ML properties more robust than only including in situ observations, which is a strength of the ARMOR dataset.

Response: We added the phrases to Section 2.3:

"The number of in situ vertical temperature profiles in the MIZ area of the Greenland Sea (Fig. 1) is very limited. Between 1993 and 2016 the number of casts varies from 13 to 350 per year, with the median of 90 casts per year. Even less profiles are performed in the Greenland shelf, which is out of the scope of this study. In the ARMOR dataset, the use of satellite information provides a more precise and detailed picture of spatial and temporal variability of the thermohaline characteristics, than from interpolation of in situ profiles alone (as, for example, it is done in the World Ocean Atlas data-set, https://www.nodc.noaa.gov/OC5/indprod.html), and adds robustness to the results."

GC2: Without discussing or investigating the role of increased atmospheric temperature and changes in ocean-atmosphere heat fluxes you cannot really make the conclusions on page 13 – lines 14-15, page 15 – lines 8-9, and page 16 – lines 10-12. You have shown that an increase in AW heat flux CAN contribute to sea ice volume loss, but not that it actually does. The correlation and trend estimates support your hypothesis, but does not imply causation. I understand that a thorough investigation of the atmospheric component requires some work, and could be an entire separate study. However, if you want to make the conclusions above, you cannot avoid to include some discussion on changes in the atmosphere and its potential role for the observed changes in sea ice volume. You do not necessarily need to include any new analysis, but could base a brief discussion on previous studies (ie. Moore et al. 2015 etc.).

Response: We have estimated the atmospheric heat convergence near the sea-surface in the area of the central Greenland Sea.
The heat convergence is estimated as the sum of atmospheric heat fluxes across the northern, southern, eastern and western boundaries of the Greenland Sea (positive fluxes are in the study region), using ERA-Interim reanalysis. On average, from October to April next year, we obtained always negative atmospheric heat convergence over the Greenland Sea (1000 to 900 GPa) of -120 TW on average, varying from -170 to -90 TW. The negative atmospheric heat convergence is roughly balanced by the integral heat release from the ocean to the atmosphere over the same area on the order of +130±40 TW, assuming the regional mean winter heat release by the ocean of 150±50 W m$^{-2}$ (Moore et al., 2015). The sign is consistent with winter typical winds from the

Arctic or Greenland over the MIZ (see, for example, Greme et al., 2011), being warmed while passing over the region.

However, we agree with the reviewer that less negative atmospheric heat convergence will result in more heat left in the region. In fact, the negative atmospheric heat convergence has tendency to decrease in the absolute value from 1993 to 2016 by about 4 TW, accompanied by a rise of the area-mean winter air temperature by about 1$^{\circ}$C. The oceanic southwards heat advection through 77.5$^{\circ}$N in the upper 200-m layer increases by 1 TW, during the same period. The source of atmospheric warming lies possibly in the northwest, in the south-eastern Fram Strait, - the known region of high oceanic heat flux into the atmosphere (see, for example, Dukhovskoy et al., 2006)

The information above is added to Discussion.

GC3: The discussion on the link between the NAO index and the ocean circulation is improved in the current version of the paper, but since you already include the winter NAO index in fig. 7: what is the correlation between the winter NAO index and the water temperature at Svinøy? Some studies indicate that the ocean circulation has been decoupled from the NAO in periods during recent decades, and that it is more closely linked to the wind stress curl (Lohmann et al., 2009; Foukal and Lozier, 2017; Asbjørnsen et al., 2019).

Response: We agree with the reviewer that NAO-type forcing is not the only factor affecting the interannual variability of the oceanic heat flux in the Nordic Seas. AO, EA and Arctic Dipole patterns may also have their input into the variations. The mechanism of NAO influencing the heat fluxes is that NAO, largely determining the path and intensity of the westerlies and the intensity of the Atlantic drift (Foukal and Lozier, 2017). However, being a large-scale regional pattern, NAOI phase partly depends on variations in the intensity and position of the Azores anticyclone, which is less important for oceanic circulation in the Nordic Seas. The latter may cause variations of NAOI being different from those of the depth of the Islandic minimum. Additionally, not only the intensity, but also the position of the Islandic minimum is important for determining the local winds-stress curl over the Nordic Seas (Foukal and Lozier, 2017), as well as for the circulation intensity.

We also note that the NAO forcing affects the upper ocean temperature through variation of, at least, two different heat fluxes not necessarily coupled: the variation in the ocean-atmosphere heat exchange and that of ocean heat advection. In Moore et al. (2015) an absence of significant correlation of NAO with turbulent heat fluxes from the ocean has been noted for a limited area in the central Greenland Sea. We obtained the same insignificant correlations over the whole Greenland Sea. However, there is a significant correlation between NAOI and oceanic heat advection with the Norwegian current at Svinoy (0.5 for the heat flux integrated over the upper 500-m layer), although there is no correlation between NAOI and water temperature in the section. This is because the advective heat fluxes in the ocean are determined by variations in the intensity of the upper ocean circulation on interannual time scales (Skagseth, 2004; Asbjornsen et al., 2019 and references within). Correlation of NAOI with southward heat flux in the Fram recirculation is also positive, but not significant (0.3). The decrease is due to damping of the advected heat anomalies in the Norwegian Sea by eddy heat transport and ocean-atmosphere exchange (Asbjornsen et al., 2019). However, on the decadal time scales (of particular interest for this study), we observe positive trend in the heat transport along the whole path of the NwAC.

We also agree that EA  pattern may be also important in determining the position of the Islandic minimum (Woollings et al., 2010; Moore and Renfrew, 2012; Foukal and Lozier, 2017), affecting the wind-stress curl pattern over the Nordic Seas.

The information above is added to Discussion.

Specific comments:

Page 1 – Line 16-17: Only refer to Marshall and Schott (1999) or find more appropriate references than Visbeck et al. (1995) and Brakstad et al. (2019) to this line/statement.

Response: The reference Visbeck et al. (1995) is now removed. Brakstad et al. (2019) is left as it describes the latest advances in water mass formations due to deep convection in the Greenland Sea. We also add reference (Buckley and Marshall, 2016), where AMOC water mass transport from the convection sites is discussed.

Page 1 – Line 18: I do not think that any of these references state that more than 50% of the deep AMOC originates from "the Greenland Sea". It should be "the Nordic Seas" (ie. Chafik and Rossby, 2019). The fraction of contribution from the Greenland Sea to the Nordic Seas overflow water (the dense water that spill south across the Greenland-Scotland Ridge) is still not fully known. Some references looking at the origin of the Nordic Seas overflow waters are: Eldevik et al. (2009), Mastropole et al. (2017), and Jeansson et al. (2017).

Response: We removed this phrase.

Page 2 – Line 1-2: What drives the remaining 35% of the freshwater anomaly? Is this due to variability in the northward flowing AW?

Response: The rest of the FW flux should come from the ice melt in the Greenland Sea and from variations in salt transport with the recirculating Atlantic water.

Page 2 – Line 14-16: I would rather state that the Greenland Sea Intermediate Water and the Greenland Sea Deep Water (GSDW) are formed by wintertime convection in the central Greenland Sea. Note also that the GSDW has not been ventilated/formed since before the 1980s.

Response: We do not agree. Water masses, entering from Fram Strait influence the Greenland Sea Intermediate and Deep water (Langehaug and Falck, 2012). In particular, this is well seen for the Greenland Sea Intermediate water, which, in the absence of winter convection becomes warmer and mode saline due to mixing with the recirculating AW.
Also our estimates suggest that 2000m has been reached in the central Greenland Sea at least at 2008, 2011-2013. Wadhams et al. (2004) found a 3000-m deep chimney in winters 2001/2003. This means the GSDW has been ventilated.

Page 2 – Line 17: Latarius and Quadfasel (2016) do not show MLDs exceeding 2000m. I think this statement is incorrect (see also Brakstad et al., 2019).

Response: We agree that expression "often exceeds 2000 m" is two strong, as this depth is not reached often. However, our estimates suggest that 2000 m has been reached in the central Greenland Sea at least at 2008, 2011-2013 (Fedorov et al., 2018; Bashmachnikov et al., 2019). This is consistent with Johannessen et al. (1991, 2005) and Wadhams et al. (2002), who observed MLD over 2000 m in the Greenland Sea.
Latarius and Quadfasel (2016) used only Argo floats, therefore the authors missed some of the local chimneys with the maximum convection depth. Brakstad et al. (2019) used method by Kara et al. (2003) and by Lorbacher et al. (2006). Both are not optimal for an automatic detection of

deep MLD in a weakly stratified deep subpolar seas (see Fedorov et al., 2018; Bashmachnikov et al., 2019).

We change the phrase to "often exceeds 1500 m" and added reference to Wadhams et al. (2002)

Page 5 – Line 9: It is still not clear from the text what you mean with "different weights". Please add the information from your response to the text (ie: "taken with different weights based on the inverse distance and type of measurement (in situ observations were given higher weights).").

Response: The suggested information is added to the text.

Page 7- Line 19: There is a bias towards deeper mixed layers in the central Greenland Sea. The estimated MLDs in fig. 7 are always deeper than 600m, while both Latarius and Quadfasel (2016) and Brakstad et al. (2019) observe MLDs shallower than 500m in 2003 and 2012 for instance. This should be noted.

Response: The following text is added:
"All the results show an increase of the convection depth from the mid-1990-s to the 2000-s. There are some minor differences in the absolute values of MLD which arise from the use of different data sets (e.g. Latarius and Quadfasel (2016) used only Argo floats) and methodologies for MLD detection. These minor differences do not break the tendency for the maximum winter MLD to increase since mid-1990s."

Page 8 – Line 7: What do you mean by "is conditioned by study of the role of heat fluxes on melting sea ice"?

Response:  The sentence is rephrased:  "The reference temperature was set to the sea ice melt temperature in order to investigate the contribution of ocean heat fluxes to sea ice melt. "

Page 10 – Line 3: Tendencies in heat? Please clarify.

Response: The sentence is rephrased: "Our analysis shows that the obtained tendencies of increase of water temperature with time, derived in the next paragraphs, are largely independent from the choice of the water layer"

Page 10 – Line 8-9: Here you state that the ocean always melt sea ice in the MIZ. This sounds like a contradiction to line 12 on the same page where you state that sea ice is formed locally in winter. I know that you, in the first line (8-9), talk about the mean conditions over the upper 50m in the entire MIZ, but I would rephrase this line to avoid confusion. At least make it clear that the temperature can be well below 0 degrees C in certain regions of the MIZ, and that your average is affected by the defined area of investigation (you average over water temperatures > 1 degrees C in some places (ie. AW in the northeast, fig. 5) which is not representative for the real MIZ characteristics).

Response: The sentence is rephrased:
"When averaged over the fixed region, corresponding to the mean winter MIZ area (Figure 1), the mixed layer seawater temperature is always above the freezing point, i.e., overall, the ocean melts the sea ice in this area all the year-round."

Page 11 – Line 25: Lauvset et al. (2018) do not examine the MIZ and heat fluxes toward the sea ice. They investigate how increased salinity and temperature in the northward flowing AW have

led to deeper mixed layers in the Greenland Sea gyre. It would be more appropriate to include this reference in the next line (25-26).

Response: Thank you, corrected.

Page 13 – Line 31: Clarify what you mean by "All the processes".

Response: Thank you, corrected to "All those processes"

Page 14 – Line 5-11: These lines are not necessary since the same information is repeated/elaborated in the next paragraph. Please remove these lines and make a smooth transition to the discussion of the NAO.

Response: Here we left the text unhanged. The mentioned lines are general facts which are used to elaborate on the results in the next paragraph.

Page 14 – Line 15-16: Which regional studies? Please include reference.

Response: Some references from below are added to the end of this phrase.

Page 16 – Line 12-15: These lines repeat lines 7-12 and are therefore not necessary. Please remove.
Response: Thank you, the lines were removed.

Technical corrections:

Generally: Remove ''the'' in ''through the Fram Strait/Denmark Strait''.
Response: Corrected.

Page 1 – Line 3: Replace ''sea ice volume (SIV)'' with ''SIV'' since SIV is defined in line 1 already.
Response: Thank you, corrected.

Page 1 – Line 18: Replace ''originated'' with ''originates''.
Response: Thank you, corrected.

Page 2 – Line 6: Use ''The'' instead of ''A'' before ''general surface circulation''.
Response: Thank you, corrected.

Page 2 – Line 7: Remove comma after ''temperature'' and add comma after ''Polar Water (PW)'' and before ''and the Atlantic Water (AW)''.
Response: Thank you, corrected.

Page 2 – Line 9-10: I would first state where the maximum PW is found then that it quickly decreases in the off shelf direction.

Response: Rephrased: "The maximum PW content is found in the upper 200 m of the Greenland shelf, it quickly decreases in the off shelf direction"

Page 2 – Line 11: Remove ''s'' in ''centrals''.
Response: Thank you, corrected.

Page 2 – Line 12 and 13: Do you mean ''Greenland Sea Intermediate Water''?
Response: Thank you, corrected.

Page 2 – Line 20: Use ''are'' instead of ''is'' before ''primarily controlled''.
Response: Thank you, corrected.

Page 2 – Line 23: Should be ''winds that drive'' not ''drives'' since it refers to "winds". Single vs. plurals.
Response: Thank you, corrected.

Page 2 – Line 24: Remove ''between''. It is written twice.
Response: Thank you, corrected.

Page 2 – Line 28: Replace ''that'' with ''since it'' before ''explains a higher fraction''.
Response: Thank you, corrected.

Page 2 – Line 35: Replace ''is'' with ''was'' after ''the Odden sea ice tongue'' since it is ~20 years since this last occurred.
Response: Thank you, corrected.

Page 3 – Line 1: Remove ''and'' before ''northwest of Jan Mayen''?
Response: Thank you, corrected.

Page 3 – Line 5: Add ''the'' before ''large''.
Response: Thank you, corrected.

Page 3 – Line 15: Replace ''The overall'' with ''An overall''.
Response: Thank you, corrected.

Page 3 – Line 16: Add ''In particular" or "particularly" to highlight the large changes in the Odden ice tongue area?
Response: Thank you, corrected.

Page 3 – Line 21: Add ''the'' before ''1990s''.
Response: Thank you, corrected.

Page 3 – Line 22: Add ''is'' before ''not possible''.
Response: Thank you, corrected.

Page 3- Line 32: Use ''daily sea ice concentration'' instead of ''sea daily ice concentration''.
Response: Thank you, corrected.

Page 3 – Line 33: Remove ''the'' before ''sea-surface temperature''.
Response: Thank you, corrected.

Page 4 – Line 7: Remove ''s'' in ''affects'' and insert ''sea ice'' before ''thickness and volume''.
Response: Thank you, corrected.

Page 4 – Line 17: Remove ''In the Greenland Sea'' at the beginning of the sentence since it is written at the end of the sentence as well.
Response: Thank you, corrected.

Page 4 – Line 30: Remove ''In addition,'' since ''also'' is used later in the sentence.
Response: Thank you, corrected.

Page 4 – Line 31: Replace ''timeliness'' with ''timing'' and remove ''the'' before ''satellite passes''.
Response: Thank you, corrected.

Page 5 – Line 14: Remove 'the'' before ''satellite altimetry''.
Response: Thank you, corrected.

Page 5 – Line 21-23: Replace ''However,'' with ''Since'', add comma after ''2000s'', remove ''. In this paper'', and use capital ''C'' in ''West Spitsbergen Current''.
Response: Thank you, corrected.

Page 5 – Line 25: Add ''the'' before ''cold season''.
Response: Thank you, corrected.

Page 5 – Line 28: Use ''a'' instead of ''the'' before ''kriging''.
Response: There is no article before "kriging" in the text.

Page 5 – Line 28-29: Replace ''over the months the most densely covered with data'' with ''over the months with densest data coverage''.
Response: Thank you, replaced.

Page 6 – Line 20: ''20E'' should be ''20W''.
Response: Thank you, corrected.

Page 6 – Line 20: Replace ''the'' with ''a'' before ''sea ice volume''.
Response: Thank you, corrected.

Page 6 – Line 21: Replace "access" with ''assess''.
Response: Thank you, corrected.

Page 6 – Line 24: Use ''gate locations'' instead of ''gates locations''.
Response: Thank you, corrected.

Page 7 – Line 3: Replace ''and boarder on the east'' with ''and by the boarder in the east''.
Response: Thank you, corrected.

Page 7 – Line 8-9: There is something strange with this sentence. It makes more sense if ''to'' before ''sea ice volume loss'' is removed.
Response: The sentence was rephrased.

Page 7 – Line 17-18: Remove ''the'' before ''weakly'', before ''Dukhovskoy's'', and after ''mean distribution of''.
Response: Thank you, corrected.

Page 7 – Line 25: Remove ''a'' before ''new ice''.
Response: Thank you, corrected.

Page 7 – Line 25: Replace ''All these distant factors'' with ''All of these factors''.
Response: Thank you, corrected.

Page 7 – Line 27-28: Replace ''the ice volume in the sea'' with ''the sea ice volume''.

Response: Thank you, corrected.

Page 7 – Line 28: Replace ''define'' with ''examine'' after ''In this study we''.

Response: Thank you, corrected.

Page 8 – Line 6: Insert ''and'' before ''v is current velocity''.

Response: Thank you, corrected.

Page 8 – Line 10: Remove ''in region, PIOMAS'' and add comma before ''monthly''.

Response: Thank you, corrected.

Page 8 – Line 12: Should be ''values'' instead of ''value''.

Response: Thank you, corrected.

Page 8 – Line 24: Replace ''off'' with ''of''.

Response: Thank you, corrected.

Page 9 – Line 9: Insert ''the'' before ''long-term''.

Response: Thank you, corrected.

Page 9 – Line 19: Remove ''a'' before ''half of the years''.

Response: Thank you, corrected.

Page 9 – Line 22: Remove ''s'' in ''trends''.

Response: Thank you, corrected.

Page 9 – Line 26: Remove ''in'' before ''and sea ice volume flux''.

Response: Thank you, the phrase is chnages.

Page 9 – Line 29: Remove ''s'' in ''shows'' after ''estimates of winter MLD''.

Response: Thank you, corrected.

Page 10 – Line 6: Add ''a'' before ''maximum''.

Response: Thank you, corrected.

Page 10 – Line 7: Remove ''a'' before ''somewhat weaker''.

Response: Thank you, corrected.

Page 10 – Line 9: Remove ''the'' after ''ocean melts''.

Response: Thank you, corrected.

Page 10 – Line 12: Replace ''formed by'' with ''at'', and add ''the'' before ''winter cooling''.

Response: Thank you, corrected.

Page 10 – Line 12: ''localy'' should be ''locally''.

Response: Thank you, corrected.

Page 10 – Line 12-17: I would first explain the evolution of the 2degC isotherm (lines 15-17), then explain that this evolution is consistent for different isotherms etc. (lines 12-15). The beginning of the first sentence (line 12) is also a bit confusing and unnecessary. I would rather start this sentence with ''The tendency of the isotherm to approach the shelf break is consistent for different isotherms etc.''.

Response: Thank you, corrected.

Page 10 – Line 16: Remove ''s'' in ''westwards''.

Response: Thank you, corrected.

Page 10 – Line 25: Remove comma after ''area''.

Response: Thank you, corrected.

Page 10 – Line 26: Rather start this sentence with: "The decreasing temperature in both of these areas is consistent with …."

Response: Thank you, corrected.

Page 10 – Line 30: Remove ''s'' in ''trends''.

Response: Thank you, corrected.

Page 11 – Line 10: Remove comma before ''accumulated'' and after ''during summer''.

Response: Thank you, corrected.

Page 11 – Line 15: Remove ''effect''

Response: Thank you, corrected.

Page 11 – Line 17: Remove ''the'' before ''eddy formation'' and replace ''for the'' with ''during''.

Response: Thank you, corrected.

Page 11 – Line 18: ''northely'' should be ''northerly''.

Response: Thank you, corrected.

Page 11 – Line 18: Replace ''off'' with ''seaward''. Also replace "increase in bottom" with "increased bottom".

Response: Thank you, corrected.

Page 11 – Line 23: Remove ''the'' before ''winter mixing''.

Response: Thank you, corrected.

Page 11 – Line 25: Replace ''of'' with ''in'' before ''MLD''.

Response: Thank you, corrected.

Page 11 – Line 31: Add ''and'' before ''the MIZ area''.

Response: Thank you, corrected.

Page 11 – Line 31: Replace ''computations show'' with ''computations give''.

Response: Thank you, corrected.

Page 12 – Line 1: Replace ''by'' with ''of''.

Response: Thank you, corrected.

Page 12 – Line 3: Remove ''of'' after ice density.

Response: Thank you, corrected.

Page 12 – Line 8: Replace ''contacting with'' with ''to reach''. Also remove comma before ''the autumn warming''.

Response: Thank you, corrected.

Page 12 – Line 9: Replace ''the amount heat far exceeding the amount, sufficient'' with ''more than enough heat to account''

Response: Thank you, corrected.

Page 12 – Line 17: Use ''than'' instead of ''that'' after ''the trend is lower''.

Response: Thank you, corrected.

Page 12 – Line 19: Replace ''be also'' with ''also be''.

Response: Thank you, corrected.

Page 12 – Line 23: "myltiyear'' should be ""multiyear".

Response: Thank you, corrected.

Page 12 – Line 32: Replace "are the main gates" with ''is the main gate''.

Response: Thank you, corrected.

Page 13 – Line 6: Remove ''the'' before ''sea ice volume'' and add ''the'' before ''Greenland Sea''.

Response: Thank you, corrected.

Page 13 – Line 24: Remove ''for'' before ''in the Nansen Basin''.

Response: Thank you, corrected to "for the Nansen Basin".

Page 14 – Line 14: Replace ''leading to'' with ''which results in''.

Response: Thank you, corrected.

Page 14 – Line 18: Remove ''of'' before ''the cyclonic circulation''.

Response: Thank you, corrected.

Page 14 – Line 22: Add ''The'' before ''NAO phase''.

Response: Thank you, corrected.

Page 14 – Line 25: Add ''of'' before ''PW''.

Response: Thank you, corrected.

Page 14 – Line 26: Remove parenthesis before ''Blindheim et al.''.

Response: Thank you, corrected.

Page 14 – Line 29: Remove ''a'' before ''higher heat fluxes''.

Response: Thank you, corrected.

Page 14 – Line 30: Insert ''the'' before ''1970s''.

Response: Thank you, corrected.

Page 14 – Line 34: Replace ''of'' with ''on the'' before ''order of''.

Response: Thank you, corrected.

Page 15 – Line 11: Replace ''of'' with ''for'' before ''the shelf area''. Also remove parenthesis before ''Alekseev et al.''.

Response: Thank you, corrected.

Page 15 – Line 12-13: Replace ''as well as along the EGC, as well as along the NwAC, increases during recent decades'' with ''as well as along the EGC and in the NwAC, has increased during recent decades''.

Response: Thank you, corrected.

Page 15 – Line 13: Remove parenthesis around ''d'' in ''(Fig. 5(d))''. Also the case on line 25.

Response: Thank you, corrected.

Page 15 – Line 24: Insert ''the'' before ''1990s''.

Response: Thank you, corrected.

Page 15 – Line 28: Insert ''the'' before ''1980s''.

Response: Thank you, corrected.

Page 15 – Line 33: Replace ''for'' with ''from'' before ''1979 to 2016''.

Response: Thank you, corrected.

Page 16 – Line 1-2: Replace ''It shows'' with ''We found''.

Response: Thank you, corrected.

Page 16 – Line 3: Replace ''ice SIF by'' with ''in SIF of ''.

Response: Thank you, corrected.

Page 16 – Line 5: Do you mean ''thickness of thick sea ice''?

Response: Thank you, corrected.

Page 16 – Line 5-6: Replace ''the actual'' with ''a weaker'' and remove ''to be weaker'' after ''SIF trend''. Also replace ''to'' with ''may'' before ''be stronger''.

Response: Thank you, corrected.

Page 16 – Line 9: Replace ''value of additional'' with ''amount of the''.

Response: Thank you, corrected.

Page 16 – Line 11: Add ''to'' after ''largely contribute''.

Response: Thank you, corrected.

Figure 4: In the figure caption: Replace ''and'' with comma after ''(December-April)'' in the first sentence.

Response: Thank you, corrected.

Figure 5: In the figure caption: The dotted lines are shown in all panels. Hence, remove "in panels (b) and (d)" in the last sentence. Also remove comma after "region" in the same sentence.

Response: Thank you, corrected.

Figure 7: Please clarify which scale belongs to which graph.

Response: The scales are given in captions. In the same panel all lines are in scale. We slightly changed the figure caption.

Figure 7: You have plotted MLD in the Greenland Sea, but from the values I assume that you mean within the Greenland Sea gyre/interior basin and not an average over your entire domain/green box? Please clarify.

Response: Yes, this is over the Greenland Sea. The caption is modified.

[revised manuscript text omitted]